# MAST: MODEL-AGNOSTIC SPARSIFIED TRAINING

**Yury Demidovich    Grigory Malinovsky    Egor Shulgin    Peter Richtárik**
King Abdullah University of Science and Technology (KAUST), Saudi Arabia*

## ABSTRACT

We introduce a novel optimization problem formulation that departs from the conventional way of minimizing machine learning model loss as a black-box function. Unlike traditional formulations, the proposed approach explicitly incorporates an initially pre-trained model and random sketch operators, allowing for sparsification of both the model and gradient during training. We establish the insightful properties of the proposed objective function and highlight its connections to the standard formulation. Furthermore, we present several variants of the Stochastic Gradient Descent (SGD) method adapted to the new problem formulation, including SGD with general sampling, a distributed version, and SGD with variance reduction techniques. We achieve tighter convergence rates and relax assumptions, bridging the gap between theoretical principles and practical applications, covering several important techniques such as Dropout and Sparse training. This work presents promising opportunities to enhance the theoretical understanding of model training through a sparsification-aware optimization approach.

## 1 INTRODUCTION

Efficient optimization methods have played an essential role in the advancement of modern Machine Learning (ML), given that many supervised problems ultimately reduce to the task of minimizing an abstract loss function — a process that can be formally expressed as:

$$\min_{x \in \mathbb{R}^d} f(x), \tag{1}$$

where $f : \mathbb{R}^d \to \mathbb{R}$ is a loss function of a model with parameters/weights $x \in \mathbb{R}^d$. The problem (1) has been comprehensively analyzed within the domain of optimization literature. Within the ML community, substantial attention has been directed towards studying this problem, particularly in the context of the Stochastic Gradient Descent (SGD) method (Robbins & Monro, 1951). SGD stands as a foundational and highly effective optimization algorithm within the realm of ML (Bottou et al., 2018). The pervasive use of SGD in the field attests to its versatility and success in training a diverse array of models (Goodfellow et al., 2016). Notably, contemporary deep learning practices owe a substantial debt to SGD, as it is the cornerstone for many state-of-the-art training techniques (Sun, 2020).

While problem (1) has been of primary interest in mainstream optimization research, this approach is not always best suited for representing recent ML techniques, such as sparse/quantized training (Wu et al., 2016; Hoefler et al., 2021), resource-constrained distributed learning (Caldas et al., 2018; Wang et al., 2018), model personalization (Smith et al., 2017; Hanzely & Richtárik, 2020; Mansour et al., 2020), and meta-learning (Schmidhuber, 1987; Finn et al., 2017). Although various attempts have been made to analyze some of these settings (Khaled & Richtárik, 2019; Lin et al., 2019; Mohtashami et al., 2022), there is still no satisfactory optimization theory that can explain the success of these techniques in deep learning. Previous works analyze variants of SGD trying to solve problem (1), which often results in vacuous convergence bounds and/or overly restrictive assumptions on the class of functions and algorithms involved (Shulgin & Richtárik, 2024). We assert that these issues arise due to mismatches between the method used and the problem formulation being solved.

In this work, to address the issues mentioned above, we propose a new optimization problem formulation called Model-Agnostic Sparsified Training (MAST):

$$\min_{x \in \mathbb{R}^d} \left[ f_{\mathcal{D}}(x) \stackrel{\text{def}}{=} \mathbb{E}\left[f_{\mathbf{S}}(x)\right] \right], \tag{2}$$

---

*Contacts: {yury.demidovich, grigorii.malinovskii, egor.shulgin, peter.richtarik}@kaust.edu.sa

where $f_{\mathbf{S}}(x) \stackrel{\text{def}}{=} f(v + \mathbf{S}(x - v))$ for $v \in \mathbb{R}^d$ (e.g., a pre-trained model, possibly compressed), $\mathbf{S} \in \mathbb{R}^{d \times d}$ is a random matrix (i.e., a **sketch**) sampled from distribution $\mathcal{D}$.

When $v = 0$, the function takes the following form: $f_{\mathbf{S}}(x) = f(\mathbf{S}x)$. This scenario may be considered a process in which the architecture requires training from scratch, with quantization as a central consideration. Such an approach involves a substantial increase in training time and the necessity of hyperparameter tuning to achieve effectiveness. In terms of theory, formulation (2) can be viewed as a nested (stochastic) composition optimization (Bertsekas, 1977; Polyak, 1979; Ermoliev, 1988). Moreover, our formulation is partially related to the setting of splitting methods (Condat et al., 2023). However, due to their generality, other setups do not consider the problem instance's peculiarities and focus on other applications. Conversely, when $v$ is non-zero and pre-trained weights are utilized, the newly formulated approach can be interpreted as acquiring a "meta-model" $x$. Solving problem (2) then ensures that the sketched model $v + \mathbf{S}(x - v)$ exhibits strong performance on average. This interpretation shares many similarities with Model-Agnostic Meta-Learning (Finn et al., 2017).

We argue that this framework may be better suited for modeling various practical ML techniques, as discussed below. Furthermore, our proposed framework facilitates thorough theoretical analysis and holds potential independent interest for a broader audience. While our analysis and algorithms work for a quite general class of sketches, we focus on applications that are relevant for sparse $\mathbf{S}$.

## 1.1 Motivating examples

**Dropout** (Hanson, 1990; Hinton et al., 2012; Frazier-Logue & Hanson, 2018) is a regularization technique initially introduced to prevent overfitting in neural networks by dropping some of the model's units (activations) during training. Later, this approach was generalized to incorporate Gaussian noise (Wang & Manning, 2013; Srivastava et al., 2014) (instead of Bernoulli masks) and zeroing the weights via DropConnect (Wan et al., 2013) or entire layers (Fan et al., 2019). It was also observed (Srivastava et al., 2014) that training with Dropout induces sparsity in the activations. In addition, using the Dropout-like technique (Gomez et al., 2019; LeJeune et al., 2021) can make the resulting network more amenable to subsequent sparsification (via pruning) before deployment. Modern DL has seen a huge increase in the size of models (Villalobos et al., 2022). This has resulted in growing energy and computational costs, necessitating optimizations of neural networks' training pipeline (Yang et al., 2017). Among others, pruning and sparsification were proposed as effective techniques due to the overparametrization properties of large models (Chang et al., 2021).

**Sparse training** algorithms (Mocanu et al., 2016; Guo et al., 2016; Mocanu et al., 2018), in particular, suggest working with a smaller subnetwork during every optimization step. This increases efficiency by reducing the model size (via compression), which naturally brings memory and computation acceleration benefits to the inference stage. Moreover, sparse training has recently been shown to speed up optimization by leveraging sparse backpropagation (Nikdan et al., 2023).

**On-device learning** also creates a need for sparse or submodel computations due to the memory, energy, and computational constraints of edge devices. In settings like cross-device federated learning (Konečný et al., 2016; McMahan et al., 2017; Kairouz et al., 2021), models are trained in a distributed way across a population of heterogeneous nodes, such as mobile phones. The heterogeneity of the clients' hardware makes it necessary to adapt the (potentially large) server model to the needs of low-tier devices (Caldas et al., 2018; Bouacida et al., 2021; Horváth et al., 2021).

**Contributions.** The main results of this work include:

• A rigorous formalization of a new optimization formulation, as shown in Equation (2), which can encompass various important practical settings as special cases, such as Dropout and Sparse training.

• In-depth theoretical characterization of the proposed problem's properties, highlighting its connections to the standard formulation in Equation (1). Notably, our problem is efficiently solvable with practical methods.

• The development of optimization algorithms that naturally emerge from the formulation in Equation (2), along with insightful convergence analyses in non-convex and (strongly) convex settings.

• The generalization of the problem and methods to the distributed scenario, expanding the range of applications even further, including scenarios like IST and Federated Learning.

• An experimental study of the proposed algorithms and MAST properties in the context of machine learning, highlighting the advantages of the proposed approach.

**Paper organization.** We introduce the basic formalism and discuss sketches in Section 2. The properties of the suggested formulation (2) are analyzed in Section 3. Section 4 contains convergence results for full-batch, stochastic, and variance-reduced methods for solving problem (2). Extensions to the distributed case are presented in Section 5. Section 6 describes some of our experimental results. The last Section 7 concludes the paper and outlines potential directions of future work. All proofs are provided in the Appendix.

## 2    SKETCHES

(Stochastic) gradient-based methods are mostly used to solve problem (1). Applying this paradigm to our formulation (2) requires computing the gradient of $f_{\mathcal{D}} = \mathbb{E}[f_{\mathbf{S}}]$. In the case of the general distribution $\mathbf{S} \sim \mathcal{D}$, such computation may not be possible due to expectation $\mathbb{E}$. Therefore, practical algorithms typically rely on gradient estimates. An elegant property of the MAST problem (2) is that the gradient estimator takes the form

$$\nabla f_{\mathbf{S}}(x) = \mathbf{S}^\top \nabla f(v + \mathbf{S}(x - v)), \tag{3}$$

due to the chain rule, as matrix $\mathbf{S}$ is independent of $x$. Sketches/matrices $\mathbf{S}$ are random and sampled from distribution $\mathcal{D}$. Note that estimator (3) sketches both the model $x$ and the gradient of $f$. Next, we explore some of the sketches' important properties and give practical examples.

**Assumption 1.** *The sketching matrix $\mathbf{S}$ satisfies:*

$$\mathbb{E}[\mathbf{S}] = \mathbf{I}, \qquad \text{and} \qquad \mathbb{E}[\mathbf{S}^\top \mathbf{S}] \text{ is finite}, \tag{4}$$

*where $\mathbf{I}$ is the identity matrix. Note that $\mathbf{S}^\top \nabla f(x)$ is an unbiased estimator of the gradient $\nabla f(x)$.*

Denote $L_{\mathbf{S}} \stackrel{\text{def}}{=} \lambda_{\max}(\mathbf{S}^\top \mathbf{S})$, $\mu_{\mathbf{S}} \stackrel{\text{def}}{=} \lambda_{\min}(\mathbf{S}^\top \mathbf{S})$, and $L_{\mathcal{D}} \stackrel{\text{def}}{=} \lambda_{\max}(\mathbb{E}[\mathbf{S}^\top \mathbf{S}])$, $\mu_{\mathcal{D}} \stackrel{\text{def}}{=} \lambda_{\min}(\mathbb{E}[\mathbf{S}^\top \mathbf{S}])$, where $\lambda_{\max}$ and $\lambda_{\min}$ represent the largest and smallest eigenvalues. Clearly, $L_{\mathbf{S}} \geq \mu_{\mathbf{S}} \geq 0$ and $L_{\mathcal{D}} \geq \mu_{\mathcal{D}} \geq 0$. If Assumption 1 is satisfied, then $\mathbb{E}[\mathbf{S}^\top \mathbf{S}] \succeq \mathbf{I}$, which means that $\mu_{\mathcal{D}} \geq 1$ and

$$\|x\|^2 \leq \mu_{\mathcal{D}} \|x\|^2 \leq \mathbb{E}\left[\|\mathbf{S}x\|^2\right] \leq L_{\mathcal{D}} \|x\|^2.$$

### 2.1    DIAGONAL SKETCHES

Let $c_1, c_2, \ldots, c_d$ be a collection of random variables and define a matrix with $c_i$-s on the diagonal

$$\mathbf{S} = \text{Diag}(c_1, c_2, \ldots, c_d), \tag{5}$$

which satisfies Assumption 1 when $\mathbb{E}[c_i] = 1$ and $\mathbb{E}[c_i^2]$ is finite for every $i$.

The following example illustrates how our framework can be used to model Dropout.

**Example 1.** *The **independent Bernoulli sparsification** operator is defined as a diagonal sketch (5), where every $c_i$ is an (independent) scaled Bernoulli random variable:*

$$c_i = \begin{cases} 1/p_i, & \text{with probability } p_i \\ 0, & \text{with probability } 1 - p_i \end{cases}, \tag{6}$$

*for $p_i \in (0, 1]$ and $i \in [d] \stackrel{\text{def}}{=} \{1, \ldots, d\}$.*

It can be shown that for independent Bernoulli sparsifiers,

$$L_{\mathcal{D}} = \max_i p_i^{-1} \stackrel{\text{def}}{=} p_{\min}^{-1}, \quad \mu_{\mathcal{D}} = \min_i p_i^{-1} \stackrel{\text{def}}{=} p_{\max}^{-1}. \tag{7}$$

Notice that when $p_i \equiv p$, $i \in [d]$, and $v = 0$, gradient estimator (3) results in a sparse update as $\mathbf{S}^\top \nabla f(\mathbf{S}x)$ drops out $d(1 - p)$ (on average) components of model weights and the gradient. The difference from the Dropout described by Hinton et al. (2012) is that they do not use scaling $1/p_i$ in equation (6) during training to ensure unbiasedness. Experimental comparison is presented in Appendix H.2.4.

Next, we show another practical example of a random sketch often used for reducing communication costs in distributed learning (Konečný et al., 2016; Wangni et al., 2018; Stich et al., 2018).

**Example 2.** *Random $K$ sparsification (in short, Rand$-K$ for $K \in [d]$) operator is defined by*

$$\mathbf{S}_{Rand-K} \overset{def}{=} \frac{d}{K} \sum_{i \in S} e_i e_i^\top, \tag{8}$$

*where $e_1, \ldots, e_d \in \mathbb{R}^d$ are standard unit basis vectors, and $S$ is a random subset of $[d]$ sampled from the uniform distribution over all subsets of $[d]$ with cardinality $K$.*

Rand$-K$ belongs to the class of diagonal sketches (5). $\mathbf{S}x$ preserves $K$ non-zero coordinates out of total $d$ coordinates. Since $\mathbb{E}\left[\mathbf{S}^\top \mathbf{S}\right] = \mathbf{I} \cdot d/K$, this sketch satisfies $L_\mathcal{D} = \mu_\mathcal{D} = d/K$. This example is suitable for modeling fixed budget sparse training when the proportion $(K/d)$ of network parameters being updated remains constant during optimization (Mocanu et al., 2018; Evci et al., 2020).

## 3 PROBLEM PROPERTIES

We show that the proposed formulation (2) inherits the smoothness and convexity properties of the original problem (1). Let us introduce the most standard assumptions in the optimization field.

**Assumption 2.** *Function $f$ is differentiable and $L_f$-smooth, i.e., there is $L_f > 0$ such that*

$$f(x + h) \leq f(x) + \langle \nabla f(x), h \rangle + \frac{L_f}{2} \|h\|^2 \quad \forall x, h \in \mathbb{R}^d.$$

*We also require $f$ to be lower bounded by $f^{\inf} \in \mathbb{R}$.*

**Assumption 3.** *Function $f$ is differentiable and $\mu_f$-strongly convex, i.e., there is $\mu_f > 0$ such that*

$$f(x + h) \geq f(x) + \langle \nabla f(x), h \rangle + \frac{\mu_f}{2} \|h\|^2 \quad \forall x, h \in \mathbb{R}^d.$$

Next, we show how the choice of the sketch $\mathbf{S}$ affects the smoothness parameters of $f_\mathbf{S}$ and $f_\mathcal{D}$.

**Lemma 1** (Consequences of $L_f$-smoothness). *If $f$ is $L_f$-smooth, then*

(i) $f_\mathbf{S}$ is $L_{f_\mathbf{S}}$-smooth with $L_{f_\mathbf{S}} \leq L_\mathbf{S} L_f$.

(ii) $f_\mathcal{D}$ is $L_{f_\mathcal{D}}$-smooth with $L_{f_\mathcal{D}} \leq L_\mathcal{D} L_f$.

(iii) $f_\mathcal{D}(x) \leq f(x) + \frac{(L_\mathcal{D}-1)L_f}{2} \|x - v\|^2 \quad \forall x \in \mathbb{R}^d$.

In particular, property $(iii)$ in Lemma 1 demonstrates that the gap between the sketched loss $f_\mathcal{D}$ and the original function $f$ depends on the model weights and the smoothness parameter of function $f$.

**Lemma 2** (Consequence of Convexity). *If $f$ is convex, then $f_\mathcal{D}$ is convex and $f_\mathcal{D}(x) \geq f(x)$.*

It shows that the convexity of $f$ is preserved, and the "sketched" loss is always greater than the original loss. Moreover, Lemma 2 (along with other results in this section) offers a huge advantage of the proposed problem formulation over the sparsification-promoting alternatives based on $\ell_0$-norm regularization (Louizos et al., 2018; Peste et al., 2021), that make the problem hard to solve.

**Lemma 3** (Consequences of $\mu_f$-convexity). *If $f$ is $\mu_f$-convex, then*

(i) $f_\mathbf{S}$ is $\mu_{f_\mathbf{S}}$-convex with $\mu_{f_\mathbf{S}} \geq \mu_\mathbf{S} \mu_f$.

(ii) $f_\mathcal{D}$ is $\mu_{f_\mathcal{D}}$-convex with $\mu_{f_\mathcal{D}} \geq \mu_\mathcal{D} \mu_f$.

(iii) $f_\mathcal{D}(x) \geq f(x) + \frac{(\mu_\mathcal{D}-1)\mu_f}{2} \|x - v\|^2 \quad \forall x \in \mathbb{R}^d$.

As a consequence, we get the following result for the condition number of the proposed problem:

$$\kappa_{f_\mathcal{D}} \overset{def}{=} \frac{L_{f_\mathcal{D}}}{\mu_{f_\mathcal{D}}} \leq \frac{L_\mathcal{D} L_f}{\mu_\mathcal{D} \mu_f} = \frac{L_\mathcal{D}}{\mu_\mathcal{D}} \kappa_f. \tag{9}$$

Therefore, $\kappa_{f_\mathcal{D}} \leq \kappa_f \cdot L_\mathcal{D}$ as $\mu_\mathcal{D} \geq 1$. Thus, the resulting condition number may increase, which indicates that $f_\mathcal{D}$ may be harder to optimize, which agrees with the intuition that compressed training is harder (Evci et al., 2019). In addition, for independent Bernoulli sparsifiers (6),

$$\kappa_\mathcal{D} \overset{def}{=} L_\mathcal{D}/\mu_\mathcal{D} = p_{\max}/p_{\min}, \tag{10}$$

which shows that the upper bound on the ratio $\kappa_{f_{\mathcal{D}}}/\kappa_f$ can be made as large as possible by choosing a small enough $p_{\min}$. At the same time, $\kappa_{\mathcal{D}} = 1$ for classical Dropout: $p_i \equiv p$, $i \in [d]$, indicating that training with Dropout may be no harder than optimizing the original model.

**Relation between $f$ and $f_{\mathcal{D}}$ minima.** Let $\mathcal{X}^\star$ be the solutions to problem 1, and $\mathcal{X}_{\mathcal{D}}^\star$ the solutions of the new MAST problem (2). We now show that a solution $x_{\mathcal{D}}^\star \in \mathcal{X}_{\mathcal{D}}^\star$ of (2) is an approximate solution of the original problem (1).

**Theorem 1.** *Let Assumptions 2 and 3 hold, and let $x_{\mathcal{D}}^\star \in \mathcal{X}_{\mathcal{D}}^\star$ and $x^\star \in \mathcal{X}^\star$. Then*

$$f(x^\star) \leq f(x_{\mathcal{D}}^\star) \leq f(x^\star) + \frac{(L_{\mathcal{D}} - 1)L_f}{2} \|x^\star - v\|^2 - \frac{(\mu_{\mathcal{D}} - 1)\mu_f}{2} \|x_{\mathcal{D}}^\star - v\|^2 \, ;$$

$$f(x^\star) + \frac{(\mu_{\mathcal{D}} - 1)\mu_f}{2} \|x_{\mathcal{D}}^\star - v\|^2 \leq f_{\mathcal{D}}(x_{\mathcal{D}}^\star) \leq f(x^\star) + \frac{(L_{\mathcal{D}} - 1)L_f}{2} \|x^\star - v\|^2 \, .$$

Consider `Rand-K` as a sketch. If $K = (1 - \varepsilon)d$ for some $\varepsilon \in [0, 1)$, which corresponds to dropping roughly an $\varepsilon$ share of coordinates, then $L_{\mathcal{D}} - 1 = \mu_{\mathcal{D}} - 1 = d/K - 1 = \varepsilon/(1-\varepsilon)$. Theorem 1 then states that

$$f(x^\star) \leq f(x_{\mathcal{D}}^\star) \leq f(x^\star) + \frac{\varepsilon}{2(1 - \varepsilon)} \left( L_f \|x^\star - v\|^2 - \mu_f \|x_{\mathcal{D}}^\star - v\|^2 \right) \, ;$$

$$f(x^\star) + \frac{\varepsilon \mu_f}{2(1 - \varepsilon)} \|x_{\mathcal{D}}^\star - v\|^2 \leq f_{\mathcal{D}}(x_{\mathcal{D}}^\star) \leq f(x^\star) + \frac{\varepsilon L_f}{2(1 - \varepsilon)} \|x^\star - v\|^2 \, .$$

If $\varepsilon$ is small (a "light" sparsification), or if the pre-trained model $v$ is close to $x^\star$, then $x_{\mathcal{D}}^\star$ will have a small loss, comparable to the loss of the optimal model $x^\star$; $x_{\mathcal{D}}^\star$ will be close to the pre-trained model $v$; MAST loss will be small comparable to the loss of the optimal uncompressed model $x^\star$.

## 4 INDIVIDUAL NODE SETTING

In this section, we discuss the properties of the SGD-type algorithm applied to problem (2)

$$x^{t+1} = x^t - \gamma \nabla f_{\mathbf{S}^t}(x^t) = x^t - \gamma \left( \mathbf{S}^t \right)^\top \nabla f(y^t) \tag{11}$$

for $y^t = v + \mathbf{S}^t(x^t - v)$, where $\mathbf{S}^t$ is sampled from $\mathcal{D}$. One advantage of the proposed formulation (2) is that it naturally gives rise to the described method, which generalizes standard (Stochastic) Gradient Descent. As noted before, due to the properties of the gradient estimator (3), recursion (11) defines an Algorithm 1 (I) that sketches both the model and the gradient of $f$.

Let us introduce a notation frequently used in our convergence results. This quantity is determined by the spectral properties of the sketches being used.

$$L_{\mathbf{S}}^{\max} \stackrel{\text{def}}{=} \sup_{\mathbf{S}} L_{\mathbf{S}} = \sup_{\mathbf{S}} \lambda_{\max} \left( \mathbf{S}^\top \mathbf{S} \right) , \tag{12}$$

where $\sup_{\mathbf{S}} L_{\mathbf{S}}$ represents the tightest constant such that $L_{\mathbf{S}} \leq \sup_{\mathbf{S}} L_{\mathbf{S}}$ almost surely. For independent random sparsification sketches (6): $L_{\mathbf{S}}^{\max} = 1/p_{\min}^2$. If $\mathbf{S}$ is `Rand-K` (8) then $L_{\mathbf{S}}^{\max} = d^2/K^2$. Our convergence analysis relies on the following Lemma:

**Lemma 4.** *Assume that $f$ is $L_f$-smooth (2) and $\mathbf{S}$ satisfies Assumption 1. Then we have that $\forall x \in \mathbb{R}^d$*

$$\mathbb{E} \left[ \|\nabla f_{\mathbf{S}}(x)\|^2 \right] \leq 2 L_f L_{\mathbf{S}}^{\max} \left( f_{\mathcal{D}}(x) - f^{\inf} \right) ,$$

*where the expectation is taken with respect to $\mathbf{S}$.*

It generalizes a standard property of smooth functions often used in the non-convex analysis of SGD (Khaled & Richtárik, 2023). Now, we are ready to present our first convergence result.

**Theorem 2.** *Assume that $f$ is $L_f$-smooth (2), $\mu_f$-strongly convex (3), and $\mathbf{S}$ satisfies Assumption 1. Then, for stepsize $\gamma \leq 1/(L_f L_{\mathbf{S}}^{\max})$, the iterates of Algorithm 1 (I) satisfy*

$$\mathbb{E} \left[ \|x^T - x_{\mathcal{D}}^\star\|^2 \right] \leq (1 - \gamma \mu_{\mathcal{D}} \mu_f)^T \|x^0 - x_{\mathcal{D}}^\star\|^2 + \frac{2\gamma L_f L_{\mathbf{S}}^{\max}}{\mu_f \mu_{\mathcal{D}}} \left( f_{\mathcal{D}}^{\inf} - f^{\inf} \right) . \tag{13}$$

---

**Algorithm 1** Double Sketched (S)GD

---

1: **Parameters:** learning rate $\gamma > 0$; distribution $\mathcal{D}$; initial model and shift $x^0, v \in \mathbb{R}^d$.
2: **for** $t = 0, 1, 2 \ldots$ **do**
3:      Sample a sketch: $\mathbf{S}^t \sim \mathcal{D}$
4:      Form a gradient estimator:
$$g^t = \begin{cases} \nabla f_{\mathbf{S}^t}(x^t) & \triangleright \text{ exact (I)} \\ g_{\mathbf{S}^t}(x^t) & \triangleright \text{ (stochastic) inexact (II)} \end{cases}$$
5:      Perform a gradient-type step: $x^{t+1} = x^t - \gamma g^t$
6: **end for**

---

This theorem establishes a linear convergence rate with a constant stepsize up to a neighborhood of the MAST problem (2) solution. Our result is similar to the convergence of SGD (Gower et al., 2019) for standard formulation (1) with two differences, which are discussed below.

**1.** Both terms of the upper bound (13) depend not only on the smoothness and convexity parameters of the original function $f$, but also on the *spectral properties* of sketches $\mathbf{S}$. Thus, for independent Bernoulli sparsification sketches (6) with $p_i \equiv p$ (or Rand$-K$), the linear convergence term deteriorates and the neighborhood size is increased by $1/p^2$ ($d^2/K^2$ respectively). Therefore, we conclude that higher sparsity makes optimization harder.

**2.** Interestingly, the neighborhood size of (13) depends on the difference between the minima of $f_{\mathcal{D}}$ and $f$ in contrast to the variance of stochastic gradients at the optimum typical for SGD (Gower et al., 2019). Thus, the method may even linearly converge to the exact solution when $f_{\mathcal{D}}^{\text{inf}} = f^{\text{inf}}$, which we refer to as the *interpolation* condition. This condition may naturally hold when the original and sketched models are sufficiently overparametrized (allowing minimization of the loss to zero). Notable examples when similar phenomena have been observed in practice are training with Dropout (Srivastava et al., 2014) and the "lottery ticket hypothesis" (Frankle & Carbin, 2018).

Next, we provide results in the non-convex setting.

**Theorem 3.** *Assume that $f$ is $L_f$-smooth (2) and $\mathbf{S}$ satisfies Assumption 1. Then, for the stepsize $\gamma \leq 1/(L_f\sqrt{L_{\mathcal{D}}L_{\mathbf{S}}^{\max}T})$, the iterates of Algorithm 1 (I) satisfy*

$$\min_{0 \leq t < T} \mathbb{E}\left[\left\|\nabla f_{\mathcal{D}}(x^t)\right\|^2\right] \leq \frac{3\left(f_{\mathcal{D}}(x^0) - f_{\mathcal{D}}^{\text{inf}}\right)}{\gamma T} + \gamma L_f^2 L_{\mathcal{D}} L_{\mathbf{S}}^{\max}\left(f_{\mathcal{D}}^{\text{inf}} - f^{\text{inf}}\right).$$

This theorem shows an $\mathcal{O}(1/\sqrt{T})$ convergence rate for reaching a stationary point. Our result shares similarities with the theory of SGD (Khaled & Richtárik, 2023) for problem (1), with the main difference that the rate depends on the distribution on sketches as $\sqrt{L_{\mathcal{D}}L_{\mathbf{S}}^{\max}}$. Moreover, the second term depends on the difference between the minima of $f_{\mathcal{D}}$ and $f$, as in the strongly convex case. However, the first term depends on the gap between the initialization and the lower bound of the loss function, which is more common in non-convex settings (Khaled & Richtárik, 2023).

**Corollary 1** (Informal). *Fix $\varepsilon > 0$ and denote $\delta^t \overset{def}{=} \mathbb{E}\left[f_{\mathcal{D}}(x^t) - f_{\mathcal{D}}^{\text{inf}}\right], r^t \overset{def}{=} \mathbb{E}\left[\left\|\nabla f_{\mathcal{D}}(x^t)\right\|^2\right]$.*

*Then, for the stepsize $\gamma = \min\left\{\frac{1}{\sqrt{DT}}, \frac{\varepsilon^2}{2D\left(f_{\mathcal{D}}^{\text{inf}} - f^{\text{inf}}\right)}\right\}$, where $D \overset{def}{=} L_f\sqrt{L_{\mathcal{D}}L_{\mathbf{S}}^{\max}}$, Algorithm 1 (I)*

*needs $T \geq \frac{12\delta^0 D}{\varepsilon^4}\max\left\{3\delta^0, f_{\mathcal{D}}^{\text{inf}} - f^{\text{inf}}\right\}$ iterations to reach a stationary point $\min_{0 \leq t < T} r^t \leq \varepsilon^2$, which is order $\mathcal{O}(\varepsilon^{-4})$ optimal (Ghadimi & Lan, 2013; Drori & Shamir, 2020).*

In the Appendix, we also provide a general convex analysis.

## 4.1 (STOCHASTIC) INEXACT GRADIENT

Algorithm 1 (I) is probably the simplest approach for solving the MAST problem (2). Analyzing this algorithm isolates and highlights the unique properties of the proposed problem formulation. Algorithm 1 (I) requires exact (sketched) gradient computations at every iteration, which may not be feasible/efficient for modern ML applications. Hence, we consider the following generalization:

$$x^{t+1} = x^t - \gamma g_{\mathbf{S}^t}(x^t), \tag{14}$$

where $g_{\mathbf{S}^t}(x^t)$ is a gradient estimator satisfying

$$\mathbb{E}\left[g_{\mathbf{S}}(x)\right] = \nabla f_{\mathbf{S}}(x), \tag{15}$$

$$\mathbb{E}\left[\|g_{\mathbf{S}}(x)\|^2\right] \leq 2A\left(f_{\mathbf{S}}(x) - f_{\mathbf{S}}^{\inf}\right) + B\|\nabla f_{\mathbf{S}}(x)\|^2 + C, \tag{16}$$

for $\forall x \in \mathbb{R}^d$ and some constants $A, B, C \geq 0$.

The first condition (15) is an unbiasedness assumption standard for analyzing SGD-type methods. The second (so-called "ABC") inequality (16) is one of the most general assumptions covering bounded stochastic gradient variance, subsampling/minibatching of data, and gradient compression (Khaled & Richtárik, 2023; Demidovich et al., 2023). Note that the expectation in (15) and (16) is taken with respect only to the randomness of the stochastic gradient estimator and not the sketch $\mathbf{S}$.

Algorithm 1 (II) describes the resulting method in detail. We state the convergence result for it.

**Theorem 4.** *Assume that $f$ is $L_f$-smooth (2), $\mathbf{S}$ satisfies Assumption 1, and the gradient estimator $g(x)$ satisfies conditions 15, 16. Denote $D_{A,B} = A + BL_f L_{\mathbf{S}}^{\max}$ then, for stepsize $\gamma \leq 1/\sqrt{L_f L_{\mathcal{D}} D_{A,B} T}$, the iterates of Algorithm 1 (II) satisfy*

$$\min_{0 \leq t < T} \mathbb{E}\left[\|\nabla f_{\mathcal{D}}(x^t)\|^2\right] \leq \frac{3\left(f_{\mathcal{D}}(x^0) - f_{\mathcal{D}}^{\inf}\right)}{\gamma T} + \frac{\gamma L_f L_{\mathcal{D}}}{2}\left\{C + 2D_{A,B}\left(f_{\mathcal{D}}^{\inf} - f^{\inf}\right)\right\}.$$

Similarly to Theorem 3, this result establishes an $\mathcal{O}(1/\sqrt{T})$ convergence rate. However, the upper bound in (4) is affected by constants $A, B, C$ due to the inexactness of the gradient estimator. The case of $A = C = 0, B = 1$ sharply recovers our previous Theorem 3. When $B = 1, C = \sigma^2$, we obtain convergence of SGD with bounded (by $\sigma^2$) variance of stochastic gradients. Moreover, when the loss is represented as a finite sum

$$f_{\mathcal{D}}(x) = \frac{1}{n}\sum_{i=1}^{n}\mathbb{E}\left[f_i(v + \mathbf{S}(x - v))\right], \tag{17}$$

where each $f_i$ is $L_{f_i}$-smooth and lower-bounded by $f_i^{\inf}$, then $A = \max_i L_{f_i}, C = 2A\left(\frac{1}{n}\sum_{i=1}^{n}f^{\inf} - f_i^{\inf}\right)$ if losses $f_i$ are sampled uniformly at every iteration. Finally, our result (4) guarantees optimal $\mathcal{O}(\varepsilon^{-4})$ complexity in a similar manner to Corollary 1.

We direct the reader to the Appendix for the convex and strongly convex results.

## 4.2 DISCUSSION OF RELATED WORKS

**Compressed (sparse) model training.** To our knowledge, the first work that analyzed convergence of (full batch) Gradient Descent with compressed iterates (model updates) is the work of Khaled & Richtárik (2019). They considered general unbiased compressors and, in the strongly convex setting, showed linear convergence to the irreducible neighborhood, depending on the norm of the model at the optimum $\|x^\star\|^2$. In addition, their analysis requires the variance of the compressor ($d/K$ for Rand$-K$) to be lower than the inverse condition number of the problem $\mu_f/L_f$, which basically means that the compressor has to be close to the identity mapping in practical settings. These results were extended using a modified method to distributed training with compressed model updates (Chraibi et al., 2019; Shulgin & Richtárik, 2022). Lin et al. (2019) consider dynamic pruning with feedback inspired by the Error Feedback mechanism (Seide et al., 2014; Alistarh et al., 2018; Stich & Karimireddy, 2020). Their result is similar to (Khaled & Richtárik, 2019), as the method also converges only to the irreducible neighborhood, the size of which is proportional to the norm of model weights. However, in (Lin et al., 2019), the norm of stochastic gradients is required to be uniformly upper-bounded, narrowing the class of losses. The partial SGD method proposed in (Mohtashami et al., 2022) allows general perturbations of the model weights where the gradient (additionally sparsified) is computed. Unfortunately, their analysis (Wang et al., 2022) was recently shown to be vacuous (Szlendak et al., 2024).

**Dropout convergence analysis.** Despite the wide empirical success of Dropout, there is limited theoretical understanding of its behavior and success. A few recent works (Mianjy & Arora, 2020) suggest convergence analysis of this technique. However, these attempts typically focus on a certain

---

**Algorithm 2** Distributed Double Sketched GD

---

1: **Parameters:** learning rate $\gamma > 0$; sketch distributions $\mathcal{D}_1, \ldots, \mathcal{D}_M$; initial model and shift $x^0, v \in \mathbb{R}^d$
2: **for** $t = 0, 1, 2 \ldots$ **do**
3:     Sample sketches: $\mathbf{S}_i^t \sim \mathcal{D}_i$
4:     Compute $y_i^t = v + \mathbf{S}_i^t(x^t - v)$ for $i \in [M]$ and broadcast to corresponding nodes
5:     **for** $i = 1, \ldots, M$ in parallel **do**
6:         Compute local gradient: $\nabla f_i(y_i^t)$
7:         Send gradient $(\mathbf{S}_i^t)^\top \nabla f_i(y_i^t)$ to the server
8:     **end for**
9:     Aggregate messages and make a gradient-type step: $x^{t+1} = x^t - \frac{\gamma}{M} \sum_{i=1}^{M} (\mathbf{S}_i^t)^\top \nabla f_i(y_i^t)$
10: **end for**

---

class of models, such as shallow linear Neural Networks (NNs) (Senen-Cerda & Sanders, 2022) or deep NNs with ReLU activations (Senen-Cerda & Sanders, 2020). Moreover, Liao & Kyrillidis (2022) analyzed overparameterized single-hidden layer perceptron with a regression loss in the context of Dropout. In contrast, our approach is model-agnostic and requires only mild assumptions like the smoothness of the loss (2) (and convexity (3) for some of the results).

## 5 DISTRIBUTED SETTING

Consider $f$ being a finite sum over a number of participants, i.e., in the distributed setup:

$$\min_{x \in \mathbb{R}^d} \frac{1}{M} \sum_{i=1}^{M} f_{i, \mathcal{D}_i}(x), \tag{18}$$

where $f_{i, \mathcal{D}_i}(x) \overset{\text{def}}{=} \mathbb{E}\left[f_{i, \mathbf{S}_i}(x)\right] = \mathbb{E}\left[f_i(v + \mathbf{S}_i(x - v))\right]$. This setting is more general than problem (39) as every node $i$ has its own distribution of sketches $\mathbf{S}_i \sim \mathcal{D}_i$. Every machine performs local computations with a model of different size, which is crucial for scenarios with heterogeneous computing hardware. The shift model $v$ is shared across all $f_{i, \mathcal{D}_i}$. We solve (18) with the method

$$x^{t+1} = x^t - \frac{\gamma}{M} \sum_{i=1}^{M} \left(\mathbf{S}_i^t\right)^\top \nabla f_i(y_i^t), \tag{19}$$

where $y_i^t = v + \mathbf{S}_i^t(x^t - v)$. Algorithm 2 describes the proposed approach in detail. Local gradients can be computed for sketched (sparse) model weights, which decreases the computational load on the computing nodes. Moreover, the local gradients are sketched as well, which brings communication efficiency in the case of sparsifiers $\mathbf{S}_i$.

Recursion (19) is closely related to the distributed Independent Subnetwork Training (IST) framework (Yuan et al., 2022). At every iteration of IST, a large model $x^t$ is decomposed into submodels $\mathbf{S}_i^t x^t$ for independent computations (e.g., local training), which are then aggregated on the server to update the whole model. IST efficiently combines model and data parallelism, allowing the training of huge models that cannot fit onto a single device. IST was shown to be very successful for a range of DL applications (Dun et al., 2022; Wolfe et al., 2023). Shulgin & Richtárik (2024) analyzed the convergence of IST for a quadratic model decomposed with permutation sketches (Szlendak et al., 2022), which satisfy Assumption 1. They also showed that naively applying IST to standard distributed optimization problems ((18) for $\mathbf{S}_i \equiv \mathbf{I}$) results in a biased method and may not converge.

Resource-constrained Federated Learning (FL) (Kairouz et al., 2021; Konečný et al., 2016; McMahan et al., 2017) is another important practical scenario covered by Algorithm 2. In cross-device FL, local computations are typically performed by edge devices (e.g., mobile phones), which have limited memory, computational power, and energy (Caldas et al., 2018). Thus, this forces practitioners to rely on smaller (potentially less capable) models or use techniques such as Dropout in distributed setting (Alam et al., 2022; Bouacida et al., 2021; Charles et al., 2022; Chen et al., 2022; Diao et al., 2021; Horváth et al., 2021; Jiang et al., 2022; Qiu et al., 2022; Wen et al., 2022; Yang et al., 2022; Dun et al., 2023). Despite extensive experimental studies of this problem setting, the principled theoretical

understanding remains minimal. Our work can be considered the first rigorous analysis in the most general setting without restrictive assumptions.

**Theorem 5.** *Assume that each $f_i$ is $L_{f_i}$-smooth* (2) *and sketches* $\mathbf{S}_i$ *satisfy Assumption 1. Let $D_{\max} \overset{\text{def}}{=} \max_i \left( L_{f_i}^2 L_{\mathcal{D}_i} L_{\mathbf{S}_i}^{\max} \right)$. Then, for $\gamma \leq 1/\sqrt{D_{\max}T}$, the iterates of Algorithm 2 satisfy*

$$\min_{0 \leq t < T} \mathbb{E} \left[ \left\| \nabla f_{\mathcal{D}}(x^t) \right\|^2 \right] \leq \frac{3 \left( f_{\mathcal{D}}(x^0) - f_{\mathcal{D}}^{\inf} \right)}{\gamma T} + \gamma D_{\max} \left( f_{\mathcal{D}}^{\inf} - \frac{1}{M} \sum_{i=1}^{M} f_i^{\inf} \right).$$

This result resembles our previous Theorem 3 in the single-node setting. Namely, Algorithm 2 reaches a stationary point at rate $\mathcal{O}(1/\sqrt{T})$. However, due to the distributed setup, convergence depends on $D_{\max}$ expressed as the *maximum* product of local smoothness and constants related to the sketches' properties. Thus, clients with more aggressive sparsification may slow down the method, given the same local smoothness constant $L_{f_i}$. Yet, "easier" local problems (with smaller $L_{f_i}$) can allow the use of "harsher" sparsifiers (with larger $L_{\mathcal{D}_i} L_{\mathbf{S}_i}^{\max}$) without negatively affecting the convergence.

## 5.1 DISCUSSION OF RELATED WORKS

A notable distinction between our result and the theory of methods like Distributed Compressed Gradient Descent (Khirirat et al., 2018) lies in the second convergence term of Theorem 5. Instead of relying on the variance of local gradients at the optimum, given by $\delta^2 = \frac{1}{n} \sum_{i=1}^{n} \|\nabla f_i(x^\star)\|^2$, our result depends on the average difference between the lower bounds of the global and local losses: $f_{\mathcal{D}}^{\inf} - f_i^{\inf}$. This term measures heterogeneity within a distributed setting (Khaled et al., 2020). Furthermore, our findings may provide a better explanation of the empirical efficacy of distributed methods. Namely, $\delta^2$ is less likely to be equal to zero, unlike our term, which can be very small when models are over-parameterized, allowing local losses to be minimized to zero.

Yuan et al. (2022) analyzed convergence of Independent Subnetwork Training in their original work using the framework of Khaled & Richtárik (2019). Their analysis was performed in the single-node setting and required additional assumptions on the gradient estimator, which were recently shown to be problematic (Shulgin & Richtárik, 2024). In a federated setting, Zhou et al. (2022) suggested a method that combines model pruning with local compressed Gradient Descent steps. They provided non-convex convergence analysis relying on bounded stochastic gradient assumption, which results in "pathological" bounds (Khaled et al., 2020) for a heterogeneous distributed case.

# 6 EXPERIMENTS

To empirically validate our theoretical framework and its implications, we focus on carefully controlled settings that satisfy the assumptions of our work. Specifically, we consider an $\ell_2$-regularized logistic regression optimization problem with the *a5a* dataset from the LibSVM repository (Chang & Lin, 2011). See Appendix H for further details and Appendix H.2 for more results on other methods and sketches.

In Figure 1, we compare the test accuracy of sparsified solutions for the standard (ERM) problem (1) and introduced MAST formulation (2). Visualization is performed using the boxplot method from Seaborn (version 0.11.0) library (Waskom, 2021) with default parameters. For ERM, we find the exact (up to machine precision) optimum, which is subsequently used for the accuracy evaluation. For the MAST optimization problem, we run DSGD with exact sketched gradient $\nabla f_{\mathcal{D}}$ for every sparsity level.

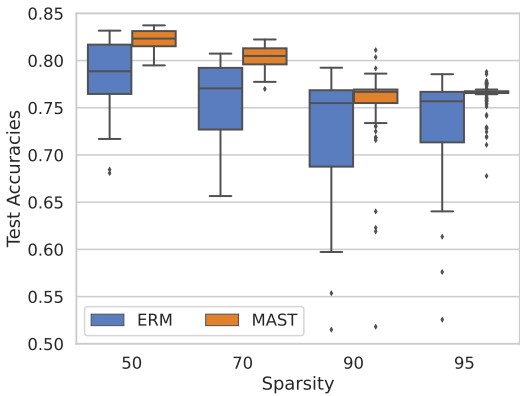

Figure 1: Test accuracies distributions of sparsified solutions for the ERM formulation (1) and MAST problem (2). "Sparsity" corresponds to the percentage of zeroed weights.

After the ERM and MAST models ($x^T$) are obtained, we apply partition sketches (47) to model weights and evaluate the test accuracy of the sparsified solutions ($\mathbf{S}x^T$).

Figure 1 reveals that models obtained using the MAST approach exhibit greater robustness to random pruning compared to their ERM counterparts given the same sparsity. Moreover, the ERM model suffers from greater accuracy variability, while the median test accuracies of the MAST models are markedly higher. Increasing the sparsity leads to the degradation of the performance of both approaches.

**Neural network results.** Next we present a subset of our distributed deep learning results (full details are provided in Appendix H.2.3). Our experimental setup closely follows that of Liao & Kyrillidis (2022), which is based on the ResNet-50 model (He et al., 2016). We study the Algorithm 2 with Bernoulli sketches (6) and $p_i \equiv p$ for the standard (ERM) loss (18) with $\mathbf{S}_i \equiv \mathbf{I}$.

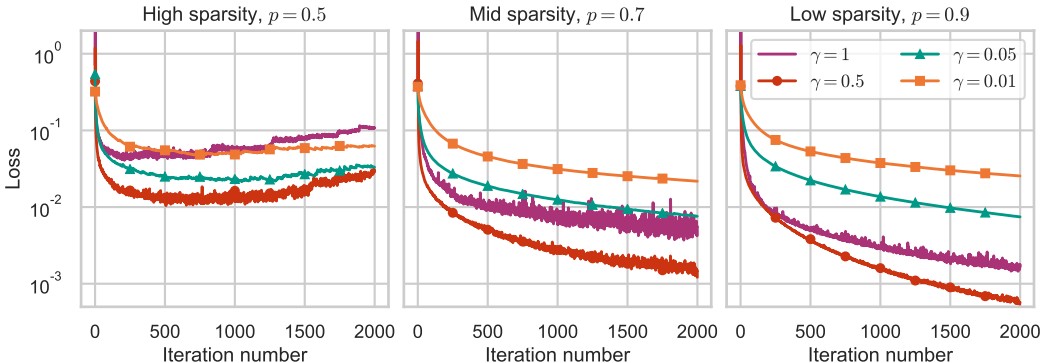

Figure 2: Performance of Algorithm 2 with Bernoulli sketches (6) on standard loss (18) (for $\mathbf{S}_i \equiv \mathbf{I}$)

Figure 2 illustrates the impact of sparsity level ($p$) and step size ($\gamma$) on the method's performance. Across all sparsity levels, we observe an optimal "sweet spot" ($\gamma = 0.5$) for the step size, beyond which increasing $\gamma$ results in slower convergence. Crucially, a nuanced interplay of $\gamma$ with sparsity level exists. Namely, at $\gamma = 1$, convergence slows down for $p = 0.9$, while for $p = 0.7$, performance degrades due to high variance, eventually being outperformed by a smaller step size.

Notably, high sparsity ($p = 0.5$) leads to a quick loss stagnation even with a small step size $\gamma = 0.01$ in contrast to $p \in \{0.7, 0.9\}$. Remarkably, the left plot in Figure 2 illustrates that an excessively large step size may even lead to divergence of the method. This can indicate that high sparsity significantly alters the minimized loss, confirming that Sparse/Dropout training indeed optimizes a formulation distinct from standard ERM. In general, larger step sizes and more aggressive sparsification (lower $p$) result in increased loss variance, aligning with our theoretical predictions from Sections 2 and 4.

One of the key practical insights derived from our theoretical analysis is that the step size $\gamma$ (learning rate) must be decreased for sparse optimization and training with Dropout. Our results demonstrate that this insight applies not only to convex models (Figure 5) but also to a broader range of neural networks.

## 7 CONCLUSIONS AND FUTURE WORK

This work introduced a novel theoretical framework for sketched model learning. We rigorously formalized a new optimization paradigm that captures practical scenarios like Dropout and Sparse training. Efficient optimization algorithms tailored to the proposed formulation were developed and analyzed in multiple settings. We expanded this methodology to distributed environments, encompassing areas such as IST and Federated Learning, underscoring its broad applicability.

In future research, it would be interesting to expand the class of linear matrix sketches to encompass other compression techniques, particularly those exhibiting conic variance property (contractive compressors). Such an extension might offer insights into (magnitude-based) pruning methods and quantized training. Nevertheless, a potential challenge to be considered is the non-differentiability of such compression techniques.

## ACKNOWLEDGEMENTS

We would like to thank anonymous reviewers for their helpful comments and suggestions to improve the manuscript.

The work was supported by funding from King Abdullah University of Science and Technology (KAUST): i) KAUST Baseline Research Scheme, ii) Center of Excellence for Generative AI, under award number 5940, iii) SDAIA-KAUST Center of Excellence in Artificial Intelligence and Data Science.

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

CONTENTS

# A  BASIC FACTS

For all $a, b \in \mathbb{R}^d$ and $\alpha > 0, p \in (0,1]$ the following relations hold:

$$2\langle a, b \rangle = \|a\|^2 + \|b\|^2 - \|a - b\|^2 \tag{20}$$

$$\|a + b\|^2 \leq (1 + \alpha)\|a\|^2 + \left(1 + \alpha^{-1}\right)\|b\|^2 \tag{21}$$

$$-\|a - b\|^2 \leq -\frac{1}{1 + \alpha}\|a\|^2 + \frac{1}{\alpha}\|b\|^2, \tag{22}$$

$$(1 - p)\left(1 + \frac{p}{2}\right) \leq 1 - \frac{p}{2}, \quad p \geq 0. \tag{23}$$

**Lemma 5** (Lemma 1 from (Mishchenko et al., 2020)). *Let $X_1, \ldots, X_n \in \mathbb{R}^d$ be fixed vectors, $\overline{X} \stackrel{def}{=} \frac{1}{n}\sum_{i=1}^n X_i$ be their average. Fix any $k \in \{1, \ldots, n\}$, let $X_{\pi_1}, \ldots X_{\pi_k}$ be sampled uniformly without replacement from $\{X_1, \ldots, X_n\}$ and $\overline{X}_\pi$ be their average. Then, the sample average and variance are given by*

$$\mathbb{E}\left[\overline{X}_\pi\right] = \overline{X}$$

$$\mathbb{E}\left[\left\|\overline{X}_\pi - \overline{X}\right\|^2\right] = \frac{n - k}{k(n - 1)}\frac{1}{n}\sum_{i=1}^n \left\|X_i - \overline{X}\right\|^2$$

**Lemma 6.** *(Lemma 5 from (Richtárik et al., 2021)). Let $a, b > 0$. If $0 \leq \gamma \leq \frac{1}{\sqrt{a}+b}$, then $a\gamma^2 + b\gamma \leq 1$. The bound is tight up to the factor of 2 since $\frac{1}{\sqrt{a}+b} \leq \min\left\{\frac{1}{\sqrt{a}}, \frac{1}{b}\right\} \leq \frac{2}{\sqrt{a}+b}$.*

**Proposition 1.** *Nonzero eigenvalues of $\mathbf{SS}^\top$ and $\mathbf{S}^\top\mathbf{S}$ coincide.*

*Proof.* Indeed, suppose $\lambda \neq 0$ is an eigenvalue of $\mathbf{S}^\top\mathbf{S}$ with an eigenvector $v \in \mathbb{R}^d$, then $\lambda$ is an eigenvalue of $\mathbf{SS}^\top$ with an eigenvector $\mathbf{S}v$. $\square$

**Lemma 7.** *Suppose that $f(x)$ is $L_f$-smooth, differentiable, and bounded from below by $f^{\inf}$. Then*

$$\|\nabla f(x)\|^2 \leq 2L_f\left(f(x) - f^{\inf}\right), \quad \forall x \in \mathbb{R}^d. \tag{24}$$

*Proof.* Let $x^+ = x - \frac{1}{L_f}\nabla f(x)$, then using the $L_f$-smoothness of $f$, we obtain

$$f(x^+) \leq f(x) + \langle \nabla f(x), x^+ - x \rangle + \frac{L_f}{2}\left\|x^+ - x\right\|^2.$$

Since $f^{\inf} \leq f(x^+)$ and the definition of $x^+$ we have,

$$f^{\inf} \leq f(x^+) \leq f(x) - \frac{1}{L_f}\|\nabla f(x)\|^2 + \frac{1}{2L_f}\|\nabla f(x)\|^2 = f(x) - \frac{1}{2L_f}\|\nabla f(x)\|^2.$$

Rearrangement of the terms provides the claimed result. $\square$

# B AUXILIARY FACTS ABOUT FUNCTIONS $f_\mathcal{D}(x)$ AND $f_\mathbf{S}(x)$

For a differentiable function $f : \mathbb{R}^d \to \mathbb{R}$ and $x,y \in \mathbb{R}^d$ Bregman divergence associated with $f$ is $D_f(x,y) \overset{\text{def}}{=} f(x) - f(y) - \langle \nabla f(y), x - y \rangle$.

**Lemma 8** (Bregman divergence). *If $f$ is continuously differentiable, then $D_{f_\mathcal{D}}(x,y) = \mathbb{E}\left[D_{f_\mathbf{S}}(x,y)\right].$*

*Proof.* Since $f$ is continuously differentiable, we can interchange integration and differentiation. The result follows from the linearity of expectation. $\square$

## B.1 CONSEQUENCES OF $L_f$-SMOOTHNESS

Recall the $L_f$-smoothness assumption.

**Assumption 2.** Function $f$ is differentiable and $L_f$-**smooth**, i.e., there is $L_f > 0$ such that $\forall x, h \in \mathbb{R}^d$

$$f(x + h) \leq f(x) + \langle \nabla f(x), h \rangle + \frac{L_f}{2} \|h\|^2 \,.$$

We also require $f$ to be lower bounded by $f^{\text{inf}} \in \mathbb{R}$.

**Lemma 9** (Consequences of $L_f$-smoothness). *If $f$ is $L_f$-smooth, then*

*(i) $f_\mathbf{S}$ is $L_{f_\mathbf{S}}$-smooth with $L_{f_\mathbf{S}} \leq L_\mathbf{S} L_f$. That is,*

$$f_\mathbf{S}(x + h) \leq f_\mathbf{S}(x) + \langle \nabla f_\mathbf{S}(x), h \rangle + \frac{L_\mathbf{S} L_f}{2} \|h\|^2 \,, \quad \forall x, h \in \mathbb{R}^d.$$

*(ii) $f_\mathcal{D}$ is $L_{f_\mathcal{D}}$-smooth with $L_{f_\mathcal{D}} \leq L_\mathcal{D} L_f$. That is,*

$$f_\mathcal{D}(x + h) \leq f_\mathcal{D}(x) + \langle \nabla f_\mathcal{D}(x), h \rangle + \frac{L_\mathcal{D} L_f}{2} \|h\|^2 \,, \quad \forall x, h \in \mathbb{R}^d.$$

*(iii)*

$$f_\mathcal{D}(x) \leq f(x) + \frac{(L_\mathcal{D} - 1)L_f}{2} \|x - v\|^2 \,, \quad \forall x \in \mathbb{R}^d. \tag{25}$$

*Proof.* (i) For any $x, h \in \mathbb{R}^d$, we have

$$
\begin{aligned}
f_\mathbf{S}(x + h) \quad &= \quad f(v + \mathbf{S}(x + h - v)) \\
&= \quad f(v + \mathbf{S}(x - v) + \mathbf{S}h) \\
&\overset{\text{Asn. 2}}{\leq} \quad f(v + \mathbf{S}(x - v)) + \langle \nabla f(v + \mathbf{S}(x - v)), \mathbf{S}h \rangle + \frac{L_f}{2} \|\mathbf{S}h\|^2 \\
&= \quad f(v + \mathbf{S}(x - v)) + \langle \mathbf{S}^\top \nabla f(v + \mathbf{S}(x - v)), h \rangle + \frac{L_f}{2} \langle \mathbf{S}^\top \mathbf{S}h, h \rangle \\
&\leq \quad f_\mathbf{S}(x) + \langle \nabla f_\mathbf{S}(x), h \rangle + \frac{L_\mathbf{S} L_f}{2} \|h\|^2 \,.
\end{aligned}
$$

(ii) For any $x, h \in \mathbb{R}^d$, we have

$$
\begin{aligned}
f_\mathcal{D}(x + h) \quad &= \quad \mathbb{E}\left[f(v + \mathbf{S}(x + h - v))\right] \\
&= \quad \mathbb{E}\left[f(v + \mathbf{S}(x - v) + \mathbf{S}h)\right] \\
&\overset{\text{Asn. 2}}{\leq} \quad \mathbb{E}\left[f(v + \mathbf{S}(x - v)) + \langle \nabla f(v + \mathbf{S}(x - v)), \mathbf{S}h \rangle + \frac{L_f}{2} \|\mathbf{S}h\|^2\right] \\
&= \quad \mathbb{E}\left[f(v + \mathbf{S}(x - v))\right] + \langle \mathbb{E}\left[\mathbf{S}^\top \nabla f(v + \mathbf{S}(x - v))\right], h \rangle \\
&\quad + \frac{L_f}{2} \mathbb{E}\left[\|\mathbf{S}h\|^2\right] \\
&= \quad f_\mathcal{D}(x) + \langle \nabla f_\mathcal{D}(x), h \rangle + \frac{L_f}{2} \langle \mathbb{E}\left[\mathbf{S}^\top \mathbf{S}\right] h, h \rangle \\
&\leq \quad f_\mathcal{D}(x) + \langle \nabla f_\mathcal{D}(x), h \rangle + \frac{L_\mathcal{D} L_f}{2} \|h\|^2 \,.
\end{aligned}
$$

(iii) For any $x \in \mathbb{R}^d$, we have

$$
\begin{aligned}
f_{\mathcal{D}}(x) \quad &= \quad \mathbb{E}\left[f(v + \mathbf{S}(x - v))\right] \\
&\overset{\text{Asn. } 2}{\leq} \quad \mathbb{E}\left[f(x) + \langle \nabla f(x), \mathbf{S}(x - v) - (x - v)\rangle + \frac{L_f}{2}\|\mathbf{S}(x - v) - (x - v)\|^2\right] \\
&= \quad f(x) + \langle \nabla f(x), \mathbb{E}\left[\mathbf{S}(x - v) - (x - v)\right]\rangle \\
&\quad + \quad \frac{L_f}{2}\mathbb{E}\left[\|\mathbf{S}(x - v) - (x - v)\|^2\right] \\
&\overset{(4)}{=} \quad f(x) + \langle \nabla f(x), 0\rangle + \frac{L_f}{2}(x - v)^\top \left(\mathbb{E}\left[\mathbf{S}^\top \mathbf{S}\right] - \mathbf{I}\right)(x - v) \\
&= \quad f(x) + \frac{(L_{\mathcal{D}} - 1)L_f}{2}\|x - v\|^2.
\end{aligned}
$$

$\square$

## B.2 Consequences of convexity

We do not assume differentiability of $f$ here. Recall that function $f$ is convex if, for all $x, y \in \mathbb{R}^d$ and $\alpha \in [0,1]$, we have that $f(\alpha x + (1 - \alpha)y) \leq \alpha f(x) + (1 - \alpha)f(y)$.

**Lemma 10.** *If $f$ is convex, then $f_{\mathcal{D}}$ is convex and $f_{\mathcal{D}}(x) \geq f(x)$ for all $x \in \mathbb{R}^d$.*

*Proof.*   (i) Let $x, y \in \mathbb{R}^d$ and $\alpha \in [0,1]$. Then

$$
\begin{aligned}
f_{\mathcal{D}}(\alpha x + (1 - \alpha)y) \quad &\overset{(2)}{=} \quad \mathbb{E}\left[f(v + \mathbf{S}(\alpha x + (1 - \alpha)y - v))\right] \\
&= \quad \mathbb{E}\left[f(\alpha\left(v + \mathbf{S}(x - v)\right) + (1 - \alpha)\left(v + \mathbf{S}(y - v)\right))\right] \\
&\leq \quad \mathbb{E}\left[\alpha f(v + \mathbf{S}(x - v)) + (1 - \alpha)f(v + \mathbf{S}(y - v))\right] \\
&= \quad \alpha\mathbb{E}\left[f(v + \mathbf{S}(x - v))\right] + (1 - \alpha)\mathbb{E}\left[f(v + \mathbf{S}(y - v))\right] \\
&\overset{(2)}{=} \quad \alpha f_{\mathcal{D}}(x) + (1 - \alpha)f_{\mathcal{D}}(y).
\end{aligned}
$$

Alternative proof: Each $f_{\mathbf{S}}$ is obviously convex, and expectation of convex functions is a convex function.

(ii) Fix $x \in \mathbb{R}^d$ and let $g \in \partial f(x)$ be a subgradient of $f$ at $x$. Then

$$
\begin{aligned}
f_{\mathcal{D}}(x) \quad &\overset{(2)}{=} \quad \mathbb{E}\left[f(v + \mathbf{S}(x - v))\right] \\
&\geq \quad \mathbb{E}\left[f(x) + \langle g, \mathbf{S}(x - v) - (x - v)\rangle\right] \\
&= \quad f(x) + \langle g, \mathbb{E}\left[\mathbf{S}(x - v) - (x - v)\right]\rangle \\
&\overset{(4)}{=} \quad f(x) + \langle g, 0\rangle \\
&= \quad f(x).
\end{aligned}
$$

Alternative proof: Using Jensen's inequality, $f(v + \mathbf{S}(x - v)) = \mathbb{E}\left[v + \mathbf{S}(x - v)\right] \geq f(\mathbb{E}\left[v + \mathbf{S}(x - v)\right]) = f(x)$.

$\square$

## B.3 Consequences of $\mu_f$-convexity

Recall the $\mu_f$-strong convexity (or, for simplicity, $\mu_f$-convexity) assumption.

**Assumption 3.** Function $f$ is differentiable and $\mu_f$-**strongly convex**, i.e., there is $\mu_f > 0$ such that $\forall x, h \in \mathbb{R}^d$

$$
f(x + h) \geq f(x) + \langle \nabla f(x), h\rangle + \frac{\mu_f}{2}\|h\|^2.
$$

**Lemma 11** (Consequences of $\mu_f$-convexity)**.** *If $f$ is $\mu_f$-convex, then*

(i) *$f_{\mathbf{S}}$ is $\mu_{f_{\mathbf{S}}}$-convex with $\mu_{f_{\mathbf{S}}} \geq \mu_{\mathbf{S}}\mu_f$. That is,*

$$
f_{\mathbf{S}}(x + h) \geq f_{\mathbf{S}}(x) + \langle \nabla f_{\mathbf{S}}(x), h\rangle + \frac{\mu_{\mathbf{S}}\mu_f}{2}\|h\|^2, \quad \forall x, h \in \mathbb{R}^d.
$$

*(ii)* $f_{\mathcal{D}}$ *is* $\mu_{f_{\mathcal{D}}}$*-convex with* $\mu_{f_{\mathcal{D}}} \geq \mu_{\mathcal{D}}\mu_f$*. That is,*

$$f_{\mathcal{D}}(x + h) \geq f_{\mathcal{D}}(x) + \langle \nabla f_{\mathcal{D}}(x), h \rangle + \frac{\mu_{\mathcal{D}}\mu_f}{2} \|h\|^2, \quad \forall x, h \in \mathbb{R}^d.$$

*(iii)*

$$f_{\mathcal{D}}(x) \geq f(x) + \frac{(\mu_{\mathcal{D}} - 1)\mu_f}{2} \|x - v\|^2, \quad \forall x \in \mathbb{R}^d. \tag{26}$$

*Proof.*    (i) For any $x, h \in \mathbb{R}^d$, we have

$$
\begin{aligned}
f_{\mathbf{S}}(x + h) &= f(v + \mathbf{S}(x + h - v)) \\
&= f(v + \mathbf{S}(x - v) + \mathbf{S}h) \\
&\overset{\text{Asn. 3}}{\geq} f(v + \mathbf{S}(x - v)) + \langle \nabla f(v + \mathbf{S}(x - v)), \mathbf{S}h \rangle + \frac{\mu_f}{2} \|\mathbf{S}h\|^2 \\
&= f(v + \mathbf{S}(x - v)) + \langle \mathbf{S}^\top \nabla f(v + \mathbf{S}(x - v)), h \rangle + \frac{\mu_f}{2} \langle \mathbf{S}^\top \mathbf{S}h, h \rangle \\
&\geq f_{\mathbf{S}}(x) + \langle \nabla f_{\mathbf{S}}(x), h \rangle + \frac{\mu_{\mathbf{S}}\mu_f}{2} \|h\|^2.
\end{aligned}
$$

(ii) For any $x, h \in \mathbb{R}^d$, we have

$$
\begin{aligned}
f_{\mathcal{D}}(x + h) &= \mathbb{E}\left[ f(v + \mathbf{S}(x + h - v)) \right] \\
&= \mathbb{E}\left[ f(v + \mathbf{S}(x - v) + \mathbf{S}h) \right] \\
&\overset{\text{Asn. 3}}{\geq} \mathbb{E}\left[ f(v + \mathbf{S}(x - v)) + \langle \nabla f(v + \mathbf{S}(x - v)), \mathbf{S}h \rangle + \frac{\mu_f}{2} \|\mathbf{S}h\|^2 \right] \\
&= \mathbb{E}\left[ f(v + \mathbf{S}(x - v)) \right] + \left\langle \mathbb{E}\left[ \mathbf{S}^\top \nabla f(v + \mathbf{S}(x - v)) \right], h \right\rangle \\
&\quad + \frac{\mu_f}{2} \mathbb{E}\left[ \|\mathbf{S}h\|^2 \right] \\
&= f_{\mathcal{D}}(x) + \langle \nabla f_{\mathcal{D}}(x), h \rangle + \frac{\mu_f}{2} \left\langle \mathbb{E}\left[ \mathbf{S}^\top \mathbf{S} \right] h, h \right\rangle \\
&\geq f_{\mathcal{D}}(x) + \langle \nabla f_{\mathcal{D}}(x), h \rangle + \frac{\mu_{\mathcal{D}}\mu_f}{2} \|h\|^2.
\end{aligned}
$$

(iii) For any $x \in \mathbb{R}^d$, we have

$$
\begin{aligned}
f_{\mathcal{D}}(x) &= \mathbb{E}\left[ f(v + \mathbf{S}(x - v)) \right] \\
&\overset{\text{Asn. 3}}{\geq} \mathbb{E}\left[ f(x) + \langle \nabla f(x), \mathbf{S}(x - v) - (x - v) \rangle + \frac{\mu_f}{2} \|\mathbf{S}(x - v) - (x - v)\|^2 \right] \\
&= f(x) + \langle \nabla f(x), \mathbb{E}\left[ \mathbf{S}(x - v) - (x - v) \right] \rangle \\
&\quad + \frac{\mu_f}{2} \mathbb{E}\left[ \|\mathbf{S}(x - v) - (x - v)\|^2 \right] \\
&\overset{(4)}{=} f(x) + \langle \nabla f(x), 0 \rangle + \frac{\mu_f}{2}(x - v)^\top \left( \mathbb{E}\left[ \mathbf{S}^\top \mathbf{S} \right] - \mathbf{I} \right)(x - v) \\
&= f(x) + \frac{(\mu_{\mathcal{D}} - 1)\mu_f}{2} \|x - v\|^2.
\end{aligned}
$$

$\square$

## C    RELATION BETWEEN MINIMA OF $f$ AND $f_\mathcal{D}$.

**Theorem 6.** *Let Assumptions 2 and 3 hold, and let $x_\mathcal{D}^\star \in \tilde{\mathcal{X}}$ and $x^\star \in \mathcal{X}^\star$. Then*

$$f(x^\star) \leq f(x_\mathcal{D}^\star) \leq f(x^\star) + \frac{(L_\mathcal{D} - 1)L_f}{2} \|x^\star - v\|^2 - \frac{(\mu_\mathcal{D} - 1)\mu_f}{2} \|x_\mathcal{D}^\star - v\|^2,$$

$$f(x^\star) + \frac{(\mu_\mathcal{D} - 1)\mu_f}{2} \|x_\mathcal{D}^\star - v\|^2 \leq f_\mathcal{D}(x_\mathcal{D}^\star) \leq f(x^\star) + \frac{(L_\mathcal{D} - 1)L_f}{2} \|x^\star - v\|^2.$$

*Proof.* To obtain the result, combine inequalities (25) and (26):

$$f(x_\mathcal{D}^\star) + \frac{(\mu_\mathcal{D} - 1)\mu_f}{2} \|x_\mathcal{D}^\star - v\|^2 \overset{(26)}{\leq} f_\mathcal{D}(x_\mathcal{D}^\star) \leq f_\mathcal{D}(x^\star) \overset{(25)}{\leq} f(x^\star) + \frac{(L_\mathcal{D} - 1)L_f}{2} \|x^\star - v\|^2.$$

$\square$

**Theorem 7.** *Let Assumption 2 hold, and let $x_\mathcal{D}^\star \in \mathcal{X}_\mathcal{D}^\star$ and $x^\star \in \mathcal{X}^\star$. Then*

$$f(x^\star) \leq f_\mathcal{D}(x_\mathcal{D}^\star) \leq f(x^\star) + \frac{(L_\mathcal{D} - 1)L_f}{2} \|x^\star - v\|^2.$$

*Proof.* To obtain the result, use inequality (25). Also, note that, since for every $\mathbf{S} \sim \mathcal{D}$, we have $f_\mathbf{S}(x) = f(v + \mathbf{S}(x - v)) \geq f(x^\star)$, for all $x \in \mathbb{R}^d$, we can conclude that $f_\mathcal{D}(x_\mathcal{D}^\star) = \mathbb{E}[f_\mathbf{S}(x_\mathcal{D}^\star)] \geq \mathbb{E}[f(x^\star)] = f(x^\star)$. $\square$

### C.1    CONSEQUENCES OF LIPSCHITZ CONTINUITY OF THE GRADIENT

The gradient of $f(x)$ is $L_f$-Lipschitz if, for all $x, y \in \mathbb{R}^d$ we have that $\|\nabla f(x) - \nabla f(y)\| \leq L_f \|x - y\|$.

**Lemma 12.** *If $\nabla f$ is $L_f$-Lipschitz, then $\nabla f_\mathcal{D}$ is $L_{f_\mathcal{D}}$-Lipschitz with*

$$L_{f_\mathcal{D}} \leq L_f \mathbb{E}\left[\|\mathbf{S}^\top\| \|\mathbf{S}\|\right].$$

*Proof.* We have that

$$
\begin{aligned}
\|\nabla f_\mathcal{D}(x) - \nabla f_\mathcal{D}(y)\| &= \|\nabla \mathbb{E}[f(v + \mathbf{S}(x - v))] - \nabla \mathbb{E}[f(v + \mathbf{S}(y - v))]\| \\
&= \|\mathbb{E}[\mathbf{S}^\top \nabla f(v + \mathbf{S}(x - v))] - \mathbb{E}[\mathbf{S}^\top \nabla f(v + \mathbf{S}(y - v))]\| \\
&= \|\mathbb{E}[\mathbf{S}^\top \nabla f(v + \mathbf{S}(x - v)) - \mathbf{S}^\top \nabla f(v + \mathbf{S}(y - v))]\| \\
&\leq \mathbb{E}[\|\mathbf{S}^\top \nabla f(v + \mathbf{S}(x - v)) - \mathbf{S}^\top \nabla f(v + \mathbf{S}(y - v))\|] \\
&\leq \mathbb{E}[\|\mathbf{S}^\top\| \|\nabla f(v + \mathbf{S}(x - v)) - \nabla f(v + \mathbf{S}(y - v))\|] \\
&\leq \mathbb{E}[\|\mathbf{S}^\top\| L_f \|\mathbf{S}x - \mathbf{S}y\|] \\
&\leq L_f \mathbb{E}[\|\mathbf{S}^\top\| \|\mathbf{S}\|] \|x - y\|.
\end{aligned}
$$

$\square$

# D DOUBLE SKETCHED GD

Recall that $L_{\mathbf{S}}^{\max} = \sup_{\mathbf{S}} \left\{ \lambda_{\max} \left( \mathbf{S}^\top \mathbf{S} \right) \right\} = \sup_{\mathbf{S}} \left\{ \lambda_{\max} \left( \mathbf{S}\mathbf{S}^\top \right) \right\}$ (we used Proposition 1).

## D.1 NONCONVEX ANALYSIS: PROOF OF THEOREM 3

The following lemma is a restated Lemma 4 from the main part of the paper.

**Lemma 13.** *For all $x \in \mathbb{R}^d$, we have that*

$$\mathbb{E} \left[ \|\nabla f_{\mathbf{S}}(x)\|^2 \right] \leq 2 L_f L_{\mathbf{S}}^{\max} \left( f_{\mathcal{D}}(x) - f^{\inf} \right).$$

*where the expectation is taken with respect to $\mathbf{S}$.*

*Proof.* Due to $L_f$-smoothness of $f$, we have that

$$
\begin{aligned}
\mathbb{E} \left[ \|\nabla f_{\mathbf{S}}(x)\|^2 \right] &= \mathbb{E} \left[ \left\| \mathbf{S}^\top \nabla f(y) |_{y = v + \mathbf{S}(x-v)} \right\|^2 \right] \\
&= \mathbb{E} \left[ \left\langle \mathbf{S}^\top \nabla f(y), \mathbf{S}^\top \nabla f(y) \right\rangle |_{y = v + \mathbf{S}(x-v)} \right] \\
&= \mathbb{E} \left[ \left\langle \mathbf{S}\mathbf{S}^\top \nabla f(y), \nabla f(y) \right\rangle |_{y = v + \mathbf{S}(x-v)} \right] \\
&\leq \mathbb{E} \left[ \lambda_{\max} \left( \mathbf{S}\mathbf{S}^\top \right) \left\| \nabla f(y) |_{y = v + \mathbf{S}(x-v)} \right\|^2 \right] \\
&\leq L_{\mathbf{S}}^{\max} \mathbb{E} \left[ \left\| \nabla f(y) |_{y = v + \mathbf{S}(x-v)} \right\|^2 \right] \\
&\overset{(24)}{\leq} 2 L_f L_{\mathbf{S}}^{\max} \mathbb{E} \left[ f \left( v + \mathbf{S}(x - v) \right) - f^{\inf} \right] \\
&= 2 L_f L_{\mathbf{S}}^{\max} \left( f_{\mathcal{D}}(x) - f^{\inf} \right).
\end{aligned}
$$

$\square$

All convergence results in the nonconvex scenarios rely on the following key lemma:

**Lemma 14.** *The iterates $\{x^t\}_{t \geq 0}$ of SGD satisfy*

$$\gamma r^t \leq \left( 1 + \gamma^2 M_1 \right) \delta^t - \delta^{t+1} + \gamma^2 M_2, \tag{27}$$

*where $M_1$ and $M_2$ are non-negative constants, $\delta^t \overset{def}{=} \mathbb{E} \left[ f_{\mathcal{D}}(x^t) - f_{\mathcal{D}}^{\inf} \right]$ and $r^t \overset{def}{=} \mathbb{E} \left[ \|\nabla f_{\mathcal{D}}(x^t)\|^2 \right]$. Fix $w_{-1} > 0$ and, for all $t \geq 0$, define $w_t = \frac{w_{t-1}}{1 + \gamma^2 M_1}$. Then, for any $T \geq 1$, the iterates $\{x^t\}_{t \geq 0}$ satisfy*

$$\sum_{t=0}^{T-1} w_t r^t \leq \frac{w_1}{\gamma} \delta^0 - \frac{w_{T-1}}{\gamma} \delta^T + \gamma M_2 \sum_{t=0}^{T-1} w_t.$$

*Proof.* Multiplying both sides of (27) by $\frac{w_t}{\gamma}$, we obtain

$$w_t r^t \leq \frac{w_{t-1}}{\gamma} \delta^t - \frac{w_t}{\gamma} \delta^{t+1} + \gamma w_t M_2.$$

For every $0 \leq t \leq T - 1$, sum these inequalities. We arrive at

$$\sum_{t=0}^{T-1} w_t r^t \leq \frac{w_{-1}}{\gamma} \delta^0 - \frac{w_{T-1}}{\gamma} \delta^T + \gamma M_2 \sum_{t=0}^{T-1} w_t.$$

$\square$

Recall that $D \overset{def}{=} L_f \sqrt{L_{\mathcal{D}} L_{\mathbf{S}}^{\max}}$.

**Theorem 8.** *Let Assumptions 1 and 2 hold. For every $t \geq 0$, put $\delta^t \overset{def}{=} \mathbb{E} \left[ f_{\mathcal{D}}(x^t) - f_{\mathcal{D}}^{\inf} \right]$ and $r^t \overset{def}{=} \mathbb{E} \left[ \|\nabla f_{\mathcal{D}}(x^t)\|^2 \right]$. Then, for any $T \geq 1$, the iterates $\{x^t\}_{t=0}^{T-1}$ of Algorithm 1 satisfy*

$$\min_{0 \leq t < T} r^t \leq \frac{\left( 1 + D^2 \gamma^2 \right)^T}{\gamma T} \delta^0 + D^2 \gamma \left( f_{\mathcal{D}}^{\inf} - f^{\inf} \right). \tag{28}$$

*Proof.* Due to $L_f$-smoothness of $f$, we have that

$$f(s + \mathbf{S}^{t+1}(x^{t+1} - s)) \leq f(s + \mathbf{S}^{t+1}(x^t - s))$$
$$+ \left\langle \nabla f(y)|_{y=s+\mathbf{S}^{t+1}(x^t-s)}, \mathbf{S}^{t+1}\left(x^{t+1} - s\right) - \mathbf{S}^{t+1}\left(x^t - s\right) \right\rangle$$
$$+ \frac{L_f}{2} \left\| \mathbf{S}^{t+1}\left(x^{t+1} - s\right) - \mathbf{S}^{t+1}\left(x^t - s\right) \right\|^2$$
$$= f(s + \mathbf{S}^{t+1}(x^t - s)) - \gamma \left\langle \nabla f(y)|_{y=s+\mathbf{S}^{t+1}(x^t-s)}, \mathbf{S}^{t+1}\nabla f_{\mathbf{S}^t}(x^t) \right\rangle$$
$$+ \frac{L_f \gamma^2}{2} \left\| \mathbf{S}^{t+1}\nabla f_{\mathbf{S}^t}(x^t) \right\|^2$$
$$\leq f(s + \mathbf{S}^{t+1}(x^t - s)) - \gamma \left\langle \nabla f_{\mathbf{S}^{t+1}}\left(x^t\right), \nabla f_{\mathbf{S}^t}(x^t) \right\rangle$$
$$+ \frac{L_f \gamma^2}{2} \left\langle \left(\mathbf{S}^{t+1}\right)^\top \mathbf{S}^{t+1}\nabla f_{\mathbf{S}^t}(x^t), \nabla f_{\mathbf{S}^t}(x^t) \right\rangle.$$

Taking the expectation with respect to $\mathbf{S}^{t+1}$ yields

$$f_{\mathcal{D}}\left(x^{t+1}\right) \leq f_{\mathcal{D}}\left(x^t\right) - \gamma \left\langle \nabla f_{\mathcal{D}}\left(x^t\right), \nabla f_{\mathbf{S}^t}(x^t) \right\rangle$$
$$+ \frac{L_f \gamma^2}{2} \left\langle \mathbb{E}\left[ \left(\mathbf{S}^{t+1}\right)^\top \mathbf{S}^{t+1} \right] \nabla f_{\mathbf{S}^t}(x^t), \nabla f_{\mathbf{S}^t}(x^t) \right\rangle$$
$$\leq f_{\mathcal{D}}\left(x^t\right) - \gamma \left\langle \nabla f_{\mathcal{D}}\left(x^t\right), \nabla f_{\mathbf{S}^t}(x^t) \right\rangle + \frac{L_f L_{\mathcal{D}} \gamma^2}{2} \left\| \nabla f_{\mathbf{S}^t}(x^t) \right\|^2.$$

Conditioned on $x^t$, take expectation with respect to $\mathbf{S}^t$ :

$$\mathbb{E}\left[ f_{\mathcal{D}}\left(x^{t+1}\right) | x^t \right] \leq f_{\mathcal{D}}\left(x^t\right) - \gamma \left\| \nabla f_{\mathcal{D}}\left(x^t\right) \right\|^2 + \frac{L_f L_{\mathcal{D}} \gamma^2}{2} \mathbb{E}\left[ \left\| \nabla f_{\mathbf{S}^t}(x^t) \right\|^2 \right].$$

From Lemma 4, we obtain that

$$\mathbb{E}\left[ f_{\mathcal{D}}\left(x^{t+1}\right) | x^t \right] \leq f_{\mathcal{D}}\left(x^t\right) - \gamma \left\| \nabla f_{\mathcal{D}}\left(x^t\right) \right\|^2$$
$$+ \frac{L_f L_{\mathcal{D}} \gamma^2}{2} \left( 2 L_f L_{\mathbf{S}}^{\max}\left(f_{\mathcal{D}}(x) - f_{\mathcal{D}}^{\inf}\right) + 2 L_f L_{\mathbf{S}}^{\max}\left(f_{\mathcal{D}}^{\inf} - f^{\inf}\right) \right).$$

Subtract $f_{\mathcal{D}}^{\inf}$ from both sides, take expectations on both sides, and use the tower property:

$$\mathbb{E}\left[ f_{\mathcal{D}}\left(x^{t+1}\right) - f_{\mathcal{D}}^{\inf} \right] \leq \left(1 + D^2 \gamma^2\right) \mathbb{E}\left[ f_{\mathcal{D}}\left(x^t\right) - f_{\mathcal{D}}^{\inf} \right] - \gamma \mathbb{E}\left[ \left\| \nabla f_{\mathcal{D}}\left(x^t\right) \right\|^2 \right]$$
$$+ D^2 \gamma^2 \left(f_{\mathcal{D}}^{\inf} - f^{\inf}\right).$$

We obtain that

$$\gamma r^t \leq \left(1 + D^2 \gamma^2\right) \delta^t - \delta^{t+1} + D\gamma^2 \left(f_{\mathcal{D}}^{\inf} - f^{\inf}\right).$$

Notice that the iterates $\{x^t\}_{t\geq 0}$ of Algorithm 1 satisfy condition (27) of Lemma 14 with $M_1 = D^2$, and $M_2 = D^2 \left(f_{\mathcal{D}}^{\inf} - f^{\inf}\right)$. Therefore, we can conclude that, for any $T \geq 1$, the iterates $\{x^t\}_{t=0}^{T-1}$ of Algorithm 1 satisfy

$$\sum_{t=0}^{T-1} w_t r^t \leq \frac{w_{-1}}{\gamma} \delta^0 - \frac{w_{T-1}}{\gamma} \delta^T + D^2 \gamma \left(f_{\mathcal{D}}^{\inf} - f^{\inf}\right) \sum_{t=0}^{T-1} w_t.$$

Divide both sides by $\sum_{t=0}^{T-1} w_t$. From $\sum_{t=0}^{T-1} w_t \geq T w_{T-1} = \frac{T w_{-1}}{1+D^2\gamma^2}$, we can conclude that

$$\min_{0 \leq t < T} r^t \leq \frac{\left(1 + D^2 \gamma^2\right)^T}{\gamma T} \delta^0 + D^2 \gamma \left(f_{\mathcal{D}}^{\inf} - f^{\inf}\right).$$

$\square$

**Corollary 2.** *Fix $\varepsilon > 0$. Choose the stepsize $\gamma > 0$ as*

$$\gamma = \min\left\{\frac{1}{D\sqrt{T}}, \frac{\varepsilon^2}{2D^2\left(f_{\mathcal{D}}^{\mathrm{inf}} - f^{\mathrm{inf}}\right)}\right\}.$$

*Then, provided that*

$$T \geq \frac{12\delta^0 D^2}{\varepsilon^4}\max\left\{3\delta^0, f_{\mathcal{D}}^{\mathrm{inf}} - f^{\mathrm{inf}}\right\},$$

*we have*

$$\min_{0 \leq t < T} \mathbb{E}\left[\left\|\nabla f_{\mathcal{D}}(x^t)\right\|^2\right] \leq \varepsilon^2.$$

*Proof.* Since $\gamma \leq \frac{\varepsilon^2}{2D^2\left(f_{\mathcal{D}}^{\mathrm{inf}} - f^{\mathrm{inf}}\right)}$, we obtain

$$D^2\gamma\left(f_{\mathcal{D}}^{\mathrm{inf}} - f^{\mathrm{inf}}\right) \leq \frac{\varepsilon^2}{2}.$$

Since $\gamma \leq \frac{1}{D\sqrt{T}}$,

$$\left(1 + D^2\gamma^2\right)^T \leq \exp\left(TD^2\gamma^2\right) \leq \exp(1) \leq 3.$$

If $\gamma = \frac{1}{D\sqrt{T}}$, then, since

$$T \geq \frac{36\left(\delta^0\right)^2 D^2}{\varepsilon^4},$$

we have $\frac{3\delta^0}{\gamma T} \leq \frac{\varepsilon^2}{2}$. Further, if $\gamma = \frac{\varepsilon^2}{2D^2\left(f_{\mathcal{D}}^{\mathrm{inf}} - f^{\mathrm{inf}}\right)}$, then, since

$$T \geq \frac{12\delta^0 D^2\left(f_{\mathcal{D}}^{\mathrm{inf}} - f^{\mathrm{inf}}\right)}{\varepsilon^4},$$

we have $\frac{3\delta^0}{\gamma T} \leq \frac{\varepsilon^2}{2}$. Combining it with (28), we arrive at $\min_{0 \leq t < T} \mathbb{E}\left[\left\|\nabla f_{\mathcal{D}}(x^t)\right\|^2\right] \leq \varepsilon^2$. □

### D.2 STRONGLY CONVEX ANALYSIS: PROOF OF THEOREM 2

**Theorem 9.** *Let Assumptions 1, 2, and 3 hold. Let $r^t \stackrel{\mathrm{def}}{=} x^t - x_{\mathcal{D}}^\star$, $t \geq 0$. Choose a stepsize $0 < \gamma \leq \frac{1}{L_f L_{\mathbf{S}}^{\mathrm{max}}}$. Then the iterates $\{x^t\}_{t \geq 0}$ of Algorithm 1 satisfy*

$$\mathbb{E}\left[\left\|r^{t+1}\right\|^2\right] \leq (1 - \gamma\mu_{\mathcal{D}}\mu_f)\mathbb{E}\left[\left\|r^t\right\|^2\right] + 2\gamma^2 L_f L_{\mathbf{S}}^{\mathrm{max}}\left(f_{\mathcal{D}}^{\mathrm{inf}} - f^{\mathrm{inf}}\right). \tag{29}$$

*Proof.* Let $r^t \stackrel{\mathrm{def}}{=} x^t - x_{\mathcal{D}}^\star$. We get

$$\left\|r^{t+1}\right\|^2 = \left\|\left(x^t - \gamma\nabla f_{\mathbf{S}^t}(x^t)\right) - x_{\mathcal{D}}^\star\right\|^2 = \left\|x^t - x_{\mathcal{D}}^\star - \gamma\nabla f_{\mathbf{S}^t}(x^t)\right\|^2$$
$$= \left\|r^t\right\|^2 - 2\gamma\langle r^t, \nabla f_{\mathbf{S}^t}(x^t)\rangle + \gamma^2\left\|\nabla f_{\mathbf{S}^t}(x^t)\right\|^2.$$

Now we compute expectation of both sides of the inequality, conditional on $x^t$ :

$$\mathbb{E}\left[\left\|r^{t+1}\right\|^2 |x^t\right] = \left\|r^t\right\|^2 - 2\gamma\langle r^t, \mathbb{E}[\nabla f_{\mathbf{S}^t}(x^t)|x^t]\rangle + \gamma^2\mathbb{E}\left[\left\|\nabla f_{\mathbf{S}^t}(x^t)\right\|^2 |x^t\right].$$

Taking the expectation with respect to $\mathbf{S}_t$, using the fact that $f$ is continuously differentiable and using Lemma 4, we obtain that

$$\mathbb{E}\left[\left\|r^{t+1}\right\|^2\right] \leq \left\|r^t\right\|^2 - 2\gamma\langle r^t, \nabla f_{\mathcal{D}}(x^t)\rangle + 2\gamma^2 L_f L_{\mathbf{S}}^{\mathrm{max}}\left(\left(f_{\mathcal{D}}(x^t) - f_{\mathcal{D}}^{\mathrm{inf}}\right) + \left(f_{\mathcal{D}}^{\mathrm{inf}} - f^{\mathrm{inf}}\right)\right).$$

Since $f_{\mathcal{D}}$ is $\mu_{\mathcal{D}}\mu_f$-convex, we conclude that $\langle r^t, \nabla f_{\mathcal{D}}(x^t)\rangle \geq f_{\mathcal{D}}(x^t) - f_{\mathcal{D}}^{\mathrm{inf}} + \frac{\mu_{\mathcal{D}}\mu_f}{2}\left\|r^t\right\|^2$. Therefore,

$$\mathbb{E}\left[\left\|r^{t+1}\right\|^2\right] \leq (1 - \gamma\mu_{\mathcal{D}}\mu_f)\left\|r^t\right\|^2 - 2\gamma\left(f_{\mathcal{D}}(x^t) - f_{\mathcal{D}}^{\mathrm{inf}}\right)(1 - \gamma L_f L_{\mathbf{S}}^{\mathrm{max}})$$
$$+ 2\gamma^2 L_f L_{\mathbf{S}}^{\mathrm{max}}\left(f_{\mathcal{D}}^{\mathrm{inf}} - f^{\mathrm{inf}}\right).$$

Since $\gamma \leq \frac{1}{L_f L_{\mathbf{S}}^{\max}}$, taking expectation and using the tower property we get

$$\mathbb{E}\left[\left\|r^{t+1}\right\|^2\right] \leq (1 - \gamma\mu_{\mathcal{D}}\mu_f)\mathbb{E}\left[\left\|r^t\right\|^2\right] + 2\gamma^2 L_f L_{\mathbf{S}}^{\max}\left(f_{\mathcal{D}}^{\inf} - f^{\inf}\right).$$

Unrolling the recurrence, we get

$$\mathbb{E}\left[\left\|r^t\right\|^2\right] \leq (1 - \gamma\mu_{\mathcal{D}}\mu_f)^t\left\|r^0\right\|^2 + \frac{2\gamma L_f L_{\mathbf{S}}^{\max}\left(f_{\mathcal{D}}^{\inf} - f^{\inf}\right)}{\mu_{\mathcal{D}}\mu_f}.$$

$\square$

**Corollary 3.** *Fix $\delta > 0$. Choose the stepsize $\gamma > 0$ as*

$$\gamma = \min\left\{\frac{1}{L_f L_{\mathbf{S}}^{\max}}, \frac{\mu_{\mathcal{D}}\mu_f\delta\left\|r^0\right\|^2}{2L_f L_{\mathbf{S}}^{\max}\left(f_{\mathcal{D}}^{\inf} - f^{\inf}\right)}\right\}.$$

*Then, provided that*

$$t \geq \frac{L_f L_{\mathbf{S}}^{\max}}{\mu_{\mathcal{D}}\mu_f}\left\{1, \frac{2\left(f_{\mathcal{D}}^{\inf} - f^{\inf}\right)}{\mu_{\mathcal{D}}\mu_f\delta\left\|r^0\right\|^2}\right\}\log\frac{1}{\delta},$$

*we have $\mathbb{E}\left[\left\|r^t\right\|^2\right] \leq 2\delta\left\|r^0\right\|^2$.*

*Proof.* Since $\gamma \leq \frac{\mu_{\mathcal{D}}\mu_f\delta\left\|r^0\right\|^2}{2L_f L_{\mathbf{S}}^{\max}\left(f_{\mathcal{D}}^{\inf} - f^{\inf}\right)}$, we have that

$$\frac{2\gamma L_f L_{\mathbf{S}}^{\max}\left(f_{\mathcal{D}}^{\inf} - f^{\inf}\right)}{\mu_{\mathcal{D}}\mu_f} \leq \delta\left\|r^0\right\|^2.$$

If $\gamma = \frac{\mu_{\mathcal{D}}\mu_f\delta\left\|r^0\right\|^2}{2L_f L_{\mathbf{S}}^{\max}\left(f_{\mathcal{D}}^{\inf} - f^{\inf}\right)}$, then, since

$$t \geq \frac{2L_f L_{\mathbf{S}}^{\max}\left(f_{\mathcal{D}}^{\inf} - f^{\inf}\right)}{\mu_{\mathcal{D}}^2\mu_f^2\delta\left\|r^0\right\|^2}\log\frac{1}{\delta},$$

we obtain that

$$(1 - \gamma\mu_{\mathcal{D}}\mu_f)^t \leq \exp\left(-\gamma\mu_{\mathcal{D}}\mu_f t\right) \leq \delta.$$

Further, if $\gamma = \frac{1}{L_f L_{\mathbf{S}}^{\max}}$, then, since

$$t \geq \frac{L_f L_{\mathbf{S}}^{\max}}{\mu_{\mathcal{D}}\mu_f}\log\frac{1}{\delta},$$

we obtain that

$$(1 - \gamma\mu_{\mathcal{D}}\mu_f)^t \leq \exp\left(-\gamma\mu_{\mathcal{D}}\mu_f t\right) \leq \delta.$$

Thus, combining it with (29), we arrive at $\mathbb{E}\left[\left\|r^t\right\|^2\right] \leq 2\delta\left\|r^0\right\|^2$. $\square$

### D.3 CONVEX ANALYSIS

**Assumption 4.** *A set $\tilde{\mathcal{X}} = \{x_{\mathcal{D}}^\star \mid f_{\mathcal{D}}(x_{\mathcal{D}}^\star) \leq f_{\mathcal{D}}(x)\ \forall x \in \mathbb{R}^d\}$ is nonempty.*

**Theorem 10.** *Let $r^t \stackrel{def}{=} x^t - x_{\mathcal{D}}^\star$. Let Assumptions 1, 2 and 4 hold. Let $f$ be convex. Choose a stepsize $0 < \gamma \leq \frac{1}{2L_f L_{\mathbf{S}}^{\max}}$. Fix $T \geq 1$ and let $\bar{x}^T$ be chosen uniformly from the iterates $x^0, \ldots, x^{T-1}$ of Algorithm 1. Then*

$$\mathbb{E}\left[f_{\mathcal{D}}(\bar{x}^t) - f_{\mathcal{D}}^{\inf}\right] \leq \frac{\left\|r^0\right\|^2}{\gamma T} + 2\gamma L_f L_{\mathbf{S}}^{\max}\left(f_{\mathcal{D}}^{\inf} - f^{\inf}\right), \tag{30}$$

*where $r^t \stackrel{def}{=} x^t - x_{\mathcal{D}}^\star$, $t \in \{0, \ldots, T-1\}$.*

*Proof.* Let us start by analyzing the behavior of $\|x^t - x_{\mathcal{D}}^\star\|^2$. By developing the squares, we obtain

$$\left\|x^{t+1} - x_{\mathcal{D}}^\star\right\|^2 = \left\|x^t - x_{\mathcal{D}}^\star\right\|^2 - 2\gamma\langle\nabla f_{\mathbf{S}}(x^t), x^t - x_{\mathcal{D}}^\star\rangle + \gamma^2\left\|\nabla f_{\mathbf{S}}(x^t)\right\|^2$$

Hence, after taking the expectation with respect to $\mathbf{S}$ conditioned on $x^t$, we can use the convexity of $f$ and Lemma 4 to obtain:

$$\begin{aligned}
\mathbb{E}\left[\left\|x^{t+1} - x_{\mathcal{D}}^\star\right\|^2 | x^t\right] &= \left\|x^t - x_{\mathcal{D}}^\star\right\|^2 + 2\gamma\langle\nabla f_{\mathcal{D}}(x^t), x_{\mathcal{D}}^\star - x^t\rangle + \gamma^2\mathbb{E}\left[\left\|\nabla f_{\mathbf{S}}(x^t)\right\|^2\right] \\
&\leq \left\|x^t - x_{\mathcal{D}}^\star\right\|^2 + 2\gamma\left(\gamma L_f L_{\mathbf{S}}^{\max} - 1\right)\left(f_{\mathcal{D}}(x^t) - f_{\mathcal{D}}^{\inf}\right) \\
&\quad + 2\gamma^2 L_f L_{\mathbf{S}}^{\max}\left(f_{\mathcal{D}}^{\inf} - f^{\inf}\right).
\end{aligned}$$

Rearranging, taking expectation and taking into account the condition on the stepsize, we have

$$\gamma\mathbb{E}\left[f_{\mathcal{D}}(x^t) - f_{\mathcal{D}}^{\inf}\right] \leq \mathbb{E}\left[\left\|r^t\right\|^2\right] - \mathbb{E}\left[\left\|r^{t+1}\right\|^2\right] + 2\gamma^2 L_f L_{\mathbf{S}}^{\max}\left(f_{\mathcal{D}}^{\inf} - f^{\inf}\right).$$

Summing over $t = 0, \ldots, T-1$ and using telescopic cancellation gives:

$$\gamma\sum_{t=0}^{T-1}\left(\mathbb{E}\left[f_{\mathcal{D}}(x^t) - f_{\mathcal{D}}^{\inf}\right]\right) \leq \left\|r^0\right\|^2 - \mathbb{E}\left[\left\|r^T\right\|^2\right] + 2T\gamma^2 L_f L_{\mathbf{S}}^{\max}\left(f_{\mathcal{D}}^{\inf} - f^{\inf}\right).$$

Since $\mathbb{E}\left[\left\|r^T\right\|^2\right] \geq 0$, dividing both sides by $\gamma T$ gives:

$$\frac{1}{T}\sum_{t=0}^{T-1}\left(\mathbb{E}\left[f_{\mathcal{D}}(x^t) - f_{\mathcal{D}}^{\inf}\right]\right) \leq \frac{\left\|r^0\right\|^2}{\gamma T} + 2\gamma L_f L_{\mathbf{S}}^{\max}\left(f_{\mathcal{D}}^{\inf} - f^{\inf}\right).$$

We treat the $\left(\frac{1}{T}, \ldots, \frac{1}{T}\right)$ as if it is a probability vector. Indeed, using that $f_{\mathcal{D}}$ is convex together with Jensen's inequality gives

$$\mathbb{E}\left[f_{\mathcal{D}}(\bar{x}^t) - f_{\mathcal{D}}^{\inf}\right] \leq \frac{\left\|r^0\right\|^2}{\gamma T} + 2\gamma L_f L_{\mathbf{S}}^{\max}\left(f_{\mathcal{D}}^{\inf} - f^{\inf}\right).$$

$\square$

**Corollary 4.** *Fix $\delta > 0$. Choose the stepsize $\gamma > 0$ as*

$$\gamma = \min\left\{\frac{1}{2L_f\lambda_m^{\mathbf{S}}}, \frac{\delta\left\|r^0\right\|^2}{2L_f\lambda_m^{\mathbf{S}}\left(f_{\mathcal{D}}^{\inf} - f^{\inf}\right)}\right\}.$$

*Then, provided that*

$$T \geq \frac{2L_f\lambda_m^{\mathbf{S}}}{\delta}\max\left\{1, \frac{f_{\mathcal{D}}^{\inf} - f^{\inf}}{\delta\left\|r^0\right\|^2}\right\},$$

*we have $\mathbb{E}\left[f_{\mathcal{D}}(\bar{x}^t) - f_{\mathcal{D}}^{\inf}\right] \leq 2\delta\left\|r^0\right\|^2$.*

*Proof.* Since $\gamma \leq \frac{\delta\|r^0\|^2}{2L_f\lambda_m^{\mathbf{S}}\left(f_{\mathcal{D}}^{\inf} - f^{\inf}\right)}$, we have that

$$2\gamma L_f L_{\mathbf{S}}^{\max}\left(f_{\mathcal{D}}^{\inf} - f^{\inf}\right) \leq \delta\left\|r^0\right\|^2.$$

If $\gamma = \frac{\delta\|r^0\|^2}{2L_f\lambda_m^{\mathbf{S}}\left(f_{\mathcal{D}}^{\inf} - f^{\inf}\right)}$, then, since

$$T \geq \frac{2L_f\lambda_m^{\mathbf{S}}\left(f_{\mathcal{D}}^{\inf} - f^{\inf}\right)}{\delta^2\left\|r^0\right\|^2},$$

we obtain that $\frac{\|r^0\|^2}{\gamma T} \leq \delta\|r^0\|^2$. Further, if $\gamma = \frac{1}{2L_f\lambda_m^{\mathbf{S}}}$, then, since $T \geq \frac{2L_f\lambda_m^{\mathbf{S}}}{\delta}$, we have $\frac{\|r^0\|^2}{\gamma T} \leq \delta\|r^0\|^2$. Thus, combining it with (30), we arrive at $\mathbb{E}\left[f_{\mathcal{D}}(\bar{x}^t) - f_{\mathcal{D}}^{\inf}\right] \leq 2\delta\|r^0\|^2$. $\square$

# E (STOCHASTIC) INEXACT GRADIENT

## E.1 NONCONVEX ANALYSIS: PROOF OF THEOREM 4

We solve the problem (2) with the method

$$x^{t+1} = x^t - \gamma g^t, \tag{31}$$

where $g^t := g(x^t)$ is the gradient estimator that satisfies

$$\mathbb{E}\left[g(x)\right] = \nabla f_{\mathbf{S}}(x), \quad \forall x \in \mathbb{R}^d, \tag{32}$$

$$\mathbb{E}\left[\|g(x)\|^2\right] \leq 2A\left(f_{\mathbf{S}}(x) - f_{\mathbf{S}}^{\inf}\right) + B\|\nabla f_{\mathbf{S}}(x)\|^2 + C, \quad \forall x \in \mathbb{R}^d. \tag{33}$$

Recall that $D_{A,B} = A + BL_f L_{\mathbf{S}}^{\max}$.

**Theorem 11.** *Let Assumptions 1, 2, 15 and 16 hold. For every $t \geq 0$, put $\delta^t \overset{def}{=} \mathbb{E}\left[f_{\mathcal{D}}(x^t) - f_{\mathcal{D}}^{\inf}\right]$ and $r^t \overset{def}{=} \mathbb{E}\left[\|\nabla f_{\mathcal{D}}(x^t)\|^2\right]$. Then, for any $T \geq 1$, the iterates $\{x^t\}_{t=0}^{T-1}$ of Algorithm 14 satisfy*

$$\min_{0 \leq t < T} r^t \leq \frac{\left(1 + D_{A,B}L_f L_{\mathcal{D}}\gamma^2\right)^T}{\gamma T}\delta^0 + \frac{L_f L_{\mathcal{D}}\gamma}{2}\left(2\left(f_{\mathcal{D}}^{\inf} - f^{\inf}\right)D_{A,B} + C\right). \tag{34}$$

*Proof.* Due to $L_f$-smoothness of $f$, we have that

$$f(s + \mathbf{S}^{t+1}(x^{t+1} - s)) \leq f(s + \mathbf{S}^{t+1}(x^t - s))$$

$$+ \left\langle \nabla f(y)\left|_{y=s+\mathbf{S}^{t+1}(x^t-s)}, \mathbf{S}^{t+1}\left(x^{t+1} - s\right) - \mathbf{S}^{t+1}\left(x^t - s\right)\right.\right\rangle$$

$$+ \frac{L_f}{2}\left\|\mathbf{S}^{t+1}\left(x^{t+1} - s\right) - \mathbf{S}^{t+1}\left(x^t - s\right)\right\|^2$$

$$= f(s + \mathbf{S}^{t+1}(x^t - s)) - \gamma\left\langle \nabla f(y)\left|_{y=s+\mathbf{S}^{t+1}(x^t-s)}, \mathbf{S}^{t+1}g^t\right.\right\rangle$$

$$+ \frac{L_f \gamma^2}{2}\left\|\mathbf{S}^{t+1}g^t\right\|^2$$

$$\leq f(s + \mathbf{S}^{t+1}(x^t - s)) - \gamma\left\langle \nabla f_{\mathbf{S}^{t+1}}\left(x^t\right), g^t\right\rangle$$

$$+ \frac{L_f \gamma^2}{2}\left\langle \left(\mathbf{S}^{t+1}\right)^\top \mathbf{S}^{t+1}g^t, g^t\right\rangle.$$

Taking the expectation with respect to $\mathbf{S}^{t+1}$, we obtain that

$$f_{\mathcal{D}}(x^{t+1}) \leq f_{\mathcal{D}}(x^t) - \gamma\left\langle \nabla f_{\mathcal{D}}(x^t), g^t\right\rangle + \frac{L_f \gamma^2}{2}\left\langle \mathbb{E}\left[\left(\mathbf{S}^{t+1}\right)^\top \mathbf{S}^{t+1}\right]g^t, g^t\right\rangle$$

$$\leq f_{\mathcal{D}}(x^t) - \gamma\left\langle \nabla f_{\mathcal{D}}(x^t), g^t\right\rangle + \frac{L_f L_{\mathcal{D}}\gamma^2}{2}\left\|g^t\right\|^2.$$

Taking the expectation with respect to $\mathbf{S}^t$, conditional on $x^t$, using Lemma 4 and (16) we have that

$$\mathbb{E}\left[f_{\mathcal{D}}(x^{t+1})|x^t\right] \leq f_{\mathcal{D}}(x^t) - \gamma\left\langle \nabla f_{\mathcal{D}}(x^t), \mathbb{E}\left[g^t|x^t\right]\right\rangle + \frac{L_f L\gamma^2}{2}\mathbb{E}\left[\|g^t\|^2|x^t\right]$$

$$= f_{\mathcal{D}}(x^t) - \gamma\left\|\nabla f_{\mathcal{D}}(x^t)\right\|^2$$

$$+ \frac{L_f L_{\mathcal{D}}\gamma^2}{2}\mathbb{E}\left[2A\left(f_{\mathbf{S}_t}(x^t) - f_{\mathbf{S}_t}^{\inf}\right) + B\|\nabla f_{\mathbf{S}_t}(x^t)\|^2 + C\right]$$

$$\leq f_{\mathcal{D}}(x^t) - \gamma\left\|\nabla f_{\mathcal{D}}(x^t)\right\|^2 + AL_f L_{\mathcal{D}}\gamma^2\left(f_{\mathcal{D}}(x^t) - f_{\mathcal{D}}^{\inf}\right)$$

$$+ \frac{L_f L_{\mathcal{D}}B\gamma^2}{2}\left(2L_f L_{\mathbf{S}}^{\max}\left(f_{\mathcal{D}}(x^t) - f_{\mathcal{D}}^{\inf}\right) + 2L_f L_{\mathbf{S}}^{\max}\left(f_{\mathcal{D}}^{\inf} - f^{\inf}\right)\right)$$

$$+ \frac{L_f L_{\mathcal{D}}\gamma^2 C}{2} + AL_f L_{\mathcal{D}}\gamma^2\left(f_{\mathcal{D}}^{\inf} - \mathbb{E}\left[f_{\mathbf{S}^t}^{\inf}\right]\right).$$

Substitute $f_{\mathcal{D}}^{\mathrm{inf}}$ from both sides, take expectation on both sides and use the tower property:

$$\mathbb{E}\left[f_{\mathcal{D}}(x^{t+1}) - f_{\mathcal{D}}^{\mathrm{inf}}\right] \leq \mathbb{E}\left[f_{\mathcal{D}}(x^t) - f_{\mathcal{D}}^{\mathrm{inf}}\right]\left(1 + D_{A,B}L_f L_{\mathcal{D}}\gamma^2\right) - \gamma\mathbb{E}\left[\left\|\nabla f_{\mathcal{D}}(x^t)\right\|^2\right]$$
$$+ \frac{L_f L_{\mathcal{D}}\gamma^2}{2}\left(2\left(f_{\mathcal{D}}^{\mathrm{inf}} - f^{\mathrm{inf}}\right)D_{A,B} + C\right).$$

We obtain that

$$\gamma r^t \leq \left(1 + D_{A,B}L_f L_{\mathcal{D}}\gamma^2\right)\delta^t - \delta^{t+1} + \frac{L_f L_{\mathcal{D}}\gamma^2}{2}\left(2\left(f_{\mathcal{D}}^{\mathrm{inf}} - f^{\mathrm{inf}}\right)D_{A,B} + C\right).$$

Notice that the iterates $\{x^t\}_{t \geq 0}$ of Algorithm 14 satisfy condition (27) of Lemma 14 with $M_1 = D_{A,B}L_f L_{\mathcal{D}}$,

$$M_2 = \frac{L_f L_{\mathcal{D}}}{2}\left(2\left(f_{\mathcal{D}}^{\mathrm{inf}} - f^{\mathrm{inf}}\right)D_{A,B} + C\right).$$

Therefore, for any $T \geq 1$, the iterates $\{x^t\}_{t=0}^{T-1}$ of Algorithm 14 satisfy

$$\sum_{t=0}^{T-1} w_t r^t \leq \frac{w_{-1}}{\gamma}\delta^0 - \frac{w_{T-1}}{\gamma}\delta^T$$

$$+ \frac{L_f L_{\mathcal{D}}\gamma}{2}\left(2A\left(f_{\mathcal{D}}^{\mathrm{inf}} - \mathbb{E}\left[f_{\mathbf{S}^t}^{\mathrm{inf}}\right]\right) + 2BL_f L_{\mathbf{S}}^{\max}\left(f_{\mathcal{D}}^{\mathrm{inf}} - f^{\mathrm{inf}}\right) + C\right)\sum_{t=0}^{T-1} w_t.$$

Divide both sides by $\sum_{t=0}^{T-1} w_t$. From $\sum_{t=0}^{T-1} w_t \geq T w_{T-1} = \frac{T w_{-1}}{1 + D_{A,B}L_f L_{\mathcal{D}}\gamma^2}$ we can conclude that

$$\min_{0 \leq t < T} r^t \leq \frac{\left(1 + D_{A,B}L_f L_{\mathcal{D}}\gamma^2\right)^T}{\gamma T}\delta^0 + \frac{L_f L_{\mathcal{D}}\gamma}{2}\left(2\left(f_{\mathcal{D}}^{\mathrm{inf}} - f^{\mathrm{inf}}\right)D_{A,B} + C\right).$$

$\square$

**Corollary 5.** *Fix $\varepsilon > 0$. Choose the stepsize $\gamma > 0$ as*

$$\gamma = \min\left\{\frac{1}{\sqrt{L_f L_{\mathcal{D}}D_{A,B}T}}, \frac{\varepsilon^2}{L_f L_{\mathcal{D}}\left(2\left(f_{\mathcal{D}}^{\mathrm{inf}} - f^{\mathrm{inf}}\right)D_{A,B} + C\right)}\right\}.$$

*Then, provided that*

$$T \geq \frac{6\delta^0 L_f L_{\mathcal{D}}}{\varepsilon^4}\max\left\{6\delta^0 D_{A,B}, \ 2\left(f_{\mathcal{D}}^{\mathrm{inf}} - f^{\mathrm{inf}}\right)D_{A,B} + C\right\},$$

*we have*

$$\min_{0 \leq t < T}\mathbb{E}\left[\left\|\nabla f_{\mathcal{D}}(x^t)\right\|^2\right] \leq \varepsilon^2.$$

*Proof.* Since $\gamma \leq \frac{\varepsilon^2}{L_f L_{\mathcal{D}}\left(2\left(f_{\mathcal{D}}^{\mathrm{inf}} - f^{\mathrm{inf}}\right)D_{A,B} + C\right)}$, we obtain

$$\frac{L_f L_{\mathcal{D}}\gamma}{2}\left(2\left(f_{\mathcal{D}}^{\mathrm{inf}} - f^{\mathrm{inf}}\right)D_{A,B} + C\right) \leq \frac{\varepsilon^2}{2}.$$

Since $\gamma \leq \frac{1}{\sqrt{L_f L_{\mathcal{D}}D_{A,B}T}}$, we deduce that

$$\left(1 + D_{A,B}L_f L_{\mathcal{D}}\gamma^2\right)^T \leq \exp\left(T L_f L_{\mathcal{D}}\gamma^2 D_{A,B}\right)$$
$$\leq \exp(1) \leq 3.$$

If $\gamma = \frac{1}{\sqrt{L_f L_{\mathcal{D}}D_{A,B}T}}$, then, since

$$T \geq \frac{36\left(\delta^0\right)^2 L_f L_{\mathcal{D}}D_{A,B}}{\varepsilon^4},$$

we have $\frac{3\delta^0}{\gamma T} \leq \frac{\varepsilon^2}{2}$. Further, if $\gamma = \frac{\varepsilon^2}{L_f L_{\mathcal{D}}\left(2\left(f_{\mathcal{D}}^{\mathrm{inf}} - f^{\mathrm{inf}}\right)D_{A,B} + C\right)}$, then, since

$$T \geq \frac{6\delta^0 L_f L_{\mathcal{D}}\left(2\left(f_{\mathcal{D}}^{\mathrm{inf}} - f^{\mathrm{inf}}\right)D_{A,B} + C\right)}{\varepsilon^4},$$

we have $\frac{3\delta^0}{\gamma T} \leq \frac{\varepsilon^2}{2}$. Combining it with (34), we arrive at $\min_{0 \leq t < T}\mathbb{E}\left[\left\|\nabla f_{\mathcal{D}}(x^t)\right\|^2\right] \leq \varepsilon^2$. $\square$

### E.2 STRONGLY CONVEX ANALYSIS

**Theorem 12.** *Let Assumptions [1], [2], [3], [15] and [16] hold. Let $r^t \stackrel{def}{=} x^t - x_{\mathcal{D}}^\star$, $t \geq 0$. Choose a stepsize $0 < \gamma \leq \frac{1}{D_{A,B}}$. Then the iterates $\{x^t\}_{t \geq 0}$ of Algorithm [14] satisfy*

$$\mathbb{E}\left[\|r^t\|^2\right] \leq (1 - \gamma\mu_{\mathcal{D}}\mu_f)^t \|r^0\|^2 + \frac{\gamma\left(2\left(f_{\mathcal{D}}^{\text{inf}} - f^{\text{inf}}\right)D_{A,B} + C\right)}{\mu_{\mathcal{D}}\mu_f}. \tag{35}$$

*Proof.* We get

$$\|r^{t+1}\|^2 = \|(x^t - \gamma g^t) - x_{\mathcal{D}}^\star\|^2 = \|x^t - x_{\mathcal{D}}^\star - \gamma g^t\|^2$$
$$= \|r^t\|^2 - 2\gamma\langle r^t, g^t\rangle + \gamma^2 \|g^t\|^2.$$

Now we compute expectation of both sides of the inequality with respect to $\mathbf{S}^t$, conditioned on $x^t$, use the fact that $f$ is continuously differentiable and use ([16]):

$$\mathbb{E}\left[\|r^{t+1}\|^2 |x^t\right] = \|r^t\|^2 - 2\gamma\langle r^t, \mathbb{E}\left[g^t|x^t\right]\rangle + \gamma^2\mathbb{E}\left[\|g^t\|^2 |x^t\right]$$
$$\leq \|r^t\|^2 - 2\gamma\langle r^t, \mathbb{E}\left[\nabla f_{\mathbf{S}^t}(x^t)\right]\rangle$$
$$+ \gamma^2\mathbb{E}\left[\left(2A\left(f_{\mathbf{S}_t}(x^t) - f_{\mathbf{S}_t}^{\text{inf}}\right) + B\|\nabla f_{\mathbf{S}_t}(x^t)\|^2 + C\right)\right]$$
$$= \|r^t\|^2 - 2\gamma\langle r^t, \nabla f_{\mathcal{D}}(x^t)\rangle + 2A\gamma^2\left(f_{\mathcal{D}}(x^t) - f_{\mathcal{D}}^{\text{inf}}\right) + \gamma^2 B\mathbb{E}\left[\|\nabla f_{\mathbf{S}_t}(x^t)\|^2\right]$$
$$+ \gamma^2\left(C + 2A\left(f_{\mathcal{D}}^{\text{inf}} - \mathbb{E}\left[f_{\mathbf{S}_t}^{\text{inf}}\right]\right)\right).$$

Since $f_{\mathcal{D}}$ is $\mu_{\mathcal{D}}\mu_f$-convex, we conclude that $\langle r^t, \nabla f_{\mathcal{D}}(x^t)\rangle \geq f_{\mathcal{D}}(x^t) - f_{\mathcal{D}}^{\text{inf}} + \frac{\mu_{\mathcal{D}}\mu_f}{2}\|r^t\|^2$. Therefore, taking the expectation and using the tower property, we obtain

$$\mathbb{E}\left[\|r^{t+1}\|^2\right] \leq (1 - \gamma\mu_{\mathcal{D}}\mu_f)\|r^t\|^2 - 2\gamma\left(f_{\mathcal{D}}(x) - f_{\mathcal{D}}^{\text{inf}}\right)(1 - \gamma D_{A,B})$$
$$+ \gamma^2\left(2\left(f_{\mathcal{D}}^{\text{inf}} - f^{\text{inf}}\right)D_{A,B} + C\right).$$

Since $\gamma \leq \frac{1}{D_{A,B}}$, we get

$$\mathbb{E}\left[\|r^{t+1}\|^2\right] \leq (1 - \gamma\mu_{\mathcal{D}}\mu_f)\mathbb{E}\left[\|r^t\|^2\right] + \gamma^2\left(2\left(f_{\mathcal{D}}^{\text{inf}} - f^{\text{inf}}\right)D_{A,B} + C\right).$$

Unrolling the recurrence, we get

$$\mathbb{E}\left[\|r^t\|^2\right] \leq (1 - \gamma\mu_{\mathcal{D}}\mu_f)^t \|r^0\|^2 + \frac{\gamma\left(2\left(f_{\mathcal{D}}^{\text{inf}} - f^{\text{inf}}\right)D_{A,B} + C\right)}{\mu_{\mathcal{D}}\mu_f}.$$

$\square$

**Corollary 6.** *Fix $\delta > 0$. Choose the stepsize $\gamma > 0$ as*

$$\gamma = \min\left\{\frac{1}{D_{A,B}}, \frac{\mu_{\mathcal{D}}\mu_f\delta\|r^0\|^2}{2\left(f_{\mathcal{D}}^{\text{inf}} - f^{\text{inf}}\right)D_{A,B} + C}\right\}.$$

*Then, provided that*

$$t \geq \frac{1}{\mu_{\mathcal{D}}\mu_f}\left\{D_{A,B}, \frac{2\left(f_{\mathcal{D}}^{\text{inf}} - f^{\text{inf}}\right)D_{A,B} + C}{\mu_{\mathcal{D}}\mu_f\delta\|r^0\|^2}\right\}\log\frac{1}{\delta},$$

*we have $\mathbb{E}\left[\|r^t\|^2\right] \leq 2\delta\|r^0\|^2$.*

*Proof.* Since $\gamma \leq \frac{\mu_{\mathcal{D}}\mu_f\delta\|r^0\|^2}{2\left(f_{\mathcal{D}}^{\text{inf}} - f^{\text{inf}}\right)D_{A,B} + C}$, we have that

$$\frac{\gamma\left(2\left(f_{\mathcal{D}}^{\text{inf}} - f^{\text{inf}}\right)D_{A,B} + C\right)}{\mu_{\mathcal{D}}\mu_f} \leq \delta\|r^0\|^2.$$

If $\gamma = \frac{\mu_{\mathcal{D}}\mu_f \delta \|r^0\|^2}{2\left(f_{\mathcal{D}}^{\text{inf}} - f^{\text{inf}}\right)D_{A,B} + C}$, then, since

$$t \geq \frac{2\left(f_{\mathcal{D}}^{\text{inf}} - f^{\text{inf}}\right)D_{A,B} + C}{\mu_{\mathcal{D}}^2 \mu_f^2 \delta \|r^0\|^2} \log \frac{1}{\delta},$$

we obtain that

$$\left(1 - \gamma\mu_{\mathcal{D}}\mu_f\right)^t \leq \exp\left(-\gamma\mu_{\mathcal{D}}\mu_f t\right) \leq \delta.$$

Further, if $\gamma = \frac{1}{D_{A,B}}$, then, since

$$t \geq \frac{D_{A,B}}{\mu_{\mathcal{D}}\mu_f} \log \frac{1}{\delta},$$

we obtain that

$$\left(1 - \gamma\mu_{\mathcal{D}}\mu_f\right)^t \leq \exp\left(-\gamma\mu_{\mathcal{D}}\mu_f t\right) \leq \delta.$$

Thus, combining it with (35), we arrive at $\mathbb{E}\left[\|r^t\|^2\right] \leq 2\delta \|r^0\|^2$. $\qquad\square$

### E.3 Convex analysis

**Theorem 13.** *Let Assumptions 1, 2, 4, 15 and 16 hold. Let $f$ be convex. Choose a stepsize $0 < \gamma \leq \frac{1}{2D_{A,B}}$. Fix $T \geq 1$ and let $\bar{x}^T$ be chosen uniformly from the iterates $x^0, \ldots, x^{T-1}$ of Algorithm 14. Then*

$$\mathbb{E}\left[f_{\mathcal{D}}(\bar{x}^t) - f_{\mathcal{D}}^{\text{inf}}\right] \leq \frac{\|r^0\|^2}{\gamma T} + 2\gamma\left(f_{\mathcal{D}}^{\text{inf}} - f^{\text{inf}}\right)D_{A,B} + \gamma C, \tag{36}$$

*where $r^t \overset{\text{def}}{=} x^t - x_{\mathcal{D}}^\star$, $t \in \{0, \ldots, T-1\}$.*

*Proof.* We get

$$\left\|r^{t+1}\right\|^2 = \left\|\left(x^t - \gamma g^t\right) - x_{\mathcal{D}}^\star\right\|^2 = \left\|x^t - x_{\mathcal{D}}^\star - \gamma g^t\right\|^2$$
$$= \left\|r^t\right\|^2 - 2\gamma\langle r^t, g^t\rangle + \gamma^2 \left\|g^t\right\|^2.$$

Now we compute expectation of both sides of the inequality with respect to $\mathbf{S}^t$, conditioned on $x^t$, use the fact that $f$ is continuously differentiable and use (16):

$$\mathbb{E}\left[\left\|r^{t+1}\right\|^2 |x^t\right] = \left\|r^t\right\|^2 - 2\gamma\langle r^t, \mathbb{E}\left[g^t|x^t\right]\rangle + \gamma^2 \mathbb{E}\left[\left\|g^t\right\|^2 |x^t\right]$$
$$\leq \left\|r^t\right\|^2 - 2\gamma\langle r^t, \mathbb{E}\left[\nabla f_{\mathbf{S}^t}(x^t)\right]\rangle$$
$$+ \gamma^2\mathbb{E}\left[\left(2A\left(f_{\mathbf{S}_t}(x^t) - f_{\mathbf{S}_t}^{\text{inf}}\right) + B\left\|\nabla f_{\mathbf{S}_t}(x^t)\right\|^2 + C\right)\right]$$
$$= \left\|r^t\right\|^2 - 2\gamma\langle r^t, \nabla f_{\mathcal{D}}(x^t)\rangle + 2A\gamma^2\left(f_{\mathcal{D}}(x^t) - f_{\mathcal{D}}^{\text{inf}}\right) + \gamma^2 B\mathbb{E}\left[\left\|\nabla f_{\mathbf{S}_t}(x^t)\right\|^2\right]$$
$$+ \gamma^2\left(C + 2A\left(f_{\mathcal{D}}^{\text{inf}} - \mathbb{E}\left[f_{\mathbf{S}_t}^{\text{inf}}\right]\right)\right).$$

We can use the convexity of $f$ and Lemma 4 to obtain:

$$\mathbb{E}\left[\left\|r^{t+1}\right\|^2\right] = \left\|r^t\right\|^2 - 2\gamma\left(f_{\mathcal{D}}(x^t) - f_{\mathcal{D}}^{\text{inf}}\right)(1 - A\gamma) + 2\gamma^2 BL_f L_{\mathbf{S}}^{\max}\left(f_{\mathcal{D}}(x^t) - f_{\mathcal{D}}^{\text{inf}}\right)$$
$$+ 2\gamma^2 BL_f L_{\mathbf{S}}^{\max}\left(f_{\mathcal{D}}^{\text{inf}} - f^{\text{inf}}\right) + \gamma^2\left(C + 2A\left(f_{\mathcal{D}}^{\text{inf}} - \mathbb{E}\left[f_{\mathbf{S}_t}^{\text{inf}}\right]\right)\right)$$
$$\leq \left\|r^t\right\|^2 - 2\gamma\left(f_{\mathcal{D}}(x^t) - f_{\mathcal{D}}^{\text{inf}}\right)(1 - \gamma D_{A,B})$$
$$+ 2\gamma^2 BL_f L_{\mathbf{S}}^{\max}\left(f_{\mathcal{D}}^{\text{inf}} - f^{\text{inf}}\right) + \gamma^2\left(C + 2A\left(f_{\mathcal{D}}^{\text{inf}} - f^{\text{inf}}\right)\right).$$

Rearranging and taking expectation, taking into account the condition on the stepsize, we have

$$\gamma\left(f_{\mathcal{D}}(x^t) - f_{\mathcal{D}}^{\text{inf}}\right) \leq \mathbb{E}\left[\left\|r^t\right\|^2\right] - \mathbb{E}\left[\left\|r^{t+1}\right\|^2\right] + 2\gamma^2\left(f_{\mathcal{D}}^{\text{inf}} - f^{\text{inf}}\right)D_{A,B} + \gamma^2 C.$$

Summing over $t = 0, \ldots, T-1$ and using telescopic cancellation gives

$$\gamma \sum_{t=0}^{T-1} \left( \mathbb{E}\left[ f_{\mathcal{D}}(x^t) - f_{\mathcal{D}}^{\inf} \right] \right) \le \left\| r^0 \right\|^2 - \mathbb{E}\left[ \left\| r^T \right\|^2 \right] + 2\gamma^2 T \left( f_{\mathcal{D}}^{\inf} - f^{\inf} \right) D_{A,B} + \gamma^2 TC.$$

Since $\mathbb{E}\left[ \left\| r^T \right\|^2 \right] \ge 0$, dividing both sides by $\gamma T$ gives:

$$\frac{1}{T} \sum_{t=0}^{T-1} \left( \mathbb{E}\left[ f_{\mathcal{D}}(x^t) - f_{\mathcal{D}}^{\inf} \right] \right) \le \frac{\left\| r^0 \right\|^2}{\gamma T} + 2\gamma \left( f_{\mathcal{D}}^{\inf} - f^{\inf} \right) D_{A,B} + \gamma C.$$

We treat the $\left( \frac{1}{T}, \ldots, \frac{1}{T} \right)$ as if it is a probability vector. Indeed, using that $f_{\mathcal{D}}$ is convex together with Jensen's inequality gives

$$\mathbb{E}\left[ f_{\mathcal{D}}(\bar{x}^t) - f_{\mathcal{D}}^{\inf} \right] \le \frac{\left\| r^0 \right\|^2}{\gamma T} + \gamma \left( 2 \left( f_{\mathcal{D}}^{\inf} - f^{\inf} \right) D_{A,B} + C \right).$$

$\square$

**Corollary 7.** *Fix $\delta > 0$. Choose the stepsize $\gamma > 0$ as*

$$\gamma = \min \left\{ \frac{1}{2D_{A,B}}, \frac{\delta \left\| r^0 \right\|^2}{2 \left( f_{\mathcal{D}}^{\inf} - f^{\inf} \right) D_{A,B} + C} \right\}.$$

*Then, provided that*

$$T \ge \frac{2L_f \lambda_m^{\mathbf{S}}}{\delta} \max \left\{ 1, \frac{f_{\mathcal{D}}^{\inf} - f^{\inf}}{\delta \left\| r^0 \right\|^2} \right\},$$

*we have $\mathbb{E}\left[ f_{\mathcal{D}}(\bar{x}^t) - f_{\mathcal{D}}^{\inf} \right] \le 2\delta \left\| r^0 \right\|^2$.*

*Proof.* Since $\gamma \le \frac{\delta \left\| r^0 \right\|^2}{2 \left( f_{\mathcal{D}}^{\inf} - f^{\inf} \right) D_{A,B} + C}$, we have that

$$\gamma \left( 2 \left( f_{\mathcal{D}}^{\inf} - f^{\inf} \right) D_{A,B} + C \right) \le \delta \left\| r^0 \right\|^2.$$

If $\gamma = \frac{\delta \left\| r^0 \right\|^2}{2 \left( f_{\mathcal{D}}^{\inf} - f^{\inf} \right) D_{A,B} + C}$, then, since

$$T \ge \frac{2 \left( f_{\mathcal{D}}^{\inf} - f^{\inf} \right) D_{A,B} + C}{\delta^2 \left\| r^0 \right\|^2},$$

we obtain that $\frac{\left\| r^0 \right\|^2}{\gamma T} \le \delta \left\| r^0 \right\|^2$. Further, if $\gamma = \frac{1}{2D_{A,B}}$, then, since $T \ge \frac{2D_{A,B}}{\delta}$, we have $\frac{\left\| r^0 \right\|^2}{\gamma T} \le \delta \left\| r^0 \right\|^2$. Thus, combining it with (36), we arrive at $\mathbb{E}\left[ f_{\mathcal{D}}(\bar{x}^t) - f_{\mathcal{D}}^{\inf} \right] \le 2\delta \left\| r^0 \right\|^2$. $\square$

# F DISTRIBUTED SETTING

Consider $f$ being a finite sum over a number of machines, i.e., we consider the distributed setup:

$$\min_{x \in \mathbb{R}^d} \left[ f_{\mathcal{D}}(x) = \frac{1}{n} \sum_{i=1}^{n} f_{i,\mathcal{D}_i}(x) \right],$$

where $f_{i,\mathcal{D}_i} \stackrel{\text{def}}{=} \mathbb{E}\left[ f_{i,\mathbf{S}_i}(x) \right] = \mathbb{E}\left[ f_i(v + \mathbf{S}_i(x - v)) \right]$.

Recall that $D_{\max} = \max_i \left\{ L_{f_i}^2 L_{D_i} L_{\mathbf{S}_i}^{\max} \right\}$.

## F.1 NONCONVEX ANALYSIS: PROOF OF THEOREM 5

We solve the problem (2) with the method

$$x^{t+1} = x^t - \frac{\gamma}{n} \sum_{i=1}^{n} \left( \mathbf{S}_i^t \right)^\top \nabla f_i(y^t)|_{y^t = v + \mathbf{S}_i^t(x^t - v)}. \tag{37}$$

**Theorem 14.** *Assume that each $f_i$, $i \in [n]$, is differentiable, $L_{f_i}$-smooth and bounded from below by $f_i^{\text{inf}}$. For every $t \geq 0$, put $\delta^t \stackrel{\text{def}}{=} \mathbb{E}\left[ f_{\mathcal{D}}(x^t) - f_{\mathcal{D}}^{\text{inf}} \right]$ and $r^t \stackrel{\text{def}}{=} \mathbb{E}\left[ \|\nabla f_{\mathcal{D}}(x^t)\|^2 \right]$. Fix $T \geq 1$. Then the iterates $\{x^t\}_{t=0}^{T-1}$ of Algorithm 19 satisfy*

$$\min_{0 \leq t < T} r^t \leq \frac{\left( 1 + \gamma^2 D_{\max} \right)^T}{\gamma T} \delta^0 + \gamma D_{\max} \left( f_{\mathcal{D}}^{\text{inf}} - \frac{1}{n} \sum_{i=1}^{n} f_i^{\text{inf}} \right). \tag{38}$$

*Proof.* For $i \in [n]$, due to $L_{f_i}$-smoothness of $f_i$, we have that

$$f_i(s + \mathbf{S}_i^{t+1}(x^{t+1} - s)) \leq f_i(s + \mathbf{S}_i^{t+1}(x^t - s)) - \gamma \left\langle \nabla f_{i,\mathbf{S}_i^{t+1}}\left( x^t \right), \nabla f_{i,\mathbf{S}_i^t}(x^t) \right\rangle$$
$$+ \frac{L_{f_i} \gamma^2}{2} \left\langle \left( \mathbf{S}_i^{t+1} \right)^\top \mathbf{S}_i^{t+1} \nabla f_{i,\mathbf{S}_i^t}(x^t), \nabla f_{i,\mathbf{S}_i^t}(x^t) \right\rangle.$$

Taking the expectation with respect to $\mathbf{S}_i^{t+1}$ yields

$$f_{i,\mathcal{D}_i}\left( x^{t+1} \right) \leq f_{i,\mathcal{D}_i}\left( x^t \right) - \gamma \left\langle \nabla f_{i,\mathcal{D}_i}\left( x^t \right), \nabla f_{\mathbf{S}_i^t}(x^t) \right\rangle + \frac{L_{f_i} L_{\mathcal{D}_i} \gamma^2}{2} \left\| \nabla f_{i,\mathbf{S}_i^t}(x^t) \right\|^2.$$

Conditioned on $x^t$, take expectation with respect to $\mathbf{S}_i^t$:

$$\mathbb{E}\left[ f_{i,\mathcal{D}_i}\left( x^{t+1} \right) | x^t \right] \leq f_{i,\mathcal{D}_i}\left( x^t \right) - \gamma \left\| \nabla f_{i,\mathcal{D}_i}\left( x^t \right) \right\|^2 + \frac{L_{f_i} L_{\mathcal{D}_i} \gamma^2}{2} \mathbb{E}\left[ \left\| \nabla f_{i,\mathbf{S}_i^t}(x^t) \right\|^2 \right].$$

From Lemma 4 we obtain that

$$\mathbb{E}\left[ f_{i,\mathcal{D}_i}\left( x^{t+1} \right) | x^t \right] \leq f_{i,\mathcal{D}_i}\left( x^t \right) - \gamma \left\| \nabla f_{i,\mathcal{D}_i}\left( x^t \right) \right\|^2 + L_{f_i}^2 L_{\mathcal{D}_i} L_{\mathbf{S}_i}^{\max} \gamma^2 \left( f_{i,\mathcal{D}_i}(x) - f_i^{\text{inf}} \right).$$

For every $i \in [n]$, sum these inequalities, divide by $n$:

$$\mathbb{E}\left[ f_{\mathcal{D}}\left( x^{t+1} \right) | x^t \right] \leq f_{\mathcal{D}}\left( x^t \right) - \frac{\gamma}{n} \sum_{i=1}^{n} \left\| \nabla f_{i,\mathcal{D}_i}\left( x^t \right) \right\|^2 + \frac{\gamma^2 D_{\max}}{n} \sum_{i=1}^{n} \left( f_{i,\mathcal{D}_i}(x^t) - f_i^{\text{inf}} \right)$$

$$= f_{\mathcal{D}}\left( x^t \right) - \frac{\gamma}{n} \sum_{i=1}^{n} \left\| \nabla f_{i,\mathcal{D}_i}\left( x^t \right) \right\|^2 + \gamma^2 D_{\max} \left( f_{\mathcal{D}}(x^t) - \frac{1}{n} \sum_{i=1}^{n} f_i^{\text{inf}} \right)$$

$$= f_{\mathcal{D}}\left( x^t \right) - \frac{\gamma}{n} \sum_{i=1}^{n} \left\| \nabla f_{i,\mathcal{D}_i}\left( x^t \right) \right\|^2 + \gamma^2 D_{\max} \left( f_{\mathcal{D}}(x^t) - f_{\mathcal{D}}^{\text{inf}} \right)$$

$$+ \gamma^2 D_{\max} \left( f_{\mathcal{D}}^{\text{inf}} - \frac{1}{n} \sum_{i=1}^{n} f_i^{\text{inf}} \right).$$

Notice that by Jensen's inequality

$$\frac{1}{n} \sum_{i=1}^{n} \left\| \nabla f_{i, \mathcal{D}_i} \left( x^t \right) \right\|^2 \geq \left\| \frac{1}{n} \sum_{i=1}^{n} \nabla f_{i, \mathcal{D}_i} \left( x^t \right) \right\|^2 = \left\| \nabla f_{\mathcal{D}}(x^t) \right\|^2.$$

Subtract $f_{\mathcal{D}}^{\text{inf}}$ from both sides, take expectation on both sides and use the tower property:

$$\mathbb{E} \left[ f_{\mathcal{D}} \left( x^{t+1} \right) - f_{\mathcal{D}}^{\text{inf}} \right] \leq \left( 1 + \gamma^2 D_{\max} \right) \mathbb{E} \left[ f_{\mathcal{D}} \left( x^t \right) - f_{\mathcal{D}}^{\text{inf}} \right] - \gamma \mathbb{E} \left[ \left\| \nabla f_{\mathcal{D}} \left( x^t \right) \right\|^2 \right]$$
$$+ \gamma^2 D_{\max} \left( f_{\mathcal{D}}^{\text{inf}} - \frac{1}{n} \sum_{i=1}^{n} f_i^{\text{inf}} \right).$$

We obtain that

$$\gamma r^t \leq \left( 1 + \gamma^2 D_{\max} \right) \delta^t - \delta^{t+1} + \gamma^2 D_{\max} \left( f_{\mathcal{D}}^{\text{inf}} - \frac{1}{n} \sum_{i=1}^{n} f_i^{\text{inf}} \right).$$

Notice that the iterates $\{x^t\}_{t \geq 0}$ of Algorithm 14 satisfy condition (27) of Lemma 14 with $M_1 = D_{\max}$,

$$M_2 = D_{\max} \left( f_{\mathcal{D}}^{\text{inf}} - \frac{1}{n} \sum_{i=1}^{n} f_i^{\text{inf}} \right).$$

Divide both sides by $\sum_{t=0}^{T-1} w_t$. From $\sum_{t=0}^{T-1} w_t \geq T w_{T-1} = \frac{T w_{-1}}{1 + \gamma^2 D_{\max}}$ we can conclude that

$$\min_{0 \leq t < T} r^t \leq \frac{\left( 1 + \gamma^2 D_{\max} \right)^T}{\gamma T} \delta^0 + \gamma D_{\max} \left( f_{\mathcal{D}}^{\text{inf}} - \frac{1}{n} \sum_{i=1}^{n} f_i^{\text{inf}} \right).$$

$\square$

**Corollary 8.** *Fix $\varepsilon > 0$. Choose the stepsize $\gamma > 0$ as*

$$\gamma = \min \left\{ \frac{1}{\sqrt{D_{\max} T}}, \frac{\varepsilon^2}{2 D_{\max} \left( f_{\mathcal{D}}^{\text{inf}} - \frac{1}{n} \sum_{i=1}^{n} f_i^{\text{inf}} \right)} \right\}.$$

*Then, provided that*

$$T \geq \frac{12 \delta^0 D_{\max}}{\varepsilon^4} \max \left\{ 3 \delta^0, f_{\mathcal{D}}^{\text{inf}} - \frac{1}{n} \sum_{i=1}^{n} f_i^{\text{inf}} \right\},$$

*we have*

$$\min_{0 \leq t < T} \mathbb{E} \left[ \left\| \nabla f_{\mathcal{D}}(x^t) \right\|^2 \right] \leq \varepsilon^2.$$

*Proof.* Since $\gamma \leq \frac{\varepsilon^2}{2 D_{\max} \left( f_{\mathcal{D}}^{\text{inf}} - \frac{1}{n} \sum_{i=1}^{n} f_i^{\text{inf}} \right)}$, we obtain

$$\gamma D_{\max} \left( f_{\mathcal{D}}^{\text{inf}} - \frac{1}{n} \sum_{i=1}^{n} f_i^{\text{inf}} \right) \leq \frac{\varepsilon^2}{2}.$$

Since $\gamma \leq \frac{1}{\sqrt{D_{\max} T}}$, we deduce that

$$\left( 1 + \gamma^2 D_{\max} \right)^T \leq \exp \left( T \gamma^2 D_{\max} \right)$$
$$\leq \exp(1) \leq 3.$$

If $\gamma = \frac{1}{\sqrt{D_{\max} T}}$, then, since

$$T \geq \frac{36 \left( \delta^0 \right)^2 D_{\max}}{\varepsilon^4},$$

we have $\frac{3 \delta^0}{\gamma T} \leq \frac{\varepsilon^2}{2}$. Further, if $\gamma = \frac{\varepsilon^2}{2 D_{\max} \left( f_{\mathcal{D}}^{\text{inf}} - \frac{1}{n} \sum_{i=1}^{n} f_i^{\text{inf}} \right)}$, then, since

$$T \geq \frac{12 \delta^0 D_{\max} \left( f_{\mathcal{D}}^{\text{inf}} - \frac{1}{n} \sum_{i=1}^{n} f_i^{\text{inf}} \right)}{\varepsilon^4},$$

we have $\frac{3 \delta^0}{\gamma T} \leq \frac{\varepsilon^2}{2}$. Combining it with (38), we arrive at $\min_{0 \leq t < T} \mathbb{E} \left[ \left\| \nabla f_{\mathcal{D}}(x^t) \right\|^2 \right] \leq \varepsilon^2$. $\square$

### F.2 STRONGLY CONVEX ANALYSIS

**Theorem 15.** *Assume that each $f_i$, $i \in [n]$, is differentiable, $L_{f_i}$-smooth and bounded from below by $f_i^{\mathrm{inf}}$, $f$ is $\mu_f$-convex (Assumption 3 holds). Choose a stepsize $0 < \gamma \leq \frac{1}{\max_i \left\{ L_{f_i} L_{\mathbf{S}_i}^{\max} \right\}}$. Then the iterations $\{x^t\}_{t \geq 0}$ of Algorithm 19 satisfy*

$$\mathbb{E}\left[\left\| r^t \right\|^2\right] \leq (1 - \gamma \mu_{\mathcal{D}} \mu_f)^t \left\| r^0 \right\|^2 + \frac{2\gamma \max_i \left\{ L_{f_i} L_{\mathbf{S}_i}^{\max} \right\} \left( f_{\mathcal{D}}^{\mathrm{inf}} - \frac{1}{n} \sum_{i=1}^n f_i^{\mathrm{inf}} \right)}{\mu \mu_f},$$

*where $r^t \overset{def}{=} x^t - x_{\mathcal{D}}^{\star}$.*

*Proof.* We get that

$$\left\| r^{t+1} \right\|^2 = \left\| \left( x^t - \frac{\gamma}{n} \sum_{i=1}^n \nabla f_{i,\mathbf{S}_i^t}(x^t) \right) - x_{\mathcal{D}}^{\star} \right\|^2 = \left\| x^t - x_{\mathcal{D}}^{\star} - \frac{\gamma}{n} \sum_{i=1}^n \nabla f_{i,\mathbf{S}_i^t}(x^t) \right\|^2$$

$$= \left\| r^t \right\|^2 - 2\gamma \left\langle r^t, \frac{1}{n} \sum_{i=1}^n \nabla f_{i,\mathbf{S}_i^t}(x^t) \right\rangle + \gamma^2 \left\| \frac{1}{n} \sum_{i=1}^n \nabla f_{i,\mathbf{S}_i^t}(x^t) \right\|^2$$

$$\leq \left\| r^t \right\|^2 - 2\gamma \left\langle r^t, \frac{1}{n} \sum_{i=1}^n \nabla f_{i,\mathbf{S}_i^t}(x^t) \right\rangle + \gamma^2 \frac{1}{n} \sum_{i=1}^n \left\| \nabla f_{i,\mathbf{S}_i^t}(x^t) \right\|^2.$$

Conditioned on $x^t$, take expectation with respect to $\mathbf{S}_i^t$, $i \in [n]$ :

$$\mathbb{E}\left[\left\| r^{t+1} \right\|^2 | x^t\right] \leq \left\| r^t \right\|^2 - 2\gamma \left\langle r^t, \nabla f_{\mathcal{D}}(x^t) \right\rangle + \gamma^2 \frac{1}{n} \sum_{i=1}^n \mathbb{E}\left[\left\| \nabla f_{i,\mathbf{S}_i^t}(x^t) \right\|^2 | x^t\right].$$

From Lemma 4 we obtain that

$$\mathbb{E}\left[\left\| r^{t+1} \right\|^2 | x^t\right] \leq \left\| r^t \right\|^2 - 2\gamma \left\langle r^t, \nabla f_{\mathcal{D}}(x^t) \right\rangle$$

$$+ \gamma^2 \frac{2}{n} \sum_{i=1}^n L_{f_i} L_{\mathbf{S}_i}^{\max} \mathbb{E}\left[\left( f_i \left( s + \mathbf{S}_i^t(x^t - s) \right) - f_i^{\mathrm{inf}} \right)\right]$$

$$= \left\| r^t \right\|^2 - 2\gamma \left\langle r^t, \nabla f_{\mathcal{D}}(x^t) \right\rangle + \gamma^2 \frac{2}{n} \sum_{i=1}^n L_{f_i} L_{\mathbf{S}_i}^{\max} \left( f_{\mathcal{D}_i}(x^t) - f_i^{\mathrm{inf}} \right)$$

$$\leq \left\| r^t \right\|^2 - 2\gamma \left\langle r^t, \nabla f_{\mathcal{D}}(x^t) \right\rangle + \frac{2\gamma^2 \max_i \left\{ L_{f_i} L_{\mathbf{S}_i}^{\max} \right\}}{n} \sum_{i=1}^n \left( f_{\mathcal{D}_i}(x^t) - f_i^{\mathrm{inf}} \right)$$

$$= \left\| r^t \right\|^2 - 2\gamma \left\langle r^t, \nabla f_{\mathcal{D}}(x^t) \right\rangle + 2\gamma^2 \max_i \left\{ L_{f_i} L_{\mathbf{S}_i}^{\max} \right\} \left( f_{\mathcal{D}}(x^t) - f_{\mathcal{D}}^{\mathrm{inf}} \right)$$

$$+ 2\gamma^2 \max_i \left\{ L_{f_i} L_{\mathbf{S}_i}^{\max} \right\} \left( f_{\mathcal{D}}^{\mathrm{inf}} - \frac{1}{n} \sum_{i=1}^n f_i^{\mathrm{inf}} \right).$$

Since $f_{\mathcal{D}}$ is $\mu_{\mathcal{D}} \mu_f$-convex, we conclude that $\left\langle r^t, \nabla f_{\mathcal{D}}(x^t) \right\rangle \geq f_{\mathcal{D}}(x^t) - f_{\mathcal{D}}^{\mathrm{inf}} + \frac{\mu_{\mathcal{D}} \mu_f}{2} \left\| r^t \right\|^2$. Therefore,

$$\mathbb{E}\left[\left\| r^{t+1} \right\|^2 | x^t\right] \leq (1 - \gamma \mu_{\mathcal{D}} \mu_f) \left\| r^t \right\|^2 - 2\gamma \left( 1 - \gamma \max_i \left\{ L_{f_i} L_{\mathbf{S}_i}^{\max} \right\} \right) \left( f_{\mathcal{D}}(x^t) - f_{\mathcal{D}}^{\mathrm{inf}} \right)$$

$$+ 2\gamma^2 \max_i \left\{ L_{f_i} L_{\mathbf{S}_i}^{\max} \right\} \left( f_{\mathcal{D}}^{\mathrm{inf}} - \frac{1}{n} \sum_{i=1}^n f_i^{\mathrm{inf}} \right).$$

Since $\gamma \leq \frac{1}{\max_i \left\{ L_{f_i} L_{\mathbf{S}_i}^{\max} \right\}}$, taking expectation and using the tower property we get

$$\mathbb{E}\left[\left\| r^{t+1} \right\|^2\right] \leq (1 - \gamma \mu_{\mathcal{D}} \mu_f) \mathbb{E}\left[\left\| r^t \right\|^2\right] + 2\gamma^2 \max_i \left\{ L_{f_i} L_{\mathbf{S}_i}^{\max} \right\} \left( f_{\mathcal{D}}^{\mathrm{inf}} - \frac{1}{n} \sum_{i=1}^n f_i^{\mathrm{inf}} \right).$$

Unrolling the recurrence, we get

$$\mathbb{E}\left[\left\|r^t\right\|^2\right] \leq (1 - \gamma\mu_{\mathcal{D}}\mu_f)^t \left\|r^0\right\|^2 + \frac{2\gamma \max_i\left\{L_{f_i}L_{\mathbf{S}_i}^{\max}\right\}\left(f_{\mathcal{D}}^{\inf} - \frac{1}{n}\sum_{i=1}^n f_i^{\inf}\right)}{\mu_{\mathcal{D}}\mu_f}.$$

$\square$

**Corollary 9.** *Fix $\delta > 0$. Choose the stepsize $\gamma > 0$ as*

$$\gamma = \min\left\{\frac{1}{\max_i\left\{L_{f_i}L_{\mathbf{S}_i}^{\max}\right\}}, \frac{\mu_{\mathcal{D}}\mu_f\delta\left\|r^0\right\|^2}{2\max_i\left\{L_{f_i}L_{\mathbf{S}_i}^{\max}\right\}\left(f_{\mathcal{D}}^{\inf} - \frac{1}{n}\sum_{i=1}^n f_i^{\inf}\right)}\right\}.$$

*Then, provided that*

$$t \geq \frac{L_f L_{\mathbf{S}}^{\max}}{\mu_{\mathcal{D}}\mu_f}\left\{1, \frac{2\left(f_{\mathcal{D}}^{\inf} - f^{\inf}\right)}{\mu_{\mathcal{D}}\mu_f\delta\left\|r^0\right\|^2}\right\}\log\frac{1}{\delta},$$

*we have $\mathbb{E}\left[\left\|r^t\right\|^2\right] \leq 2\delta\left\|r^0\right\|^2$.*

*Proof.* Since $\gamma \leq \frac{\mu_{\mathcal{D}}\mu_f\delta\left\|r^0\right\|^2}{2\max_i\left\{L_{f_i}L_{\mathbf{S}_i}^{\max}\right\}\left(f_{\mathcal{D}}^{\inf} - \frac{1}{n}\sum_{i=1}^n f_i^{\inf}\right)}$, we have that

$$\frac{2\gamma \max_i\left\{L_{f_i}L_{\mathbf{S}_i}^{\max}\right\}\left(f_{\mathcal{D}}^{\inf} - \frac{1}{n}\sum_{i=1}^n f_i^{\inf}\right)}{\mu_{\mathcal{D}}\mu_f} \leq \delta\left\|r^0\right\|^2.$$

If $\gamma = \frac{\mu_{\mathcal{D}}\mu_f\delta\left\|r^0\right\|^2}{2\max_i\left\{L_{f_i}L_{\mathbf{S}_i}^{\max}\right\}\left(f_{\mathcal{D}}^{\inf} - \frac{1}{n}\sum_{i=1}^n f_i^{\inf}\right)}$, then, since

$$t \geq \frac{2\max_i\left\{L_{f_i}L_{\mathbf{S}_i}^{\max}\right\}\left(f_{\mathcal{D}}^{\inf} - \frac{1}{n}\sum_{i=1}^n f_i^{\inf}\right)}{\mu_{\mathcal{D}}^2\mu_f^2\delta\left\|r^0\right\|^2}\log\frac{1}{\delta},$$

we obtain that

$$(1 - \gamma\mu_{\mathcal{D}}\mu_f)^t \leq \exp\left(-\gamma\mu_{\mathcal{D}}\mu_f t\right) \leq \delta.$$

Further, if $\gamma = \frac{1}{\max_i\left\{L_{f_i}L_{\mathbf{S}_i}^{\max}\right\}}$, then, since

$$t \geq \frac{\max_i\left\{L_{f_i}L_{\mathbf{S}_i}^{\max}\right\}}{\mu_{\mathcal{D}}\mu_f}\log\frac{1}{\delta},$$

we obtain that

$$(1 - \gamma\mu_{\mathcal{D}}\mu_f)^t \leq \exp\left(-\gamma\mu_{\mathcal{D}}\mu_f t\right) \leq \delta.$$

Thus, we arrive at $\mathbb{E}\left[\left\|r^t\right\|^2\right] \leq 2\delta\left\|r^0\right\|^2$. $\square$

### F.3 CONVEX ANALYSIS

**Theorem 16.** *Assume that each $f_i$, $i \in [n]$, is differentiable, $L_{f_i}$-smooth and bounded from below by $f_i^{\inf}$, $f$ is convex. Let Assumptions 1 and 4 hold. Let $f$ be convex. Choose a stepsize $0 < \gamma \leq \frac{1}{2\max_i\left\{L_{f_i}L_{\mathbf{S}_i}^{\max}\right\}}$. Fix $T \geq 1$ and let $\bar{x}^T$ be chosen uniformly from the iterates $x^0, \ldots, x^{T-1}$. Then*

$$\mathbb{E}\left[f_{\mathcal{D}}(\bar{x}^T) - f_{\mathcal{D}}^{\inf}\right] \leq \frac{\left\|r^0\right\|^2}{\gamma T} + 2\gamma \max_i\left\{L_{f_i}L_{\mathbf{S}_i}^{\max}\right\},$$

*where $r^t \stackrel{def}{=} x^t - x_{\mathcal{D}}^\star$, $t = \{0, \ldots, T-1\}$.*

*Proof.* We get that

$$\left\|r^{t+1}\right\|^2 = \left\|\left(x^t - \frac{\gamma}{n}\sum_{i=1}^n \nabla f_{i,\mathbf{S}_i^t}(x^t)\right) - x_{\mathcal{D}}^\star\right\|^2 = \left\|x^t - x_{\mathcal{D}}^\star - \frac{\gamma}{n}\sum_{i=1}^n \nabla f_{i,\mathbf{S}_i^t}(x^t)\right\|^2$$

$$= \left\|r^t\right\|^2 - 2\gamma\left\langle r^t, \frac{1}{n}\sum_{i=1}^n \nabla f_{i,\mathbf{S}_i^t}(x^t)\right\rangle + \gamma^2\left\|\frac{1}{n}\sum_{i=1}^n \nabla f_{i,\mathbf{S}_i^t}(x^t)\right\|^2$$

$$\leq \left\|r^t\right\|^2 - 2\gamma\left\langle r^t, \frac{1}{n}\sum_{i=1}^n \nabla f_{i,\mathbf{S}_i^t}(x^t)\right\rangle + \gamma^2\frac{1}{n}\sum_{i=1}^n \left\|\nabla f_{i,\mathbf{S}_i^t}(x^t)\right\|^2.$$

Conditioned on $x^t$, take expectation with respect to $\mathbf{S}_i^t$, $i \in [n]$ :

$$\mathbb{E}\left[\left\|r^{t+1}\right\|^2 |x^t\right] \leq \left\|r^t\right\|^2 - 2\gamma\left\langle r^t, \nabla f_{\mathcal{D}}(x^t)\right\rangle + \gamma^2\frac{1}{n}\sum_{i=1}^n \mathbb{E}\left[\left\|\nabla f_{i,\mathbf{S}_i^t}(x^t)\right\|^2 |x^t\right].$$

From Lemma 4 and from convexity of $f_{\mathcal{D}}$ we obtain that

$$\mathbb{E}\left[\left\|r^{t+1}\right\|^2 |x^t\right] \leq \left\|r^t\right\|^2 - 2\gamma\left(f_{\mathcal{D}}(x^t) - f_{\mathcal{D}}^{\inf}\right)$$

$$+ \gamma^2\frac{2}{n}\sum_{i=1}^n L_{f_i}L_{\mathbf{S}_i}^{\max}\mathbb{E}\left[\left(f_i\left(s + \mathbf{S}_{i,t}(x^t - s)\right) - f_i^{\inf}\right)\right]$$

$$= \left\|r^t\right\|^2 - 2\gamma\left(f_{\mathcal{D}}(x^t) - f_{\mathcal{D}}^{\inf}\right) + \gamma^2\frac{2}{n}\sum_{i=1}^n L_{f_i}L_{\mathbf{S}_i}^{\max}\left(f_{\mathcal{D}_\rangle}(x^t) - f_i^{\inf}\right)$$

$$\leq \left\|r^t\right\|^2 - 2\gamma\left(f_{\mathcal{D}}(x^t) - f_{\mathcal{D}}^{\inf}\right) + \frac{2\gamma^2\max_i\left\{L_{f_i}L_{\mathbf{S}_i}^{\max}\right\}}{n}\sum_{i=1}^n\left(f_{\mathcal{D}_\rangle}(x^t) - f_i^{\inf}\right)$$

$$= \left\|r^t\right\|^2 - 2\gamma\left(f_{\mathcal{D}}(x^t) - f_{\mathcal{D}}^{\inf}\right) + 2\gamma^2\max_i\left\{L_{f_i}L_{\mathbf{S}_i}^{\max}\right\}\left(f_{\mathcal{D}}(x^t) - f_{\mathcal{D}}^{\inf}\right)$$

$$+ 2\gamma^2\max_i\left\{L_{f_i}L_{\mathbf{S}_i}^{\max}\right\}\left(f_{\mathcal{D}}^{\inf} - \frac{1}{n}\sum_{i=1}^n f_i^{\inf}\right)$$

$$= \left\|r^t\right\|^2 - 2\gamma\left(f_{\mathcal{D}}(x^t) - f_{\mathcal{D}}^{\inf}\right)\left(1 - \gamma\max_i\left\{L_{f_i}L_{\mathbf{S}_i}^{\max}\right\}\right)$$

$$+ 2\gamma^2\max_i\left\{L_{f_i}L_{\mathbf{S}_i}^{\max}\right\}\left(f_{\mathcal{D}}^{\inf} - \frac{1}{n}\sum_{i=1}^n f_i^{\inf}\right).$$

Rearranging and taking expectation, taking into account the condition on the stepsize, we have

$$\gamma\left(f_{\mathcal{D}}(x^t) - f_{\mathcal{D}}^{\inf}\right) \leq \mathbb{E}\left[\left\|r^t\right\|^2\right] - \mathbb{E}\left[\left\|r^{t+1}\right\|^2\right] + 2\gamma^2\max_i\left\{L_{f_i}L_{\mathbf{S}_i}^{\max}\right\}\left(f_{\mathcal{D}}^{\inf} - \frac{1}{n}\sum_{i=1}^n f_i^{\inf}\right).$$

Summing over $t = 0, \ldots, T-1$ and using telescopic cancellation gives

$$\gamma\sum_{t=0}^{T-1}\left(\mathbb{E}\left[f_{\mathcal{D}}(x^t) - f_{\mathcal{D}}^{\inf}\right]\right) \leq \left\|r^0\right\|^2 - \mathbb{E}\left[\left\|r^T\right\|^2\right]$$

$$+ 2T\gamma^2\max_i\left\{L_{f_i}L_{\mathbf{S}_i}^{\max}\right\}\left(f_{\mathcal{D}}^{\inf} - \frac{1}{n}\sum_{i=1}^n f_i^{\inf}\right).$$

Since $\mathbb{E}\left[\left\|r^T\right\|^2\right] \geq 0$, dividing both sides by $\gamma T$ gives:

$$\frac{1}{T}\sum_{t=0}^{T-1}\left(\mathbb{E}\left[f_{\mathcal{D}}(x^t) - f_{\mathcal{D}}^{\inf}\right]\right) \leq \frac{\left\|r^0\right\|^2}{\gamma T} + 2\gamma\max_i\left\{L_{f_i}L_{\mathbf{S}_i}^{\max}\right\}\left(f_{\mathcal{D}}^{\inf} - \frac{1}{n}\sum_{i=1}^n f_i^{\inf}\right).$$

We treat the $\left(\frac{1}{T}, \ldots, \frac{1}{T}\right)$ as if it is a probability vector. Indeed, using that $f_{\mathcal{D}}$ is convex together with Jensen's inequality gives

$$\mathbb{E}\left[f_{\mathcal{D}}(\bar{x}^T) - f_{\mathcal{D}}^{\mathrm{inf}}\right] \leq \frac{\left\|r^0\right\|^2}{\gamma T} + 2\gamma \max_i \left\{L_{f_i} L_{\mathbf{S}_i}^{\mathrm{max}}\right\} \left(f_{\mathcal{D}}^{\mathrm{inf}} - \frac{1}{n}\sum_{i=1}^n f_i^{\mathrm{inf}}\right).$$

$\square$

**Corollary 10.** *Fix $\delta > 0$. Choose the stepsize $\gamma > 0$ as*

$$\gamma = \frac{1}{2\max_i\left\{L_{f_i} L_{\mathbf{S}_i}^{\mathrm{max}}\right\}} \min\left\{1, \frac{\delta\left\|r^0\right\|^2}{f_{\mathcal{D}}^{\mathrm{inf}} - \frac{1}{n}\sum_{i=1}^n f_i^{\mathrm{inf}}}\right\}.$$

*Then, provided that*

$$T \geq \frac{2L_f \lambda_m^{\mathbf{S}}}{\delta} \max\left\{1, \frac{f_{\mathcal{D}}^{\mathrm{inf}} - f^{\mathrm{inf}}}{\delta\left\|r^0\right\|^2}\right\},$$

*we have $\mathbb{E}\left[f_{\mathcal{D}}(\bar{x}^t) - f_{\mathcal{D}}^{\mathrm{inf}}\right] \leq 2\delta\left\|r^0\right\|^2$.*

*Proof.* Since $\gamma \leq \frac{\delta\left\|r^0\right\|^2}{2\max_i\left\{L_{f_i} L_{\mathbf{S}_i}^{\mathrm{max}}\right\}\left(f_{\mathcal{D}}^{\mathrm{inf}} - \frac{1}{n}\sum_{i=1}^n f_i^{\mathrm{inf}}\right)}$, we have that

$$2\gamma \max_i\left\{L_{f_i} L_{\mathbf{S}_i}^{\mathrm{max}}\right\}\left(f_{\mathcal{D}}^{\mathrm{inf}} - \frac{1}{n}\sum_{i=1}^n f_i^{\mathrm{inf}}\right) \leq \delta\left\|r^0\right\|^2.$$

If $\gamma = \frac{\delta\left\|r^0\right\|^2}{2\max_i\left\{L_{f_i} L_{\mathbf{S}_i}^{\mathrm{max}}\right\}\left(f_{\mathcal{D}}^{\mathrm{inf}} - \frac{1}{n}\sum_{i=1}^n f_i^{\mathrm{inf}}\right)}$, then, since

$$T \geq \frac{2\max_i\left\{L_{f_i} L_{\mathbf{S}_i}^{\mathrm{max}}\right\}\left(f_{\mathcal{D}}^{\mathrm{inf}} - \frac{1}{n}\sum_{i=1}^n f_i^{\mathrm{inf}}\right)}{\delta^2\left\|r^0\right\|^2},$$

we obtain that $\frac{\left\|r^0\right\|^2}{\gamma T} \leq \delta\left\|r^0\right\|^2$. Further, if $\gamma = \frac{1}{2\max_i\left\{L_{f_i} L_{\mathbf{S}_i}^{\mathrm{max}}\right\}}$, then, since $T \geq \frac{2\max_i\left\{L_{f_i} L_{\mathbf{S}_i}^{\mathrm{max}}\right\}}{\delta}$,

we have $\frac{\left\|r^0\right\|^2}{\gamma T} \leq \delta\left\|r^0\right\|^2$. Thus, we arrive at $\mathbb{E}\left[f_{\mathcal{D}}(\bar{x}^t) - f_{\mathcal{D}}^{\mathrm{inf}}\right] \leq 2\delta\left\|r^0\right\|^2$. $\square$

---

**Algorithm 3** Loopless Stochastic Variance Reduced Double Sketched Gradient (L-SVRDSG)

1: **Parameters:** learning rate $\gamma > 0$; probability $p$; sketches $\mathbf{S}_1, \ldots, \mathbf{S}_N$; initial model and shift $x^0, v \in \mathbb{R}^d$, sketch minibatch size $b$; initial sketch minibatch $\mathcal{B}^0 \subset [N]$.
2: **Initialization:** $w^0 = x^0$, $\hat{h}^0 = \frac{1}{b} \sum\limits_{i \in \mathcal{B}^0} \nabla f_{\mathbf{S}_i}(w^0)$.
3: **for** $t = 0, 1, 2 \ldots$ **do**
4:     Sample a sketch: $\mathbf{S}^t$ from $\{\mathbf{S}_1, \ldots, \mathbf{S}_N\}$
5:     Form a gradient estimator: $h^t = \nabla f_{\mathbf{S}^t}(x^t) - \nabla f_{\mathbf{S}^t}(w^t) + \hat{h}^t$.
6:     Perform a gradient-type step: $x^{t+1} = x^t - \gamma h^t$
7:     Sample a Bernoulli random variable $\beta_p$
8:     **if** $\beta_p = 1$ **then**
9:         Sample $\mathcal{B}^t$ uniformly without replacement
10:        $w^{t+1} = x^t, \quad \hat{h}^{t+1} = \frac{1}{b} \sum\limits_{i \in \mathcal{B}^t} \nabla f_{\mathbf{S}_i^t}(x^t)$
11:    **else**
12:        $w^{t+1} = w^t, \quad \hat{h}^{t+1} = \hat{h}^t$
13:    **end if**
14: **end for**

---

## G  VARIANCE REDUCTION

In Theorem 2, we established linear convergence toward a neighborhood of the solution $x_\mathcal{D}^\star$ for Algorithm 1 (I). To reach the exact solution, the stepsize must decrease to zero, resulting in slower sublinear convergence. The neighborhood's size is linked to the variance of gradient estimator at the solution. Various Variance Reduction (VR) techniques have been proposed to address this issue (Gower et al., 2020). Consider the case when distribution $\mathcal{D}$ is uniform and has finite support, i.e., $\{\mathbf{S}_1, \ldots, \mathbf{S}_N\}$ leading to a finite-sum modification of the MAST problem (2)

$$f_\mathcal{D}(x) = \frac{1}{N} \sum_{i=1}^{N} \mathbb{E}\left[f(v + \mathbf{S}_i(x - v))\right]. \tag{39}$$

In this situation, VR-methods can eliminate the neighborhood enabling linear convergence to the solution. We utilize the L-SVRG (Kovalev et al., 2020; Hofmann et al., 2015) approach, which requires computing the full gradient with probability $p$. For our formulation, calculating $\nabla f_{\mathbf{S}}$ for all possible $\mathbf{S}$ is rarely feasible. For instance, for Rand-$K$, there are $N = d!/(K!(d-K)!)$ possible operators $\mathbf{S}_i$. Therefore, in our Algorithm 3, we employ a sketch *minibatch* estimator $\hat{h}^t$ computed for a subset $\mathcal{B} \subset [N]$ (sampled uniformly without replacement) of sketches instead of the full gradient. Finally, we present the convergence results for the strongly convex case.

**Theorem 17.** *Assume that $f$ is $L_f$-smooth (2), $\mu_f$-strongly convex (3), and $\mathbf{S}$ is sampled from finite set $\{\mathbf{S}_1, \ldots, \mathbf{S}_N\}$. Then, for stepsize $\gamma \leq 1/(20 L_f L_{\mathbf{S}}^{\max})$ and sketch minibatch size $b \in (0, N]$, the iterates of Algorithm 3 satisfy*

$$\mathbb{E}\left[\Psi^T\right] \leq (1 - \rho)^T \Psi^0 + \frac{8\gamma^2 L_f L_{\mathbf{S}}^{\max}(N - b)}{\rho \max\{1, N - 1\} b} \left(f_\mathcal{D}^{\inf} - f^{\inf}\right),$$

*where $\rho \stackrel{def}{=} \max\{\gamma \mu_\mathcal{D} \mu_f, p/2\}$ and Lyapunov function*

$$\Psi^t \stackrel{def}{=} \|x^t - x_\mathcal{D}^\star\|^2 + \frac{16\gamma^2}{pN} \sum_{i=1}^{N} \left\|\nabla f_{\mathbf{S}_i}(w^t) - \nabla f_{\mathbf{S}_i}(x_\mathcal{D}^\star)\right\|^2.$$

Note that the achieved result demonstrates linear convergence towards the solution's neighborhood. However, this neighborhood is roughly reduced by a factor of $1/b$ compared to Theorem 2, and it scales with $N - b$. Thus, when employing a full gradient for $\hat{h}^t$ with $b = N$, the neighborhood shrinks to zero, resulting in a linear convergence rate to the exact solution.

### G.1  L-SVRDSG: STRONGLY CONVEX ANALYSIS

The proof of Theorem 17 relies on the following Lemma:

**Lemma 15.** *Let Assumptions 1 and 3 hold. Then the following inequality holds:*

$$\mathbb{E}\left[\left\|x^{t+1} - x_{\mathcal{D}}^\star\right\|^2\right] = (1 - \gamma\mu_{\mathcal{D}}\mu_f)\left\|x^t - x_{\mathcal{D}}^\star\right\|^2 - 2\gamma\left(f_{\mathcal{D}}(x^t) - f(x_{\mathcal{D}}^\star)\right) + \gamma^2\mathbb{E}\left[\left\|h^t\right\|^2\right].$$

*Proof.* We start from expanding squared norm:

$$\begin{aligned}
\mathbb{E}\left[\left\|x^{t+1} - x_{\mathcal{D}}^\star\right\|^2\right] &= \mathbb{E}\left[\left\|x^t - \gamma h^t - x_{\mathcal{D}}^\star\right\|^2\right] \\
&= \mathbb{E}\left[\left\|x^t - x_{\mathcal{D}}^\star\right\|^2\right] - 2\gamma\langle h^t, x^t - x_{\mathcal{D}}^\star\rangle + \gamma^2\mathbb{E}\left[\left\|h^t\right\|^2\right] \\
&= \mathbb{E}\left[\left\|x^t - x_{\mathcal{D}}^\star\right\|^2\right] - 2\gamma\langle\nabla f_{\mathcal{D}}(x^t), x^t - x_{\mathcal{D}}^\star\rangle + \gamma^2\mathbb{E}\left[\left\|h^t\right\|^2\right] \\
&= (1 - \gamma\mu\mu_f)\left\|x^t - x_{\mathcal{D}}^\star\right\|^2 - 2\gamma\left(f_{\mathcal{D}}(x^t) - f(x_{\mathcal{D}}^\star)\right) + \gamma^2\mathbb{E}\left[\left\|h^t\right\|^2\right].
\end{aligned}$$

$\square$

**Lemma 16.** *Let Assumptions 1 and 2 hold. Then the following inequality holds:*

$$\begin{aligned}
\mathbb{E}\left[\left\|h^t\right\|^2\right] &\leq 8L_f L_{\mathbf{S}^t}^{\max}\left(f_{\mathcal{D}}(x^t) - f_{\mathcal{D}}(x_{\mathcal{D}}^\star)\right) + 8\frac{1}{N}\sum_{i=1}^N\left\|\nabla f_{\mathbf{S}_i^t}(w^t) - \nabla f_{\mathbf{S}_i^t}(x_{\mathcal{D}}^\star)\right\|^2 \\
&\quad + \frac{8(N-b)}{(N-1)b}L_f L_{\mathbf{S}^t}^{\max}\left(f_{\mathcal{D}}^{\inf} - f^{\inf}\right).
\end{aligned}$$

*Proof.* We start the proof from the definition of $h^t$:

$$\begin{aligned}
\mathbb{E}\left[\left\|h^t\right\|^2\right] &= \mathbb{E}\left[\left\|\nabla f_{\mathbf{S}^t}(x^t) - \nabla f_{\mathbf{S}^t}(w^t) + \frac{1}{b}\sum_{i\in\mathcal{B}^t}\nabla f_{\mathbf{S}_i^t}(w^t)\right\|^2\right] \\
&= \mathbb{E}\left[\left\|\nabla f_{\mathbf{S}^t}(x^t) - \nabla f_{\mathbf{S}^t}(w^t) + \frac{1}{b}\sum_{i\in\mathcal{B}^t}\nabla f_{\mathbf{S}_i^t}(w^t) - \nabla f_{\mathbf{S}^t}(x_{\mathcal{D}}^\star) + \nabla f_{\mathbf{S}^t}(x_{\mathcal{D}}^\star)\right.\right. \\
&\qquad\qquad \left.\left. - \frac{1}{b}\sum_{i\in\mathcal{B}^t}\nabla f_{\mathbf{S}_i^t}(x_{\mathcal{D}}^\star) + \frac{1}{b}\sum_{i\in\mathcal{B}^t}\nabla f_{\mathbf{S}_i^t}(x_{\mathcal{D}}^\star)\right\|^2\right] \\
&\leq 4\mathbb{E}\left[\left\|\nabla f_{\mathbf{S}^t}(x^t) - \nabla f_{\mathbf{S}^t}(x_{\mathcal{D}}^\star)\right\|^2\right] + 4\mathbb{E}\left[\left\|\nabla f_{\mathbf{S}^t}(w^t) - \nabla f_{\mathbf{S}^t}(x_{\mathcal{D}}^\star)\right\|^2\right] \\
&\quad + 4\mathbb{E}\left[\left\|\frac{1}{b}\sum_{i\in\mathcal{B}^t}\left(\nabla f_{\mathbf{S}_i^t}(w^t) - \nabla f_{\mathbf{S}_i^t}(x_{\mathcal{D}}^\star)\right)\right\|^2\right] + 4\mathbb{E}\left[\left\|\frac{1}{b}\sum_{i\in\mathcal{B}^t}\nabla f_{\mathbf{S}_i^t}(x_{\mathcal{D}}^\star)\right\|^2\right] \\
&\stackrel{(24)}{\leq} 8L_f L_{\mathbf{S}^t}^{\max}\left(f_{\mathcal{D}}(x^t) - f_{\mathcal{D}}(x_{\mathcal{D}}^\star)\right) + 8\frac{1}{N}\sum_{i=1}^N\left\|\nabla f_{\mathbf{S}_i^t}(w^t) - \nabla f_{\mathbf{S}_i^t}(x_{\mathcal{D}}^\star)\right\|^2 \\
&\quad + 4\mathbb{E}\left[\left\|\frac{1}{b}\sum_{i\in\mathcal{B}^t}\nabla f_{\mathbf{S}_i^t}(x_{\mathcal{D}}^\star)\right\|^2\right].
\end{aligned}$$

Using Lemma 5 we obtain

$$\begin{aligned}
\mathbb{E}\left[\left\|h^t\right\|^2\right] &\leq 8L_f L_{\mathbf{S}^t}^{\max}\left(f_{\mathcal{D}}(x^t) - f_{\mathcal{D}}(x_{\mathcal{D}}^\star)\right) + 8\frac{1}{N}\sum_{i=1}^N\left\|\nabla f_{\mathbf{S}_i^t}(w^t) - \nabla f_{\mathbf{S}_i^t}(x_{\mathcal{D}}^\star)\right\|^2 \\
&\quad + 4\frac{N-b}{\max\{1, N-1\}b}\frac{1}{N}\sum_{i\in\mathcal{B}^t}\left\|\nabla f_{\mathbf{S}_i^t}(x_{\mathcal{D}}^\star)\right\|^2 \\
&\leq 8L_f L_{\mathbf{S}^t}^{\max}\left(f_{\mathcal{D}}(x^t) - f_{\mathcal{D}}(x_{\mathcal{D}}^\star)\right) + 8\frac{1}{N}\sum_{i=1}^N\left\|\nabla f_{\mathbf{S}_i^t}(w^t) - \nabla f_{\mathbf{S}_i^t}(x_{\mathcal{D}}^\star)\right\|^2 \\
&\quad + \frac{8(N-b)}{\max\{1, N-1\}b}L_f L_{\mathbf{S}^t}^{\max}\left(f_{\mathcal{D}}^{\inf} - f^{\inf}\right).
\end{aligned}$$

$\square$

**Lemma 17.** *Let Assumptions 1 and 2 hold. Let $D^t = \frac{16\gamma^2}{pN} \sum_{i=1}^{N} \left\| \nabla f_{\mathbf{S}_i^t}(x_{\mathcal{D}}^\star) - \nabla f_{\mathbf{S}_i^t}(w^t) \right\|^2$. Then the following inequality holds:*

$$\mathbb{E}\left[D^{t+1}\right] \leq (1-p)D^t + 32\gamma^2 L_f L_{\mathbf{S}^t}^{\max}\left(f_{\mathcal{D}}(w^t) - f_{\mathcal{D}}(x_{\mathcal{D}}^\star)\right).$$

*Proof.* Using the smoothness property, we obtain

$$\mathbb{E}\left[D^{t+1}\right] = (1-p)D^t + p\frac{16\gamma^2}{pN}\sum_{i=1}^{N}\mathbb{E}\left[\left\|\nabla f_{\mathbf{S}_i^t}(x_{\mathcal{D}}^\star) - \nabla f_{\mathbf{S}_i^t}(w^t)\right\|^2\right]$$

$$\leq (1-p)D^t + 32\gamma^2 L_f L_{\mathbf{S}^t}^{\max}\left(f_{\mathcal{D}}(w^t) - f_{\mathcal{D}}(x_{\mathcal{D}}^\star)\right).$$

$\square$

**Theorem 18.** *Assume that $f$ is $L_f$-smooth (2), $\mu_f$-strongly convex (3), and $\mathbf{S}$ is sampled from finite set $\{\mathbf{S}_1, \ldots, \mathbf{S}_N\}$. Then, for stepsize $\gamma \leq 1/(20L_f L_{\mathbf{S}}^{\max})$ and sketch minibatch size $b \in (0, N]$, the iterates of Algorithm 3 satisfy*

$$\mathbb{E}\left[\Psi^T\right] \leq (1-\rho)^T \Psi^0 + \frac{8\gamma^2 L_f L_{\mathbf{S}}^{\max}(N-b)}{\rho\max\{1, N-1\}b}\left(f_{\mathcal{D}}^{\inf} - f^{\inf}\right),$$

*where $\rho \overset{def}{=} \max\{\gamma\mu_{\mathcal{D}}\mu_f, p/2\}$ and Lyapunov function $\Psi^t \overset{def}{=} \|x^t - x_{\mathcal{D}}^\star\|^2 + \frac{16\gamma^2}{pN}\sum_{i=1}^{N}\|\nabla f_{\mathbf{S}_i}(w^t) - \nabla f_{\mathbf{S}_i}(x_{\mathcal{D}}^\star)\|^2$.*

*Proof.* We combine three previous lemmas 15, 16, 17:

$$\mathbb{E}\left[\|x^{t+1} - x_{\mathcal{D}}^\star\|^2 + D^{t+1}\right] \leq (1-\gamma\mu_{\mathcal{D}}\mu_f)\|x^t - x_{\mathcal{D}}^\star\|^2 - 2\gamma\left(f_{\mathcal{D}}(x^t) - f(x_{\mathcal{D}}^\star)\right) + \gamma^2\mathbb{E}\left[\|g^t\|^2\right]$$

$$+ (1-p)D^t + 32\gamma^2 L_f L_{\mathbf{S}}^{\max}\left(f_{\mathcal{D}}(w^t) - f_{\mathcal{D}}(x_{\mathcal{D}}^\star)\right)$$

$$\leq (1-\gamma\mu_{\mathcal{D}}\mu_f)\|x^t - x_{\mathcal{D}}^\star\|^2 - 2\gamma\left(f_{\mathcal{D}}(x^t) - f(x_{\mathcal{D}}^\star)\right) + (1-p)D^t$$

$$+ 32\gamma^2 L_f L_{\mathbf{S}}^{\max}\left(f_{\mathcal{D}}(w^t) - f_{\mathcal{D}}(x_{\mathcal{D}}^\star)\right)$$

$$+ \gamma^2\left(8L_f L_{\mathbf{S}}^{\max}\left(f_{\mathcal{D}}(w^t) - f_{\mathcal{D}}(x_{\mathcal{D}}^\star)\right) + \frac{p}{2\gamma^2}D^t\right.$$

$$\left. + \frac{8(N-b)}{\max\{1, N-1\}b}L_f L_{\mathbf{S}^t}^{\max}\left(f_{\mathcal{D}}^{\inf} - f^{\inf}\right)\right)$$

$$= (1-\gamma\mu_{\mathcal{D}}\mu_f)\|x^t - x_{\mathcal{D}}^\star\|^2$$

$$- 2\gamma\left(f_{\mathcal{D}}(x^t) - f(x_{\mathcal{D}}^\star)\right)(1 - 20\gamma L_f L_{\mathbf{S}}^{\max})$$

$$+ \left(1 - \frac{p}{2}\right)D^t + \frac{8(N-b)}{\max\{1, N-1\}b}\gamma^2 L_f L_{\mathbf{S}}^{\max}\left(f_{\mathcal{D}}^{\inf} - f^{\inf}\right).$$

Since $\gamma \leq \frac{1}{20L_f L_{\mathbf{S}}^{\max}}$, we get that

$$\Psi^{t+1} \leq \left(1 - \max\left\{\gamma\mu_{\mathcal{D}}\mu_f, \frac{p}{2}\right\}\right)\Psi^t + \frac{8(N-b)}{\max\{1, N-1\}b}\gamma^2 L_f L_{\mathbf{S}}^{\max}\left(f_{\mathcal{D}}^{\inf} - f^{\inf}\right).$$

Unrolling the recursion and using $\rho = \max\left\{\gamma\mu_{\mathcal{D}}\mu_f, \frac{p}{2}\right\}$, we obtain

$$\mathbb{E}\left[\Psi^T\right] \leq (1-\rho)^T \Psi^0 + \frac{8\gamma^2 L_f L_{\mathbf{S}}^{\max}(N-b)}{\rho\max\{1, N-1\}b}\left(f_{\mathcal{D}}^{\inf} - f^{\inf}\right),$$

$\square$

### G.1.1 CONVEX ANALYSIS

Now we formulate and prove theorem for the general (non-strongly) convex regime:

**Theorem 19.** *Assume that $f$ is $L_f$-smooth (2), convex, and $\mathbf{S}$ is sampled from finite set $\{\mathbf{S}_1, \ldots, \mathbf{S}_N\}$. Then, for stepsize $\gamma \leq 1/(40 L_f L_{\mathbf{S}}^{\max})$ and sketch minibatch size $b \in (0, N]$ the iterates of Algorithm 3 satisfy*

$$\mathbb{E}\left[f_{\mathcal{D}}(\bar{x}^t)\right] - f(x_{\mathcal{D}}^\star) \leq \frac{\Psi^0}{\gamma T} + \frac{8(N - b)}{\max\{1, N - 1\}b} \gamma L_f L_{\mathbf{S}}^{\max} \left(f_{\mathcal{D}}^{\inf} - f^{\inf}\right).$$

*where $\bar{x}^T = \frac{1}{T} \sum_{t=0}^{T-1} x^t$ and Lyapunov function $\Psi^t \overset{def}{=} \|x^t - x_{\mathcal{D}}^\star\|^2 + \frac{16\gamma^2}{pN} \sum_{i=1}^N \|\nabla f_{\mathbf{S}_i}(w^t) - \nabla f_{\mathbf{S}_i}(x_{\mathcal{D}}^\star)\|^2$.*

*Proof.* We start from the recursion in Theorem 18:

$$\mathbb{E}\left[\|x^{t+1} - x_{\mathcal{D}}^\star\|^2 + D^{t+1}\right] \leq (1 - \gamma \mu_{\mathcal{D}} \mu_f) \|x^t - x_{\mathcal{D}}^\star\|^2$$
$$- 2\gamma \left(f_{\mathcal{D}}(x^t) - f(x_{\mathcal{D}}^\star)\right)(1 - 20\gamma L_f L_{\mathbf{S}}^{\max})$$
$$+ \left(1 - \frac{p}{2}\right) D^t + \frac{8(N - b)}{\max\{1, N - 1\}b} \gamma^2 L_f L_{\mathbf{S}}^{\max} \left(f_{\mathcal{D}}^{\inf} - f^{\inf}\right).$$

Using $\mu_f = 0$ and $\left(1 - \frac{p}{2}\right) \leq 1$ we have

$$\mathbb{E}\left[\|x^{t+1} - x_{\mathcal{D}}^\star\|^2 + D^{t+1}\right] \leq \|x^t - x_{\mathcal{D}}^\star\|^2 - 2\gamma \left(f_{\mathcal{D}}(x^t) - f(x_{\mathcal{D}}^\star)\right)(1 - 20\gamma L_f L_{\mathbf{S}}^{\max})$$
$$+ D^t + \frac{8(N - b)}{\max\{1, N - 1\}b} \gamma^2 L_f L_{\mathbf{S}}^{\max} \left(f_{\mathcal{D}}^{\inf} - f^{\inf}\right).$$

Since $\gamma \leq \frac{1}{40 L_f L_{\mathbf{S}}^{\max}}$, we have $(1 - 20\gamma L_f L_{\mathbf{S}}^{\max}) \geq \frac{1}{2}$ and it leads to

$$\mathbb{E}\left[\|x^{t+1} - x_{\mathcal{D}}^\star\|^2 + D^{t+1}\right] \leq \|x^t - x_{\mathcal{D}}^\star\|^2 - \gamma \left(f_{\mathcal{D}}(x^t) - f(x_{\mathcal{D}}^\star)\right)$$
$$+ D^t + \frac{8(N - b)}{\max\{1, N - 1\}b} \gamma^2 L_f L_{\mathbf{S}}^{\max} \left(f_{\mathcal{D}}^{\inf} - f^{\inf}\right).$$

Using the tower property, we have

$$\mathbb{E}\left[\Psi^{t+1}\right] \leq \mathbb{E}\left[\Psi^t\right] - \gamma \mathbb{E}\left[\left(f_{\mathcal{D}}(x^t) - f(x_{\mathcal{D}}^\star)\right)\right]$$
$$+ \frac{8(N - b)}{\max\{1, N - 1\}b} \gamma^2 L_f L_{\mathbf{S}}^{\max} \left(f_{\mathcal{D}}^{\inf} - f^{\inf}\right)$$
$$\gamma \mathbb{E}\left[\left(f_{\mathcal{D}}(x^t) - f(x_{\mathcal{D}}^\star)\right)\right] \leq \mathbb{E}\left[\Psi^t\right] - \mathbb{E}\left[\Psi^{t+1}\right]$$
$$+ \frac{8(N - b)}{\max\{1, N - 1\}b} \gamma^2 L_f L_{\mathbf{S}}^{\max} \left(f_{\mathcal{D}}^{\inf} - f^{\inf}\right)$$
$$\gamma \frac{1}{T} \sum_{t=0}^{T-1} \mathbb{E}\left[\left(f_{\mathcal{D}}(x^t) - f(x_{\mathcal{D}}^\star)\right)\right] \leq \frac{1}{T} \sum_{t=0}^{T-1} \left(\mathbb{E}\left[\Psi^t\right] - \mathbb{E}\left[\Psi^{t+1}\right]\right)$$
$$+ \frac{8(N - b)}{\max\{1, N - 1\}b} \gamma^2 L_f L_{\mathbf{S}}^{\max} \left(f_{\mathcal{D}}^{\inf} - f^{\inf}\right)$$
$$\mathbb{E}\left[f_{\mathcal{D}}(\bar{x}^T)\right] - f(x_{\mathcal{D}}^\star) \leq \frac{\Psi^0}{\gamma T} + \frac{8(N - b)}{\max\{1, N - 1\}b} \gamma L_f L_{\mathbf{S}}^{\max} \left(f_{\mathcal{D}}^{\inf} - f^{\inf}\right).$$

$\square$

---

**Algorithm 4** Sketched ProbAbilistic Gradient Estimator (S-PAGE)

---

1: **Parameters:** learning rate $\gamma > 0$; probability $p$; sketches $\mathbf{S}_1, \dots, \mathbf{S}_N$; initial model and shift $x^0, v \in \mathbb{R}^d$, sketch minibatch sizes $b$ and $b' < b$; initial sketch minibatch $\mathcal{B}^0 \subset [N]$.
2: **Initialization:** $h^0 = \frac{1}{b} \sum\limits_{i \in \mathcal{B}^0} \nabla f_{\mathbf{S}_i}(x^0)$.

3: **for** $t = 0, 1, 2 \dots$ **do**
4:     Perform a gradient-type step: $x^{t+1} = x^t - \gamma h^t$
5:     Sample a Bernoulli random variable $\beta_p$
6:     **if** $\beta_p = 1$ **then**
7:         Sample minibatch $\mathcal{B}^t$ with size $b$ uniformly without replacement
8:         Form a gradient estimator: $h^{t+1} = \frac{1}{b} \sum\limits_{i \in \mathcal{B}^t} \nabla f_{\mathbf{S}_i^t}(x^{t+1})$

9:     **else**
10:       Sample minibatch $(\mathcal{B}^t)'$ with size $b'$ uniformly without replacement

11:       Form a gradient estimator: $h^{t+1} = h^t + \frac{1}{b'} \sum\limits_{i \in (\mathcal{B}^t)'} \left( \nabla f_{\mathbf{S}_i^t}(x^{t+1}) - \nabla f_{\mathbf{S}_i^t}(x^{t+1}) \right)$

12:     **end if**
13: **end for**

---

## G.2   S-PAGE: NONCONVEX ANALYSIS

In this section, we introduce a variant of the Probabilistic Gradient Estimator (PAGE) algorithm applied to the MAST formulation as defined in Equation 2 for non-convex setting. Li et al. (Li et al., 2021) showed that this method is optimal in the non-convex regime. We refer to this method as the Sketched Probabilistic Gradient Estimator (S-PAGE). Calculating the full gradient is not efficient, as the number of possible sketches when considering `Rand-K` is given by $\frac{n!}{(n-k)!k!}$. Consequently, we employ a minibatch estimator to achieve partial variance reduction, using a large minibatch size where $b > b'$. For the purpose of analysis, we assume that the variance of the sketch gradient is bounded.

**Assumption 5** (Bounded sketch variance). *The sketched gradient has bounded variance if exists $\sigma_{\mathcal{D}} > 0$, such that*

$$\mathbb{E}\left[ \left\| \nabla f_{\mathbf{S}}(x) - \nabla f_{\mathcal{D}}(x) \right\|^2 \right] \leq \sigma_{\mathcal{D}}^2 \quad \forall x \in \mathbb{R}^d.$$

**Lemma 18** (Lemma 2 from (Li et al., 2021)). *Suppose that function $f$ is $L$-smooth and let $x^{t+1} := x^t - \gamma g^t$. Then for any $g^t \in \mathbb{R}^d$ and $\gamma > 0$, we have*

$$f\left(x^{t+1}\right) \leq f\left(x^t\right) - \frac{\gamma}{2} \left\| \nabla f(x^t) \right\|^2 - \left( \frac{1}{2\gamma} - \frac{L}{2} \right) \left\| x^{t+1} - x^t \right\|^2 + \frac{\gamma}{2} \left\| h^t - \nabla f(x^t) \right\|^2 \quad (40)$$

**Lemma 19.** *Suppose that function $f$ is $L_f$-smooth and $\mathbf{S}$ satisfies Assumption 1 and let $x^{t+1} = x^t - \gamma h^t$. Then for any $h^t \in \mathbb{R}^d$ and $\gamma > 0$, we have*

$$f_{\mathcal{D}}\left(x^{t+1}\right) \leq f_{\mathcal{D}}\left(x^t\right) - \frac{\gamma}{2} \left\| \nabla f_{\mathcal{D}}(x^t) \right\|^2 - \left( \frac{1}{2\gamma} - \frac{L_f L_{\mathcal{D}}}{2} \right) \left\| x^{t+1} - x^t \right\|^2 + \frac{\gamma}{2} \left\| h^t - \nabla f_{\mathcal{D}}(x^t) \right\|^2. \quad (41)$$

*Proof.* Since $f$ is $L_f$-smooth and $\mathbf{S}$ satisfies Assumption 1 the function $f_{\mathcal{D}}$ is $L_{f_{\mathcal{D}}}$. Then using Lemma 19, we obtain

$$f_{\mathcal{D}}\left(x^{t+1}\right) \leq f_{\mathcal{D}}\left(x^t\right) - \frac{\gamma}{2} \left\| \nabla f_{\mathcal{D}}(x^t) \right\|^2 - \left( \frac{1}{2\gamma} - \frac{L_{f_{\mathcal{D}}}}{2} \right) \left\| x^{t+1} - x^t \right\|^2 + \frac{\gamma}{2} \left\| h^t - \nabla f_{\mathcal{D}}(x^t) \right\|^2.$$

Using Lemma 1 we have $L_{f_{\mathcal{D}}} \leq L_{\mathcal{D}} L_f$. Plugging this into the inequality, we obtain

$$f_{\mathcal{D}}\left(x^{t+1}\right) \leq f_{\mathcal{D}}\left(x^t\right) - \frac{\gamma}{2} \left\| \nabla f_{\mathcal{D}}(x^t) \right\|^2 - \left( \frac{1}{2\gamma} - \frac{L_f L_{\mathcal{D}}}{2} \right) \left\| x^{t+1} - x^t \right\|^2 + \frac{\gamma}{2} \left\| h^t - \nabla f_{\mathcal{D}}(x^t) \right\|^2. \quad (42)$$

$\square$

**Lemma 20.** *Suppose that Assumptions 1, 2 and 5 hold. If the gradient estimator $h^{t+1}$ is defined according to Algorithm 4, then we have*

$$\mathbb{E}\left[\left\|h^{t+1} - \nabla f_{\mathcal{D}}(x^{t+1})\right\|^2\right] \leq (1-p)\left\|h^t - \nabla f_{\mathcal{D}}(x^t)\right\|^2$$
$$+ \frac{N-b'}{(N-1)b'}(1-p)L_f^2 L_{\mathbf{S}}^{\max}L_{\mathcal{D}}\left\|x^{t+1} - x^t\right\|^2$$
$$+ p\frac{N-b}{(N-1)b}\sigma_{\mathcal{D}}^2$$

*Proof.* We start by considering two events:

$$H = \mathbb{E}\left[\left\|h^{t+1} - \nabla f_{\mathcal{D}}(x^{t+1})\right\|^2\right] = p\mathbb{E}\left[\left\|\frac{1}{b}\sum_{i\in\mathcal{B}^t}\nabla f_{\mathbf{S}_i^t}(x^{t+1}) - \nabla f_{\mathcal{D}}(x^{t+1})\right\|^2\right]$$
$$+ (1-p)\mathbb{E}\left[\left\|h^t + \frac{1}{b'}\sum_{i\in(\mathcal{B}^t)'}\left(\nabla f_{\mathbf{S}_i^t}(x^{t+1}) - \nabla f_{\mathbf{S}_i^t}(x^t)\right) - \nabla f_{\mathcal{D}}(x^{t+1})\right\|^2\right].$$

Using Assumption 5 and Lemma 5, we obtain

$$H = \mathbb{E}\left[\left\|h^{t+1} - \nabla f_{\mathcal{D}}(x^{t+1})\right\|^2\right]$$
$$\leq p\frac{N-b}{(N-1)b}\sigma_{\mathcal{D}}^2 + (1-p)\mathbb{E}\left[\left\|h^t + \frac{1}{b'}\sum_{i\in(\mathcal{B}^t)'}\left(\nabla f_{\mathbf{S}_i^t}(x^{t+1}) - \nabla f_{\mathbf{S}_i^t}(x^t)\right) - \nabla f_{\mathbf{S}_i^t}(x^{t+1})\right\|^2\right]$$
$$\leq p\frac{N-b}{(N-1)b}\sigma_{\mathcal{D}}^2$$
$$+ (1-p)$$
$$\times \mathbb{E}\left[\left\|h^t - \nabla f_{\mathcal{D}}\left(x^t\right) + \frac{1}{b'}\sum_{i\in(\mathcal{B}^t)'}\left(\nabla f_{\mathbf{S}_i^t}(x^{t+1}) - \nabla f_{\mathbf{S}_i^t}(x^t)\right) - \nabla f_{\mathcal{D}}\left(x^{t+1}\right) + \nabla f_{\mathcal{D}}\left(x^t\right)\right\|^2\right]$$
$$\leq p\frac{N-b}{(N-1)b}\sigma_{\mathcal{D}}^2 + (1-p)\left\|h^t - \nabla f_{\mathcal{D}}\left(x^t\right)\right\|^2$$
$$+ (1-p)\mathbb{E}\left[\left\|\frac{1}{b'}\sum_{i\in(\mathcal{B}^t)'}\left(\nabla f_{\mathbf{S}_i^t}\left(x^{t+1}\right) - \nabla f_{\mathbf{S}_i^t}\left(x^t\right)\right) - \nabla f_{\mathcal{D}}\left(x^{t+1}\right) + \nabla f_{\mathcal{D}}\left(x^t\right)\right\|^2\right]$$
$$\leq p\frac{N-b}{(N-1)b}\sigma_{\mathcal{D}}^2 + (1-p)\left\|h^t - \nabla f_{\mathcal{D}}\left(x^t\right)\right\|^2$$
$$+ p\frac{N-b'}{(N-1)b'}\mathbb{E}\left[\frac{1}{N}\sum_{i\in(\mathcal{B}^t)'}\left\|\left(\nabla f_{\mathbf{S}_i^t}\left(x^{t+1}\right) - \nabla f_{\mathbf{S}_i^t}\left(x^t\right)\right) - \left(\nabla f\left(x^{t+1}\right) - \nabla f\left(x^t\right)\right)\right\|^2\right]$$
$$\leq p\frac{N-b}{(N-1)b}\sigma_{\mathcal{D}}^2 + (1-p)\left\|h^t - \nabla f_{\mathcal{D}}\left(x^t\right)\right\|^2$$
$$+ p\frac{N-b'}{(N-1)b'}\mathbb{E}\left[\frac{1}{N}\sum_{i\in(\mathcal{B}^t)'}\left\|\left(\nabla f_{\mathbf{S}_i^t}\left(x^{t+1}\right) - \nabla f_{\mathbf{S}_i^t}\left(x^t\right)\right)\right\|^2\right].$$

Let us consider the last term:

$$
\mathbb{E}\left[\left\|\nabla f_{\mathbf{S}_i^t}\left(x^{t+1}\right) - \nabla f_{\mathbf{S}_i^t}\left(x^t\right)\right\|^2\right]
$$

$$
= \mathbb{E}\left[\left\|\left((\mathbf{S}_i^t)^\top \nabla f\left(v + \mathbf{S}_i^t(x^{t+1} - v)\right) - (\mathbf{S}_i^t)^\top \nabla f\left(v + \mathbf{S}_i^t(x^t - v)\right)\right)\right\|^2\right]
$$

$$
= \mathbb{E}\left[\left\|(\mathbf{S}_i^t)^\top \left(\nabla f\left(v + \mathbf{S}_i^t(x^{t+1} - v)\right) - \nabla f\left(v + \mathbf{S}_i^t(x^t - v)\right)\right)\right\|^2\right]
$$

$$
\leq \mathbb{E}\left[\lambda_{\max}\left[(\mathbf{S}_i^t)(\mathbf{S}_i^t)^\top\right] \left\|\left(\nabla f\left(v + \mathbf{S}_i^t(x^{t+1} - v)\right) - \nabla f\left(v + \mathbf{S}_i^t(x^t - v)\right)\right)\right\|^2\right]
$$

$$
\leq L_{\mathbf{S}}^{\max}\mathbb{E}\left[\left\|\left(\nabla f\left(v + \mathbf{S}_i^t(x^{t+1} - v)\right) - \nabla f\left(v + \mathbf{S}_i^t(x^t - v)\right)\right)\right\|^2\right].
$$

Using Lipschitz continuity of gradient of function $f$, we have

$$
\mathbb{E}\left[\left\|\nabla f_{\mathbf{S}_i^t}\left(x^{t+1}\right) - \nabla f_{\mathbf{S}_i^t}\left(x^t\right)\right\|^2\right] \leq L_{\mathbf{S}}^{\max} L_f^2 \mathbb{E}\left[\left\|\mathbf{S}_i^t(x^{t+1} - x^t)\right\|^2\right]
$$

$$
\leq L_{\mathbf{S}}^{\max} L_f^2 \lambda_{\max}\left[\mathbb{E}\left[\mathbf{S}_i^t(\mathbf{S}_i^t)^\top\right]\right] \left\|x^{t+1} - x^t\right\|^2
$$

$$
= L_{\mathbf{S}}^{\max} L_f^2 L_{\mathcal{D}} \left\|x^{t+1} - x^t\right\|^2.
$$

Plugging this inequality leads us to the final result:

$$
\mathbb{E}\left[\left\|h^{t+1} - \nabla f_{\mathcal{D}}(x^{t+1})\right\|^2\right] \leq (1-p)\left\|h^t - \nabla f_{\mathcal{D}}(x^t)\right\|^2
$$

$$
+ \frac{N - b'}{(N-1)b'}(1-p) L_f^2 L_{\mathbf{S}}^{\max} L_{\mathcal{D}} \left\|x^{t+1} - x^t\right\|^2
$$

$$
+ p\frac{N - b}{(N-1)b}\sigma_{\mathcal{D}}^2.
$$

$\square$

**Theorem 20.** *Assume that $f$ is $L_f$-smooth* (2) *and* **S** *satisfy Assumptions* 1 *and* 5. *Then, for stepsize* $\gamma \leq \frac{1}{\sqrt{\frac{1-p}{pb'}}L_f\left(L_{\mathcal{D}} + \sqrt{L_{\mathbf{S}}^{\max} L_{\mathcal{D}}}\right)}$, *the iterates of Algorithm* 4 *satisfy*

$$
\mathbb{E}\left[\left\|\nabla f_{\mathcal{D}}\left(\widehat{x}_T\right)\right\|^2\right] \leq \frac{2\mathbb{E}\left[\Psi_0\right]}{\gamma T} + \frac{N - b}{(N-1)b}\sigma_{\mathcal{D}}^2.
$$

*Proof.*

$$
\mathbb{E}\left[\Psi^{t+1}\right]
$$

$$
= \mathbb{E}\left[f_{\mathcal{D}}(x^{t+1}) - f_{\mathcal{D}}^{\inf} + \frac{\gamma}{2p}\left\|g^{t+1} - \nabla f_{\mathcal{D}}(x^{t+1})\right\|^2\right]
$$

$$
\leq \mathbb{E}\left[f_{\mathcal{D}}(x^t) - f_{\mathcal{D}}^{\inf} - \frac{\gamma}{2}\left\|\nabla f_{\mathcal{D}}(x^t)\right\|^2 - \left(\frac{1}{2\gamma} - \frac{L_{f_{\mathcal{D}}}}{2}\right)\left\|x^{t+1} - x^t\right\|^2 + \frac{\gamma}{2}\left\|g^t - \nabla f_{\mathcal{D}}(x^t)\right\|^2\right]
$$

$$
+ \mathbb{E}\left[\frac{\gamma}{2p}\left((1-p)\left\|g^t - \nabla f_{\mathcal{D}}(x^t)\right\|^2 + \frac{(1-p)L_{\mathbf{S}}^{\max}L_{\mathcal{D}}}{b'}\left\|x^{t+1} - x^t\right\|^2 + p\frac{N-b}{(N-1)b}\sigma_{\mathcal{D}}^2\right)\right]
$$

$$
= \mathbb{E}\left[f_{\mathcal{D}}(x^t) - f_{\mathcal{D}}^{\inf} - \frac{\gamma}{2}\left\|\nabla f_{\mathcal{D}}(x^t)\right\|^2 + \frac{\gamma}{2p}\left((1-p)\left\|g^t - \nabla f_{\mathcal{D}}(x^t)\right\|^2 + p\left\|g^t - \nabla f_{\mathcal{D}}(x^t)\right\|^2\right)\right]
$$

$$
+ \mathbb{E}\left[\frac{\gamma}{2}\frac{N-b}{(N-1)b}\sigma_{\mathcal{D}}^2 - \left(\frac{1}{2\gamma} - \frac{L_{f_{\mathcal{D}}}}{2} - \frac{(1-p)\gamma L_{\mathbf{S}}^{\max}L_{\mathcal{D}}}{2pb'}\right)\left\|x^{t+1} - x^t\right\|^2\right].
$$

Using stepsize $\gamma \leq \frac{1}{\sqrt{\frac{1-p}{pb'}} L_f \left( L_{\mathcal{D}} + \sqrt{L_{\mathbf{S}}^{\max} L_{\mathcal{D}}} \right)}$ and Lemma, we get

$$\mathbb{E}\left[\Psi^{t+1}\right] \leq \mathbb{E}\left[\Psi^t - \frac{\gamma}{2} \left\|\nabla f_{\mathcal{D}}(x^t)\right\|^2 + \frac{\gamma}{2} \frac{N-b}{(N-1)b} \sigma_{\mathcal{D}}^2\right] \tag{43}$$

$$\frac{\gamma}{2} \left\|\nabla f_{\mathcal{D}}(x^t)\right\|^2 \leq \mathbb{E}\left[\Psi^t\right] - \mathbb{E}\left[\Psi^{t+1}\right] + \frac{\gamma}{2} \frac{N-b}{(N-1)b} \sigma_{\mathcal{D}}^2 \tag{44}$$

$$\frac{\gamma}{2} \sum_{t=0}^{T-1} \left\|\nabla f_{\mathcal{D}}(x^t)\right\|^2 \leq \mathbb{E}\left[\Psi^T\right] - \mathbb{E}\left[\Psi^0\right] + \frac{\gamma}{2} \frac{N-b}{(N-1)b} \sigma_{\mathcal{D}}^2. \tag{45}$$

Let $\widehat{x}_T$ be randomly chosen from $\{x^t\}_{t \in [T]}$, we have

$$\mathbb{E}\left[\left\|\nabla f_{\mathcal{D}}\left(\widehat{x}_T\right)\right\|^2\right] \leq \frac{2\mathbb{E}\left[\Psi_0\right]}{\gamma T} + \frac{N-b}{(N-1)b} \sigma_{\mathcal{D}}^2.$$

$\square$

## H ADDITIONAL EXPERIMENTS AND DETAILS

First, we provide additional details on the experimental settings from Section 6.

### H.1 EXPERIMENTAL DETAILS

In Section 6 and for all further experiments, the following problem is considered:

$$f(x) \stackrel{\text{def}}{=} \frac{1}{n} \sum_{i=1}^{n} \log\left(1 + \exp(-\mathbf{A}_i^\top x \cdot b_i)\right) + \frac{\lambda}{2} \|x\|^2, \tag{46}$$

where $\mathbf{A}_i \in \mathbb{R}^d, b_i \in \{-1,1\}$ are the feature and label of $i$-th data point. The approximate "optimal" $f^\star$ solution of optimization problem (46) is obtained by running Accelerated Gradient Descent (Nesterov, 1983) until $\|\nabla f^\star\|^2 \leq 10^{-30}$. Our implementation is based on the public Github repository of Konstantin Mishchenko. Simulations were performed on a machine with $24\,\text{Intel(R)}\,\text{Xeon(R)}\,\text{Gold}$ 6246 CPU @ 3.30 GHz.

**Sketches.** Problem (2) may not be easily solvable precisely for the most general sketches. Therefore, we consider the scenario when $\mathcal{D}$ is uniform and has finite support similar to Section G. We use a special class of diagonal permutation sparsifiers formally introduced in the following example:

**Example 3.** *Assume[1] that $K$ divides $d$, let $q \stackrel{\text{def}}{=} d/K$ and $\pi = (\pi_1, \ldots, \pi_d)$ be a random permutation of $[d]$. Then for all $i \in [K]$, we define **Permutation sparsification** (in short* $\mathtt{Perm\text{-}K}$*) operator as*

$$\mathbf{S}_i \stackrel{\text{def}}{=} K \cdot \sum_{j=q(i-1)+1}^{qi} e_{\pi_j} e_{\pi_j}^\top, \tag{47}$$

*where $e_1, \ldots, e_d \in \mathbb{R}^d$ are standard unit basis vectors.*

In simple words $\mathtt{Perm\text{-}K}$ sparsifiers require random shuffling and then dividing[2] the set of indices $[d] = \{1, \ldots, d\}$ into $K$ non-overlapping subsets $C_i \subset \mathcal{G}$ of equal size such that $\cup_{C_i \in \mathcal{G}} C_i = [d]$. Then, every sketch $\mathbf{S}_i$ is formed from $e_l$ vectors based on indexes $l \in C_i$, resulting in $\mathcal{D} = \{\mathbf{S}_1, \ldots, \mathbf{S}_K\}$. To ensure unbiasedness, sketches are sampled uniformly with probability $1/K$. This class of sketches satisfies $L_{\mathcal{D}} = \mu_{\mathcal{D}} = K$ and $L_{\mathbf{S}}^{\max} = K^2$.

**Details for Figure 1 (and 3, 4).** Regularization parameter $\lambda$ in (46) is set to guarantee that the condition number of the loss function $\kappa_f$ is equal to $10^2$. The dataset is shuffled and split to train and test in 0.75 to 0.25 proportions. Initial model weights $x^0 \in \mathbb{R}^d$ are generated from a standard

---

[1]This is done for simplicity of presentation and can be easily generalized (see Appendix I.1 from Szlendak et al. (Szlendak et al., 2022)).

[2]We use array_split method from NumPy (version 1.26.2) package (Harris et al., 2020).

Gaussian distribution $\mathcal{N}(0,1)$. For every sparsity level `Perm-K` sketches $\mathcal{D} = \{\mathbf{S}_1, \ldots, \mathbf{S}_K\}$ are generated leading to different MAST problem formulations. This process is repeated 10 times with various permutations $\pi$ for changing random seeds. After ERM and MAST models $x$ are obtained the accuracy of sparsified model $\mathbf{S}_i x$ is calculated for every $i \in [K]$ on the test set. This results in distributions of accuracies for every sparsity level.

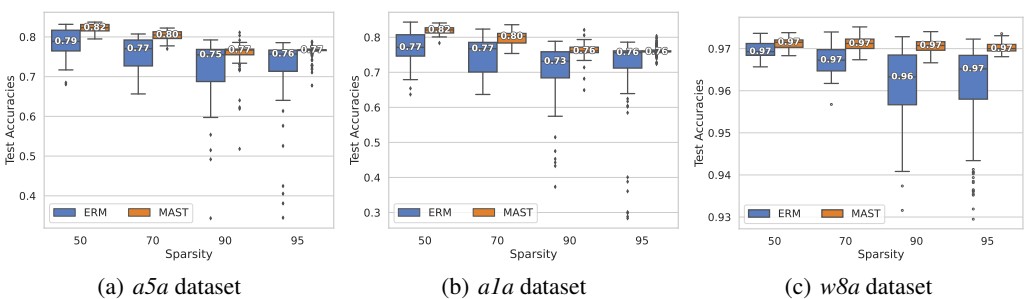

(a) *a5a* dataset      (b) *a1a* dataset      (c) *w8a* dataset

Figure 3: Accuracies distributions of sparsified solutions for the ERM (1) and MAST (2) formulations.

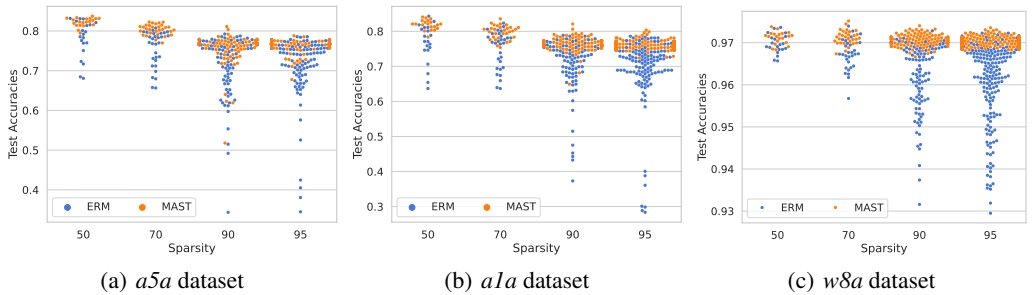

(a) *a5a* dataset      (b) *a1a* dataset      (c) *w8a* dataset

Figure 4: Test accuracies of sparsified solutions for the ERM formulation (1) and MAST problem (2).

In Figure 3, we display complete results, including the accuracies lower than 0.5 (unlike in Section 6), which occurred only for the most aggressive sparsification of ERM models. In addition, the median of accuracies (in white) is shown for every (ERM/MAST) approach and sparsification level. Additional results for *a1a* and *w8a* datasets are consistent with those in Section 6 as MAST models' performance is higher, less variable, and more resistant to sparsification. Figure 4 also shows the swarmplots for the same experiments to represent accuracy values' distributions better.

## H.2 ADDITIONAL EXPERIMENTS

### H.2.1 MAST LOSS TRAJECTORY

In the next experiment, we investigate the optimization efficacy of Double Sketched Gradient Descent (Algorithm 1 (II)) with `Perm-K` sketches (47) for MAST formulation (2) with $f$ chosen as (46) for several datasets. The model weights are divided into $K = 10$ groups, which allows us to solve the MAST problem, find $f_{\mathcal{D}}^{\text{inf}}$, and evaluate $f_{\mathcal{D}}, \nabla f_{\mathcal{D}}$ precisely. Moreover, unlike the previous experiment, the inexactness of the stochastic gradient estimator is introduced via uniform (single element) random subsampling of data $f_i$ as the problem enjoys finite-sum representation (17).

Figure 5 shows the trajectory of a method which averages across all sketches $\mathbf{S}$ which results in exact (w.r.t. $\mathcal{D}$) gradient estimator $\nabla f_{i,\mathcal{D}}$ (right column on legend) and the same algorithm but with uniform sketch subsampling $\nabla f_{i,\mathbf{S}}$ (left column). The methods are run with 3 different step sizes $\gamma$. Subsampling of data introduces oscillations preventing the algorithms from converging to the exact optimum. Sketch subsampling leads to additional variance highlighted by the curves corresponding to the same step size. Moreover, our results clearly indicate the necessity for decreasing the learning rate $\gamma$ for sparse/dropout training with SGD. The method with sketch subsampling and standard SGD

step size (dotted blue curve) fluctuates around initialization, while full averaging across sketches (dotted cyan curve) fixes the issue partially as the error floor is lower however, there is still almost no convergence. Scaling $\gamma$ inversely proportionally to sparsity level $L_{\mathcal{D}} = K$ results in clear linear convergence to the neighborhood of the solution for $\nabla f_{i,\mathcal{D}}$ estimator. However, $\nabla f_{i,\mathbf{S}}$ requires decreasing the step size by $L_{\mathbf{S}}^{\max} = K^2$ which well agrees with conclusions of Theorems 4 and 12.

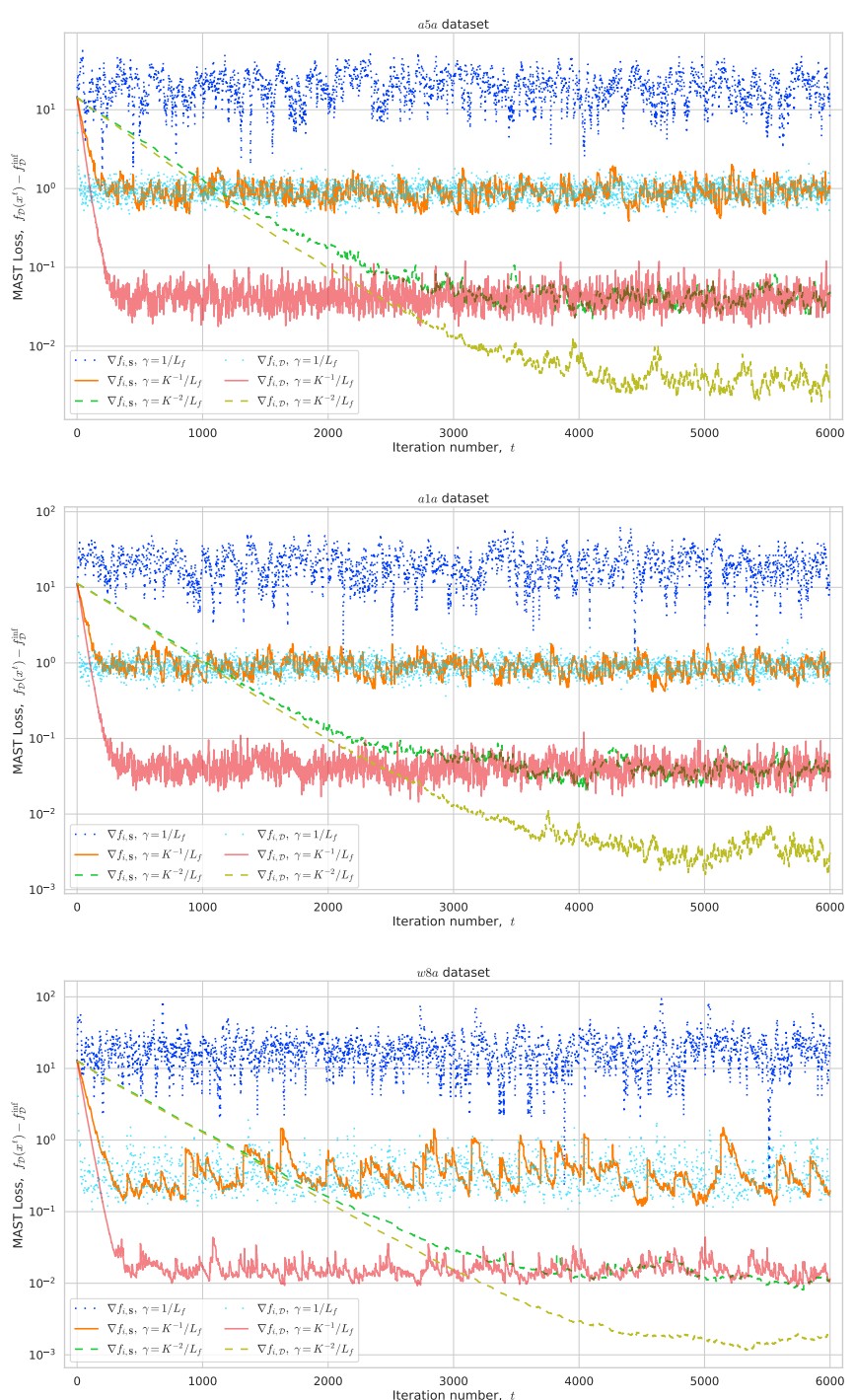

Figure 5: Finite-sum MAST loss (17) convergence for Algorithm 1 (II) with subsampling.

### H.2.2 RAND-$K$ SKETCHES

In this experiment depicted in Figure 6, we validate the claims of Theorem 2. We consider the same logistic regression optimization problem (46) with $\lambda$ set to make sure $\kappa_f = 10^3$. We use the whole dataset *a1a* and initialization $x^0 = 0$. Consider $\mathcal{D}$ as a uniform distribution over Rand-$K$ sketches for $K = 1$. Then, MAST stochastic optimization formulation (2) leads to a finite-sum problem over sketches $\mathbf{S}_i$, as defined in (17). This allows us to evaluate the performance of Algorithm 1 (I), which converges linearly for the exact MAST loss (2). Note that applying Gradient Descent requires computing double sketched gradient for all possible ($N = d$) sketches. We denote step sizes as $\widehat{\gamma}_0 = 1/(L_f L_{\mathcal{D}})$ and $\gamma_0 = 1/(L_f L_{\mathbf{S}}^{\max})$ according to theory.

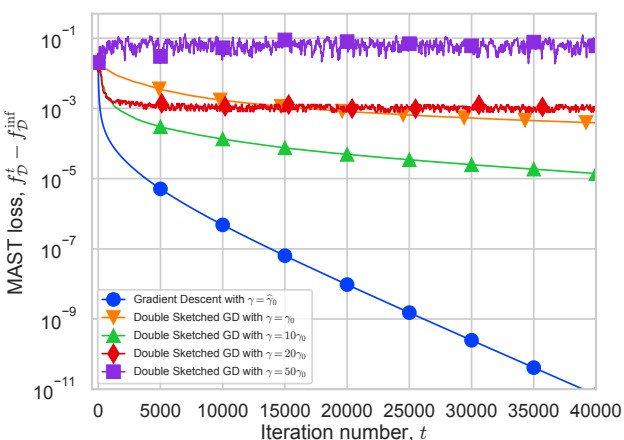

Figure 6: MAST loss (2) convergence for Algorithm 1 (I) with varying step size and Rand-1 sketches.

Our findings accentuate the pivotal role of the appropriate step size $\gamma$ in steering the trajectory of sparse training. Guided by the proposed theoretical framework, this step size must be adjusted in proportion to $K^2/d^2$ for Rand-$K$. Notably, for this particular problem at hand, a larger $\gamma$ (e.g., $\gamma = 10\gamma_0$) can accelerate convergence. Yet, surpassing a delineated boundary can result in stagnation of the progress (e.g., $\gamma = 20\gamma_0$) and, in specific scenarios, even derail the convergence altogether (e.g., $\gamma = 50\gamma_0$). Such observations underscore the imperative of modulating the learning rate, especially within the realms of sparse and Dropout training.

### H.2.3 NEURAL NETWORKS

This section presents our deep learning experimental results. Figure 7 illustrates the loss behavior of the *distributed* Algorithm 2 (for $M = 10$ clients) using Bernoulli sketches (6) for $p_i \equiv p$ on the standard loss (18) (for $\mathbf{S}_i \equiv \mathbf{I}$). Our experimental setup closely follows that of Liao & Kyrillidis (2022), and for completeness, we reiterate key details. We employ a ResNet-50 model (He et al., 2016) pre-trained on ImageNet as a feature extractor, concatenated with two fully connected layers. This combined model is then fine-tuned on the CIFAR-10 dataset (Krizhevsky et al., 2009). The outputs of the re-trained ResNet-50 serve as input embeddings, while the logit outputs of the combined model are used as labels.

The first column of Figure 7 shows how the method's performance is affected by the sparsity level ($p$) and step size ($\gamma$). Specifically, for high sparsity $p = 0.5$, Figure 7(a) illustrates that an excessively large step size ($\gamma = 1$) may even lead to divergence of the method for ERM loss. Across all sparsity levels, we observe a "sweet spot" for the step size, beyond which increasing $\gamma$ results in slower convergence. Furthermore, training with high sparsity ($p = 0.5$) leads to a quick stagnation of the loss in contrast to $p \in \{0.7, 0.9\}$. The second column of Figure 7 displays the subsequent divergence (for $p = 0.5$), indicating that high sparsity significantly alters the minimized loss, confirming that Sparse/Dropout training indeed optimizes a different formulation than standard ERM.

In general, larger step sizes and more aggressive sparsification (lower $p$) result in increased loss variance, aligning with our theoretical predictions from Sections 2 and 4. Interestingly, figs. 7(d) and 7(f) reveal that Dropout training can outperform non-sparse optimization for small step sizes ($\gamma = 0.01$) or initial iterations (up to $\sim 1200$ for $\gamma = 0.1$). However, the largest step size ($\gamma = 0.5$) is the most efficient in terms of minimizing canonical loss.

One of the key practical insights derived from our theoretical analysis is that the step size $\gamma$ (learning rate) must be decreased for sparse optimization and Dropout training. Our neural network training results demonstrate that this insight extends to a broader range of models.

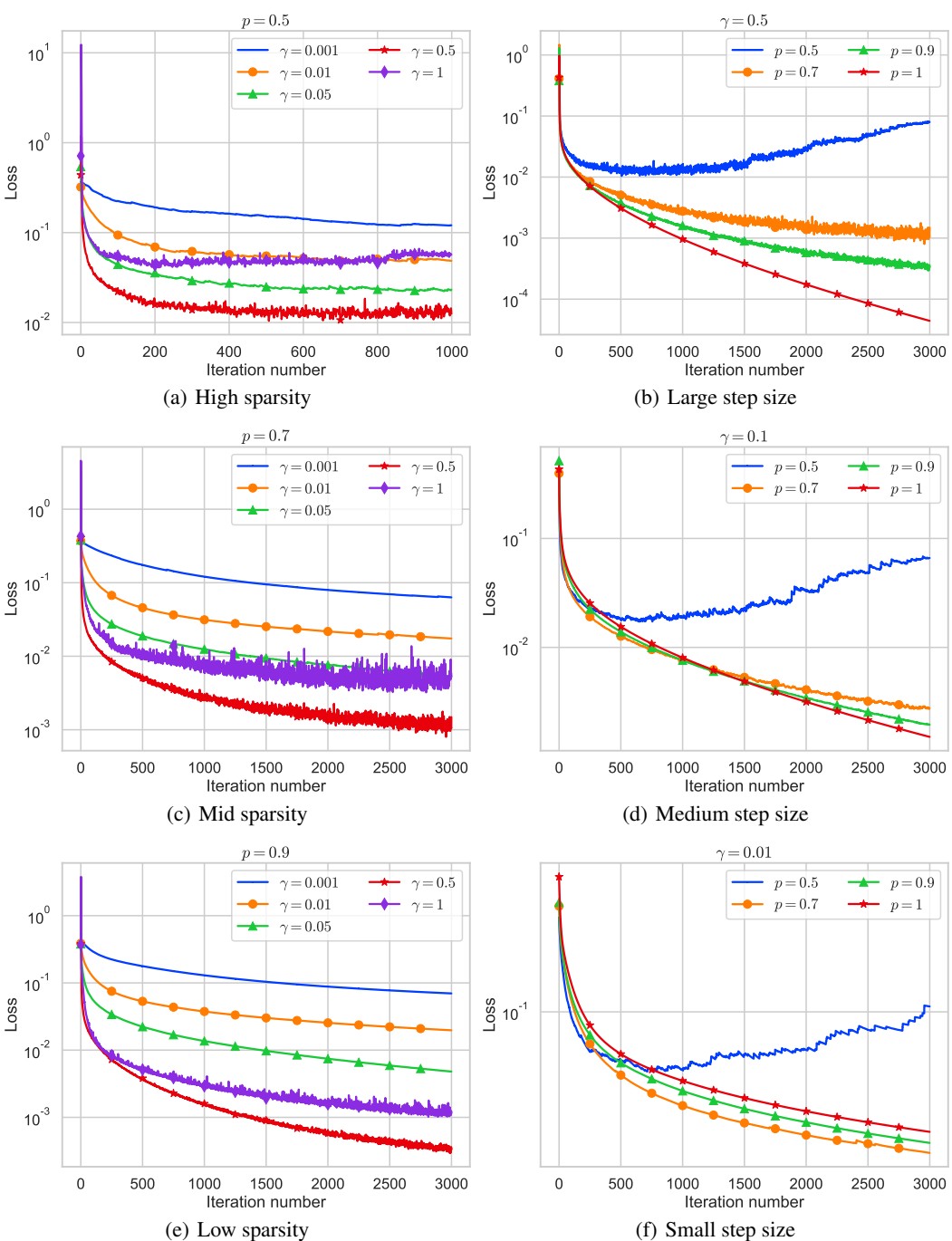

Figure 7: Performance of Algorithm 2 with Bernoulli sketches (6) on standard loss (18) (for $\mathbf{S}_i \equiv \mathbf{I}$).

### H.2.4 STANDARD VS UNBIASED DROPOUT

We supplement the results from Section H.2.3, by comparing unbiased scaled (6) and biased Dropout sketches in the same setup by running distributed Algorithm 2 on standard loss (18) (for $\mathbf{S}_i \equiv \mathbf{I}$). Figure 8 shows the training loss curves for different sparsity levels ($p = 0.7, 0.9$) and learning rates ($\gamma = 0.05, 0.5, 1.0$). The unbiased estimator includes a $1/p$ scaling factor, while the biased version omits this scaling as in the original Dropout (Hinton et al., 2012).

The results demonstrate that the unbiased estimator (solid purple lines) consistently achieves lower loss values compared to the biased estimator (red lines with markers). This effect is particularly pronounced at lower sparsity ($p = 0.7$), while at higher sparsity ($p = 0.9$) the difference becomes less dramatic. Higher learning rates lead to increased variance in both estimators, though the unbiased approach maintains better overall performance, which aligns with our previous observations about the impact of sparsity and learning rates on optimization dynamics.

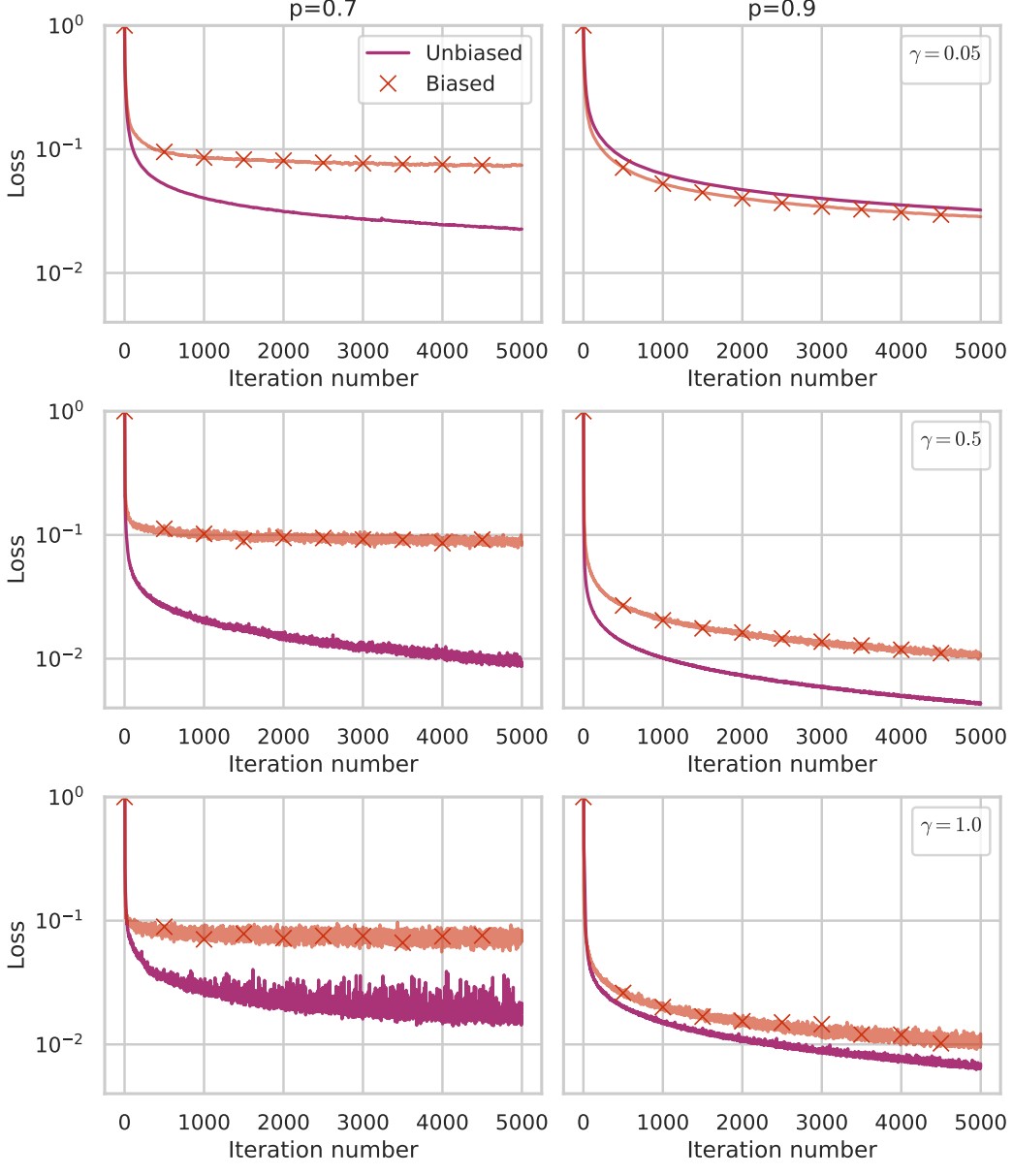

Figure 8: Comparison of the unbiased sketches (6) and original (biased) Dropout.

