# OpenReview forum: "MAST: model-agnostic sparsified training"
_ICLR.cc/2025/Conference — ICLR 2025 Poster_

### Official Review · Reviewer_8W9V · 2024-10-27

**Soundness:** 3
**Presentation:** 3
**Contribution:** 3
**Rating:** 8
**Confidence:** 4

**Summary:**

This paper presents an optimization framework that incorporates model sparsification using pre-trained models and stochastic sketches for enhanced computational efficiency. The authors propose an objective function that departs from traditional approaches, aiming to improve convergence rates in stochastic gradient descent (SGD) for sparse or resource-constrained environments. They also provide theoretical analyses on the convergence of their proposed algorithms and demonstrate potential applications for both centralized and distributed machine learning scenarios.

**Strengths:**

The paper certainly brings a fresh perspective by introducing a sparsification-aware objective function that enables gradient sparsification during training. This idea is both innovative and timely, addressing the growing need for more efficient model training techniques.

On the theoretical side, the authors provide a strong mathematical foundation. They successfully derive convergence bounds for various setups, including challenging non-convex scenarios and methods that reduce variance, which adds significant depth to their work.

The experimental results are compelling and back up the theoretical claims. They show that the MAST method outperforms traditional approaches, especially when dealing with high levels of sparsity. This makes the findings relevant and applicable to real-world situations.

**Weaknesses:**

One downside is that the experimental setup doesn't include a wide range of comparisons to other sparsification or dropout methods beyond ERM. Adding these comparisons would help better highlight where MAST excels and where it might fall short. The experiments are also somewhat limited in terms of the neural network architectures tested. The study focuses on logistic regression and ResNet-50, but there's not a strong justification for this choice.

**Questions:**

1. Can you give an idea of computational and memory overheads of using sketch matrices and sparsification compared to traditional methods?

2. I'm curious about how MAST performs with larger, non-convex models like deep neural networks. Testing it on a complex neural network, even briefly, could demonstrate its applicability beyond logistic regression.

3. How robust is MAST to variations in hyperparameters like step size and sparsity, especially in heterogeneous distributed environments?

---

> ### Author Response · Authors · 2024-11-20
>
> Dear Reviewer 8W9V,
>
> Thank you for taking the time to review our paper.  We appreciate your positive evaluation of the theoretical contributions and experimental results. Below, we address the mentioned weaknesses and respond to your questions.
> ### Comments on **Weaknesses**
>
> - > One downside is that the experimental setup doesn't include a wide range of comparisons to other sparsification or dropout methods beyond ERM. The experiments are also somewhat limited in terms of the neural network architectures tested. The study focuses on logistic regression and ResNet-50, but there's not a strong justification for this choice.
>
> The primary purpose of our work is to propose and investigate a novel theoretical framework (MAST) that lays the groundwork for understanding widely used practical techniques such as Dropout and Sparse training. Consequently, our main focus is on the theoretical analysis and algorithmic contributions, rather than proposing a new state-of-the-art method for practical applications. Nonetheless, we complement our theoretical findings with illustrative experiments to validate our framework.
>
> In Section 6, we note that *“we focus on carefully controlled settings that satisfy the assumptions of our work.”* Specifically, $\ell_2$-regularized logistic regression serves as an example of a strongly convex optimization problem, aligning with Theorem 2 and illustrating the performance of the proposed methods in a controlled, real-data setting. Logistic regression is one of the most widely used classification machine learning models in practice. Moreover, the LIBSVM [project](https://www.csie.ntu.edu.tw/~cjlin/libsvmtools/datasets/) provides diverse datasets suitable for testing optimization methods.
>
> Additionally, as many machine learning problems are non-convex, such as training neural networks, we extend our theoretical guarantees to non-convex settings in Theorems 3 and 4. These are illustrated experimentally in the second part of Section 6 (“Neural network results”) and Appendix H.2.3. ResNet-50 was chosen for its prevalence and relevance in computer vision tasks, making it a natural choice for demonstrating the applicability of MAST in a non-convex context.

---

> ### Author Response · Authors · 2024-11-20
>
> ### Answers to **Questions**
> - > Can you give an idea of computational and memory overheads of using sketch matrices and sparsification compared to traditional methods?
>
> For sparse diagonal sketches considered in this work, the per-iteration computational and memory overheads scale with the model dimension $d$. However, the specific overheads depend on the implementation and the type of sketch used.
>
> For example, Bernoulli independent sketches can be sampled at the start of each optimization step for the entire model, requiring storage for $d$ binary random variables—significantly less than the full-precision momentum and normalization parameters used in optimizers like Adam (potentially up to 32 times smaller). The computational overhead is similar to that of standard Dropout. Alternatively, if random seeds are used, these binary variables can be generated per layer without storing them, further reducing memory requirements. This makes our framework scalable even for extremely large models.
>
> - > I'm curious about how MAST performs with larger, non-convex models like deep neural networks. Testing it on a complex neural network, even briefly, could demonstrate its applicability beyond logistic regression.
>
> We provide experimental results for neural networks in Section 6 (second part) and Appendix H.2.3. These results demonstrate the effectiveness of MAST on complex non-convex models, including ResNet-50, which is widely used in computer vision.
>
> MAST is not a new method but a way to better understand how current sparsification techniques, like dropout and sparse training, work. It provides a theoretical explanation for why these methods are effective, focusing on their underlying principles instead of introducing new techniques. This helps improve and optimize the methods we already use.
>
> - > How robust is MAST to variations in hyperparameters like step size and sparsity, especially in heterogeneous distributed environments?
>
> The robustness of MAST to variations in step size and sparsity is explored in Section 6 and Appendix H. For distributed optimization (Algorithm 2), Figures 2 and 6 illustrate how the step size and sparsity level influence performance.
>
> Specifically, Figure 2 shows that MAST methods are robust to changes in step size, with convergence observed across a broad range of values (up to $\gamma = 1$) under moderate sparsity for Bernoulli sketches. However, sparsity has a more pronounced effect on convergence. Higher sparsity increases the variance of the loss and necessitates smaller step sizes; for extreme sparsification, the method may diverge with respect to the original (ERM) loss even with the smallest explored step size.
>
> Interestingly, we observe that the optimal step size range (around $\gamma = 1$) remains consistent across different sparsity levels, demonstrating the transferability of hyperparameters between settings.
>
> We hope this response clarifies our contributions and addresses your concerns. Please, feel free to share further suggestions or questions.
>
> Best regards, Authors

---

### Official Review · Reviewer_Lu4H · 2024-11-01

**Soundness:** 3
**Presentation:** 4
**Contribution:** 2
**Rating:** 6
**Confidence:** 4

**Summary:**

This work presents MAST, which optimizes the expected value of a function $f$ composed with a sketching matrix $S$. The authors analyze theoretical properties and develop a solution algorithm, demonstrating its effectiveness for sparse neural network training. The authors also provide experiments to validate their method.

**Strengths:**

The paper is well motivated, and the formalization of the problem is interesting. The authors provide a thorough analysis of the problem properties and derive an algorithm based on SGD to solve it. The assumptions are clearly stated and the theoretical results are well proven.

**Weaknesses:**

While I appreciate this research, I have a few concerns and questions regarding the contribution and novelty of the work, which I explain in detail below. I hope that these remarks are helpful in improving the work and I am happy to discuss my evaluation.

- MAST bears a clear relationship to standard (stochastic) gradient descent. In supervised learning, the objective is to minimize the expected loss of a model with respect to the distribution of features and labels. Similarly, MAST aims to minimize the expected value of a function $f$ composed with a sketching matrix $S$, where the expectation is taken over the distribution of the sketching matrices. This could be made more clear in the paper.
- While Assumption 1 appears valid since it shows $S^T\nabla f(x)$ is an unbiased estimator of $\nabla f(x)$, there is a discrepancy in Example 1. The authors state their sketching matrix $S$ formulation models Dropout, but to the best of my knowledge, standard Dropout applies a binary mask to activations without rescaling weights. The connection between these approaches needs clarification in their problem formulation.
- The problem properties in section 3 appear to be correct. However, apart from Theorem 1,these are well known properties of smooth and strongly convex functions. It is not clear how novel the results in this section are, given that the proofs are rather elementary and known.
- In Theorem 1, it is assumed that Assumption 2 and Assumption 3 hold, in particular that $f$ is strongly convex, implying there is a unique minimizer $x^*$. Why is then an optimal set of solutions $\mathcal{X}^*$ mentioned? By Lemma 3, the same should hold true for the solutions to the MAST problem. Can the theorem be simplified or made more precise?
- The paper is well written and the proofs appear to be correct. I find the setting interesting, especially that it allows modelling sparsification. However, it is not entirely clear to me how novel the results are, as many of the results are well known properties of smooth and strongly convex functions.

I appreciate the experiments, however there are a few points that could be improved:
- The plots in the experiments should have the same scale to allow for a comparison of the methods.
- Did the authors compare the method to standard gradient descent with Dropout? Since MAST introduces a rescaling of the gradient and that is not present in standard gradient descent with Dropout, it is not clear how the two methods compare to each other.
- Currently, the experiments seem to be somewhat limited. Can the authors provide more experiments, potentially at a larger scale?

**Questions:**

- The authors formulate MAST with a general sketch matrix $S$, but in the experiments they only consider diagonal matrices as sketching matrices. Could the authors provide examples of non-diagonal sketching matrices that could be useful in practice?

---

> ### Author Response · Authors · 2024-11-19
>
> Dear Reviewer Lu4H,
>
> Thank you for the time and effort devoted to reviewing our paper. We appreciate your careful reading, thoughtful evaluation, and constructive feedback. Below, we address the mentioned weaknesses and respond to your questions individually.
>
> ### Comments on Weaknesses
>
> - > MAST aims to minimize the expected value of a function $f$ composed with a sketching matrix $S$ … This could be made more clear in the paper.
>
> Thank you for this suggestion. We agree and will revise the text to make this clearer. Specifically, note that the loss function $f$ can also include an expectation over the distribution of samples. Thus, MAST inherently captures both components: sketches and data distribution, which we will emphasize in the revised version.
>
> - >  standard Dropout applies a binary mask to activations without rescaling weights. The connection between these approaches needs clarification in their problem formulation.
>
> Thank you for this insightful comment! You are correct that Example 1 in our paper differs from the original Dropout in that it applies rescaling during training to satisfy Assumption 1, ensuring unbiasedness, which is needed for theoretical analysis. Additionally, original Dropout applies sparsification to activations, whereas our formulation applies it to weights. However, as demonstrated in the DropConnect paper, Dropout can be generalized and is effectively equivalent to weight sparsification for most activation functions, including ReLU, Tanh, centered Sigmoid, GELU, etc. We will clarify this connection in the revised version of the paper.
>
> - > ... apart from Theorem 1, these are well-known properties of smooth and strongly convex functions. It is not clear how novel the results in this section are
>
> It has been known that the composition with an affine mapping preserves convexity and smoothness (Boyd & Vandenberghe, 2004). However, in our paper we analyze it from a perspective of a novel **stochastic optimization** problem. First, we adapt this theory to the machine learning context by introducing a pre-trained model shift ($v$) and a sketching (sparsifying) matrix $\mathbf{S}$. Second, we provide a precise characterization of how the spectral properties of the sketch and its distribution ($\mathcal{D}$) affect the smoothness and (strong) convexity of the resulting loss functions $f_{\mathbf{S}}$ and $f_{\mathcal{D}}$. For example, we demonstrate that more aggressive sparsification makes the problem harder to optimize.
> Additionally, Lemmas 1 and 3 (iii) quantify the impact of the shift vector $v$ on the gap between the original ERM and MAST formulations. We believe these contributions are novel. If the reviewer is aware of prior works presenting similar results, we would be grateful for relevant pointers.
>
> - > It is not entirely clear to me how novel the results are.
>
> The main novelty of our work lies in the introduction of a new optimization **problem formulation**, which naturally leads to stochastic methods incorporating sketched models and gradients. This enables us to derive meaningful theoretical results, contrasting with prior approaches that suffer from overly pessimistic or vacuous convergence bounds for sparsified model training. Additionally, we propose two novel optimization algorithms with variance reduction for sketches (Appendix G).
>
> - > In Theorem 1, it is assumed that Assumption 2 and Assumption 3 hold, in particular, that $f$ is strongly convex, implying there is a unique minimizer $x^\star$. Why is an optimal set of solutions $\mathcal{X}^\star$ mentioned?
>
> Thank you for this observation. You are correct that in the strongly convex case, the minimizer is unique, and the sets $\mathcal{X}^\star$ and $\mathcal{X}^\star_{\mathcal{D}}$ each contain a single element. However, Theorem 1 is also valid in the convex case ($\mu_f = 0$), where the minimizer may not be unique. We will clarify this distinction in the revised version.
>
> - > The plots in the experiments should have the same scale to allow for a comparison of the methods.
>
> Thank you for pointing this out. We will revise the plots to ensure consistent scales across all figures in the camera-ready version.
>
> - > Did the authors compare the method to standard gradient descent with Dropout?
>
> We appreciate this suggestion and plan to include comparisons to standard gradient descent with Dropout in the camera-ready version.
>
> - > Can the authors provide more experiments, potentially at a larger scale?
>
> We would like to note that additional experimental results are provided in Appendix H. We can expand our evaluation further, including larger-scale experiments, and include these in the appendix due to space constraints. If the reviewer has specific suggestions for additional experiments, we would be grateful to hear them.
>
> We hope this response addresses your concerns and clarifies our contributions. Please feel free to reach out with further questions or suggestions.
>
> Best regards, Authors

---

> > ### Author Response · Authors · 2024-11-19
> > **Response to question**
> >
> > ### Answer to Questions
> >
> > - > The authors formulate MAST with a general sketch matrix $S$, but in the experiments, they only consider diagonal matrices as sketching matrices. Could the authors provide examples of non-diagonal sketching matrices that could be useful in practice?*
> >
> > Our focus in this paper is on diagonal sketches, as they provide a straightforward means of sparsification, which is of the primary practical interest. However, our theoretical analysis covers any sketch distribution satisfying Assumption 1. Non-diagonal matrices can be particularly useful in accounting for dependencies among coordinates of the model or gradient vector.
> >
> > Examples of non-diagonal sketching matrices used in practice include:
> >
> > 1. **Gaussian Sketches:** These can replace Bernoulli with Gaussian random variables and have been shown to outperform standard Dropout in certain scenarios (e.g., [ICML 2013](https://proceedings.mlr.press/v28/wang13a.html)).
> >
> > 2. **Structured Random Rotation Matrices:** These reduce compression error and are effective in distributed settings (e.g., [https://arxiv.org/abs/1611.00429](https://arxiv.org/abs/1611.00429)).
> >
> > 3. **Preconditioner-Inspired Sketches:** These can incorporate smoothness information to speed up optimization, similar to the approach in [https://arxiv.org/abs/1809.09354](https://arxiv.org/abs/1809.09354).

---

> > > ### Comment · Reviewer_Lu4H · 2024-11-22
> > >
> > > Thank you for your answers to my questions.
> > >
> > > > You are correct that Example 1 in our paper differs from the original Dropout in that it applies rescaling during training to satisfy Assumption 1, ensuring unbiasedness, which is needed for theoretical analysis. Additionally, original Dropout applies sparsification to activations, whereas our formulation applies it to weights. However, as demonstrated in the DropConnect paper, Dropout can be generalized and is effectively equivalent to weight sparsification for most activation functions, including ReLU, Tanh, centered Sigmoid, GELU, etc. We will clarify this connection in the revised version of the paper.
> > >
> > > Thanks. While DropConnect relates to weight sparsification, it does so without scaling. Standard Dropout, using a Bernoulli mask, therefore does not seem to fit within the MAST framework.
> > >
> > > > [...] Additionally, Lemmas 1 and 3 (iii) quantify the impact of the shift vector on the gap between the original ERM and MAST formulations. We believe these contributions are novel. If the reviewer is aware of prior works presenting similar results, we would be grateful for relevant pointers. [...]
> > >
> > > Thank you for the clarification. The proofs of Lemma 1 and Lemma 3 rely on strong convexity, smoothness, and linearity of expectation. These seem to be standard techniques and elementary.
> > >
> > > > The main novelty of our work lies in the introduction of a new optimization problem formulation, which naturally leads to stochastic methods incorporating sketched models and gradients. This enables us to derive meaningful theoretical results, contrasting with prior approaches that suffer from overly pessimistic or vacuous convergence bounds for sparsified model training. Additionally, we propose two novel optimization algorithms with variance reduction for sketches (Appendix G).
> > >
> > > While the approach of optimizing the expected value of a function composed with a sketching matrix is interesting, I think that the theoretical contributions are somewhat limited. The proofs rely on standard properties of smooth and strongly convex functions. Additionally, the assumptions on the sketch matrix seem to exclude common approaches like Dropout, which uses unscaled binary masks rather than the rescaled matrices described in the paper. This limits the practical applicability of the theoretical framework.
> > >
> > > Despite these remarks, the authors have answered my questions and partially addressed my concerns. I am not entirely convinced that the theoretical contributions are that interesting, but I like the clarity of the paper and have decided to increase my score to 6.

---

> > > > ### Author Response · Authors · 2024-11-25
> > > > **Authors' response**
> > > >
> > > > Thank you very much for your thoughtful comment and for improving the score of our work—we sincerely appreciate it!
> > > >
> > > > We would like to clarify that the rescaling introduced in our paper is crucial to ensuring that the sketch remains unbiased. While we agree that exploring biased sketching is an intriguing idea, it lies beyond the scope of this study. Nevertheless, we believe it represents an exciting and promising direction for future research.
> > > >
> > > > Additionally, we would like to highlight that our paper includes a detailed analysis of non-convex functions (Theorems 3, 4, 5). Furthermore, we proposed a variance reduction technique to mitigate the randomness in sketching, which is applicable to strongly convex, convex, and non-convex cases (Theorems 18, 19, 20).
> > > >
> > > > If you have any further questions or need additional clarification, please do not hesitate to reach out. We greatly value your feedback and are happy to engage in further discussion.

---

### Official Review · Reviewer_j9zw · 2024-11-03

**Soundness:** 2
**Presentation:** 3
**Contribution:** 1
**Rating:** 6
**Confidence:** 4

**Summary:**

This paper introduced a new optimization formulation called MAST that considers a modified loss function $\mathbb{E}[f_S(x)] =\mathbb{E}[f(v+S(x-v))] $ rather than a standard loss $f(x)$. Here, $S$ is a random matrix sampled from distribution $\mathcal{D}$, and $v$ can be viewed as a pre-trained model. The paper claims that this problem formulation can model various practical machine learning techniques including Dropout and Sparse training with a sparse matrix $S$.

Main theoretical results include comparisons between the modified loss formulation and the standard loss, as well as convergence analysis in both non-convex and (strongly) convex settings. The paper also extends convergence analysis in the distributed setup. Experimental studies are also provided to test the proposed algorithms and MAST properties.

**Strengths:**

The presentation of the paper is good, and all the proofs are well-organized and written clearly.

**Weaknesses:**

1. It is unclear why a linear transformation $v+S(x-v)$ in $f_S$ is good enough to model practical ML techniques that authors consider in this paper. The reason for putting such a linear transformation rather than more complicated nonlinear transformations is lacking.

2. Because of this simple linear transformation, the resulted comparison and convergence analysis are quite standard and lack novelties, so is for the distributed setup. The authors also admit that the convergence results are standard for convex and non-convex analysis.

3. Rigorous mathematical connections between the model and Dropout and Sparse training are missing. Since both Dropout and Sparse training are techniques in the neural network training, it would be nice to quantitatively analyze how the model $f_S$ approximate the sparsified neural networks.

**Questions:**

1. I wonder if the convergence results could be extended under weaker assumptions on $f$, perhaps by assuming that $f$ only satisfies Lojasiewicz-type conditions?

2. Is it true that $f_{\mathcal{D}}^{inf}\geq f^{\inf}$? I don't see that in the second line of Theorem 1, but it is needed for (13), Theorem 3 and Corollary 1.

---

> ### Author Response · Authors · 2024-11-19
>
> Dear Reviewer j9zw,
>
> Thanks for your time and effort. Let us reply to the questions and weaknesses raised separately.
>
> >**It is unclear why a linear transformation $v + S(x - v) \)$ in $f_S$ is good enough to model practical ML techniques that authors consider in this paper. The reason for putting such a linear transformation rather than more complicated nonlinear transformations is lacking.**
>
> We respectfully disagree with the statement that linear transformations are not effective for modeling ML sparsification techniques. Allow us to reiterate our motivation and clarify our approach.
>
> As we discuss in **Example 1 (Section 2.1)** of our paper, Dropout is effectively modeled through a linear transformation, where $S$ is chosen as an independent Bernoulli sparsification diagonal operator. Similarly, **Example 2** demonstrates that Sparse Training can be modeled using a Random $K$-sparsification diagonal operator. These examples underscore that linear transformations in $f_S$ are sufficiently descriptive for these techniques, obviating the need for non-linear transformations in this context.
>
> Additionally, most computation in DL involves linear operations, particularly matrix multiplications. Given the large scale of modern DL models, the use of sparse diagonal matrices ensures efficiency, adding only an $\mathcal{O}(d)$ computational cost—negligible compared to the resource-intensive operations in fully connected layers. Thus, our choice of linear transformation is motivated not only by its theoretical sufficiency but also by practical considerations, as it avoids introducing significant computational overhead. We will include these clarifications in the revised version of the paper.
>
> We also respectfully disagree with the critique regarding the exclusion of non-linear transformations. As a conference paper, the work is subject to strict page limits, and it is not feasible to address all possible modeling approaches within this format. Our focus is on introducing a novel formulation and connecting it to real-world applications through illustrative examples and discussions across several settings. While we recognize the value of exploring non-linear transformations, we believe this is a direction better suited for future work.
>
> We kindly request your understanding that it is impractical to cover all possible topics within a single paper. We appreciate your feedback and will incorporate additional clarifications as outlined above to address these concerns.

---

> ### Author Response · Authors · 2024-11-19
>
> >**Because of this simple linear transformation, the resulted comparison and convergence analysis are quite standard and lack novelties, so is for the distributed setup. The authors also admit that the convergence results are standard for convex and non-convex analysis.**
>
> We respectfully and firmly disagree with the statement that our analysis is standard and lacks novelty. Allow us to clarify and provide context for our contributions.
>
> First and foremost, we propose a novel optimization problem formulation, which is neither standard nor trivial. Within this framework, we introduce several methods tailored to different settings, further demonstrating the originality and non-standard nature of our work. At no point do we, as authors, claim that our results are standard. While certain steps in our analysis may resemble those found in gradient-based method studies—a common feature in optimization research—our work incorporates unique and interesting proof techniques, particularly regarding expectations with respect to random sketches, which we have not encountered before.
>
> It is natural in science to build upon prior work, and optimization research is no exception. However, this does not imply that our contributions are trivial or lack significance. Moreover, our work extends the ideas in [1], which were well-received by the optimization and ML communities. This further validates the relevance and interest of our approach.
>
> Our convergence analysis is distinct in that it is performed for $f_D​$ rather than $f,$ as in prior literature. To the best of our knowledge, this makes our results novel. The focus of our work is on developing a new problem formulation that effectively models techniques like Dropout and Sparsified Training. For this formulation, we construct a consistent and systematic theoretical framework, as detailed in **Section 3**. The results presented in **Lemmas 1–3** and **Theorem 1** are novel and, in some cases, non-trivial to derive. This foundational work enables us to build upon standard convergence analyses while still requiring auxiliary techniques and additional steps, such as those in **Lemma 13**, which also elucidate how convergence rates depend on the operator $S.$
>
> Our analysis examines (strong) convexity and smoothness within the context of a novel **stochastic optimization** problem (MAST). Specifically, we reinterpret the problem in the ML context using the "shift" vector $v$ from a pre-trained model and a sketching matrix $S.$ We characterize, with precision, how the spectral properties of individual sketches and their distribution $D$ influence the smoothness and (strong) convexity of the resulting functions $f_S​$ and $f_D​,$ and how these properties impact convergence. Notably, we show that more aggressive sparsification of sketches leads to a more challenging optimization problem. Additionally, we explore the role of the "shift" vector $v$ on the loss function in **Lemmas 1 and 3 (iii)**, quantifying the gap between the original empirical risk minimization (ERM) problem and MAST.
>
> This framework is versatile, general, and applicable to practical problems involving Dropout and Sparsified Training. It provides theoretical insights valuable to implementers and may also interest researchers in related fields, such as On-device Learning. Our perspective and contributions are, therefore, novel and capable of stimulating further studies.
> Finally, we note that **Appendix G** contains the first-ever analysis of the method (Algorithm 3) equipped with variance reduction for sketches. This represents an entirely new contribution, explicitly mentioned in our abstract.
>
> In conclusion, while it is natural for science to build on previous work, our contributions go beyond replication or marginal improvement. They expand upon the ideas in [1], advancing the field with theoretical and practical insights that address real-world challenges in ML optimization.
>
> [1] Egor Shulgin, Peter Richtárik,  Towards a Better Theoretical Understanding of Independent Subnetwork Training. ICML, 2024.

---

> ### Author Response · Authors · 2024-11-19
>
> >**Rigorous mathematical connections between the model and Dropout and Sparse training are missing. Since both Dropout and Sparse training are techniques in the neural network training, it would be nice to quantitatively analyze how the model $f_S$ approximates the sparsified neural networks.**
>
> We would like to point out that $f:\mathbb{R}^d\to\mathbb{R}$ is not a model but a loss function of a model with parameters/weights $x,$ as we write in the Introduction.
>
> Dropout is typically implemented by randomly masking certain neurons with zeros during the training process. This masking operation sets a percentage of the neuron's outputs to zero, effectively "dropping out" these neurons from contributing to the forward pass and backpropagation. Multiplication by the diagonal Bernoulli sketches is equivalent to it and described in **Example 1** of our paper. Sparse training is characterized by fixing the portion of network parameters and is modeled by the Random $K$ sparsifying operator described in **Example 2** of our paper. We believe our explanations there are rigorous.
>
> The paper describes the theoretical connection between $f_{\mathcal{D}}(x)$ and $f(x)$ in **Lemmas 1 - 3** and in **Theorem 1**. After the statement of **Theorem 1** we provide a discussion paragraph, where we explain the connection between sketched loss $f_D$ based on Random $K$ sparsifying operators (model the Sparse training in **Example 2**) and the original function $f.$. We mathematically demonstrate how sparsity level affects the relation between the sparsified model and the original one. The bounds are obtained on both sides and allow theorists and practitioners in the field to assess their sparsified model performance. Additionally, in **Appendix H** we supply experiments that include Rand-K and Perm-K sparsification. **Figures 3 and 4** there show how different levels of sparsification affect the model performance. Namely, more aggressive sparsity degrades the test accuracy of ERM solutions, but MAST models are more robust.
>
> >**I wonder if the convergence results could be extended under weaker assumptions on $f$, perhaps by assuming that $f$ only satisfies Łojasiewicz-type conditions?**
>
> Yes, we already have the analysis in the weak setup for general non-convex functions, relying only on the $L$-smoothness assumption. This setup is weaker than the PŁ condition with smoothness. We believe that the analysis of the Polyak–Łojasiewicz (PŁ) condition and $L$-smoothness does not provide significant additional insights compared to the analysis of the strongly convex case with $L$-smoothness (**Theorems 1,2,12,15,17 and 18**) or the non-convex case with $L$-smoothness (**Theorems 3,4,5,7 and 20**). The PŁ condition is essentially a slight generalization of strong convexity, as it relaxes the requirement of a uniformly positive curvature while still ensuring similar convergence guarantees.
>
> >**Is it true that $f_D^{\inf} \geq f^{\inf}$? I don’t see that in the second line of Theorem 1, but it is needed for (13), Theorem 3 and Corollary 1.**
>
> By the definition, $f_S(x) = f(v + S(x-v)).$ Therefore, for every $S,$ the function $f_S(x)$ attains values of $f(x)$ with an affinely transformed argument. Therefore, the values of $f_S(x)$ can not be smaller than $f^{\inf}.$ In turn, $f_D(x)$ is an average (expectation) of functions $f_S(x)$ over sketches. The average of values that are not smaller than $f^{\inf}$ is not smaller than $f^{\inf}.$
>
> Formally, by the definition $f(x) \geq f^{inf}$, and $f_S(x) = f(v + S(x-v))  \geq f^{inf}$, for any $S.$ Thus $f_D(x) = \mathbb{E} f_S(x) \geq f^{inf}$.
>
> **If you are satisfied with our responses, please consider raising your score accordingly.**
>
> Best regards, authors

---

> ### Author Response · Authors · 2024-11-25
>
> Dear Reviewer j9zw,
>
> Thank you once again for your initial review.
>
> We have provided a detailed response to your comments, and with the discussion period deadline approaching, we would greatly appreciate your feedback on our response. We put significant effort into preparing it.
>
> Your input is invaluable in helping us further improve the work, and we look forward to hearing your thoughts!

---

> > ### Comment · Reviewer_j9zw · 2024-11-26
> > **Thank you for your detailed responses**
> >
> > I greatly appreciate the authors' efforts in addressing my concerns and clarifying the paper. I agree that this work is both interesting and novel from a modeling perspective, and I have raised my score accordingly.
> >
> > However, regarding the theoretical aspects, I maintain my opinion that the results are standard. In particular, for the non-convex case, the paper appears to overstate the significance of the convergence results, as none of them pertain to last-iterate convergence.

---

> > > ### Author Response · Authors · 2024-11-30
> > >
> > > Dear Reviewer j9zw,
> > >
> > > Thank you for your thoughtful discussion, valuable comments, and for increasing the score.

---

> ### Comment · Area_Chair_z7dZ · 2024-11-26
> **Please reply to authors**
>
> Hello Reviewer j9zw,
>
> Thank you very much for your work on reviewing this submission.
>
> Could you read the author's reply, check if they have addressed your comments, and acknowledge it?
>
> Further, please make sure, if necessary, to explain what your new stance is based on this information.

---

### Official Review · Reviewer_w3TH · 2024-11-07

**Soundness:** 3
**Presentation:** 4
**Contribution:** 3
**Rating:** 8
**Confidence:** 3

**Summary:**

This paper introduces Model-Agnostic Sparsified Training (MAST), a novel optimization framework tailored for sparsification in model training. By defining a new objective that incorporates pre-trained models and random sketch operators, MAST allows for simultaneous model and gradient sparsification during training. The authors propose modified SGD-based algorithms to handle this objective, offering both single-node and distributed variants. Experimental results highlight the approach’s efficacy, showing MAST’s superior robustness to pruning compared to standard models.

**Strengths:**

1. **Theoretical contribution:** The framework introduces a structured approach to sparsification in training, with rigorous theoretical analysis of convergence for both convex and non-convex cases.

2. **Relevance:** This paper addresses a very relevant topic in the field.

3. **Scalability:** Extending the method to a distributed setting enables applications in federated learning and distributed subnetwork training.

4. **Experiments:** Experiments validate MAST’s practical benefits, showing improved performance and robustness over traditional approaches.

5. **Presentation:** The paper is very well-written, with clear explanations and a consistent, clean notation.

**Weaknesses:**

Although the assumptions for the theoretical results are standard, they are overly restrictive and may limit real-world applicability.

**Questions:**

Could the MAST framework potentially be extended to conditional computing approaches, such as mixture-of-experts models or sparse gating mechanisms in neural networks? Specifically, do you see any pathways for adapting MAST to support dynamic routing or conditional activation, as seen in these architectures?

---

> ### Author Response · Authors · 2024-11-20
>
> Dear Reviewer w3TH,
>
> Thanks for your time and positive evaluation of our work. Next, we comment on the noted weaknesses and answer the questions.
>
> >**Although the assumptions for the theoretical results are standard, they are overly restrictive and may limit real-world applicability.**
>
> We would like to emphasize that the primary goal of our work is to develop theoretical insights that enhance our understanding of key practical techniques widely used in machine learning (ML). We strongly believe that theoretical analysis and convergence guarantees are essential in science, particularly in ML, as purely experimental results often apply to specific cases and may lack generalizability. The theoretical assumptions in our work, such as smoothness (non-convexity), are among the weakest possible in the optimization field.
> We respectfully disagree with the notion that these assumptions are overly restrictive for real-world applications. For instance, while the convexity assumption in optimization may initially seem limiting, it serves as a foundation for developing efficient algorithms that perform well even on non-convex problems. Let us consider Adam [1] and RMSProp [2], which use momentum. Both methods are highly effective for training deep neural networks (DNNs). While non-convex optimization theory suggests that momentum offers no benefits [3], convex optimization theory shows that it significantly accelerates convergence rates [4]. This theoretical support for momentum in convex settings aligns more closely with its practical success than the non-convex theory does.
> This inconsistency is not unique to momentum-based methods; similar trends can be observed in other leading optimization approaches, such as distributed optimization [5] and adaptive algorithms [6]. The gap between non-convex theoretical guarantees and practical performance stems from the overly broad class of non-convex function, driving research efforts to narrow it through additional assumptions.
>
> [1] Diederik P. Kingma, Jimmy Ba, Adam: A Method for Stochastic Optimization, 2014.
>
> [2] Hinton, G., Srivastava, N., and Swersky, K.,Neural networks for machine learning lecture 6a overview of mini-batch gradient descent, 2012.
>
> [3] Carmon, Y., Duchi, J. C., Hinder, O., and Sidford, A., Lower bounds for finding stationary points, 2020.
>
> [4] Nesterov, Y., A method for unconstrained convex minimization problem with the rate of convergence o (1/k2), 1983.
>
> [5] Mishchenko, K., Malinovsky, G., Stich, S., and Richtárik, P., Proxskip: Yes! local gradient steps provably lead to communication acceleration! Finally!, 2022.
>
> [6] Defazio, A., Mishchenko, K., Learning-Rate-Free Learning by D-Adaptation, 2023.
>
> >**Could the MAST framework potentially be extended to conditional computing approaches, such as mixture-of-experts models or sparse gating mechanisms in neural networks? Specifically, do you see any pathways for adapting MAST to support dynamic routing or conditional activation, as seen in these architectures?**
>
> Thank you for the insightful question!
>
> We believe that the MAST framework can be extended to conditional computation approaches in future research. One possible direction would involve introducing sketches that activate or deactivate parts of the network on a per-example basis. These sketches could be binary or sparse and continuous, as well as stochastic or deterministic. Additionally, one could introduce several experts, $E_1(x),\ldots,E_m(x)$ with $E_i(x) = f(v+S_i(x-v)),$ and each random sketch $S_i$ has its own distribution $D_i$ that depends on the specific example. The gating network (the router) would then learn which sketch distribution (expert) is most appropriate for a given input. The expectation in MAST formulation would be taken with respect to sketch distributions of the product of the router output vector with the vector of experts.
>
> We trust this response clarifies our contributions and resolves your concerns. Please do not hesitate to share any additional suggestions or questions!
>
> Best regards,
> Authors

---

### Meta-Review · Area_Chair_z7dZ · 2024-12-20

**Metareview:**

This paper presents MAST, a new optimization framework for sparsification in model training. MAST offers a theoretically sound approach with potential applications in various machine learning scenarios. While the theoretical assumptions might be restrictive for some real-world problems and the empirical evaluation could be broadened, the paper's novel framework and promising results warrant acceptance. The reviewers were enthusiastic about this work.

**Additional Comments On Reviewer Discussion:**

There was some productive discussion between reviewers and authors, no major points had to be solved.

---

### Decision · Program_Chairs · 2025-01-22

Accept (Poster)